# DEFT: Efficient Fine-Tuning of Diffusion Models by Learning the Generalised $h$-transform

**Alexander Denker***
University College London
a.denker@ucl.ac.uk

**Francisco Vargas***
University of Cambridge
fav25@cam.ac.uk

**Shreyas Padhy***
University of Cambridge
sp2058@cam.ac.uk

**Kieran Didi***
University of Cambridge
ked48@cam.ac.uk

**Simon Mathis***
University of Cambridge
svm34@cam.ac.uk

**Vincent Dutordoir**
University of Cambridge
vd309@cam.ac.uk

**Riccardo Barbano**
Atinary Technologies
rbarbano@atinary.com

**Emile Mathieu**
University of Cambridge
ebm32@cam.ac.uk

**Urszula Julia Komorowska**
University of Cambridge
ujk21@cam.ac.uk

**Pietro Lio**
University of Cambridge
pl219@cam.ac.uk

## Abstract

Generative modelling paradigms based on denoising diffusion processes have emerged as a leading candidate for *conditional* sampling in inverse problems. In many real-world applications, we often have access to large, expensively trained unconditional diffusion models, which we aim to exploit for improving conditional sampling. Most recent approaches are motivated heuristically and lack a unifying framework, obscuring connections between them. Further, they often suffer from issues such as being very sensitive to hyperparameters, being expensive to train or needing access to weights hidden behind a closed API. In this work, we unify conditional training and sampling using the mathematically well-understood *Doob's h-transform*. This new perspective allows us to unify many existing methods under a common umbrella. Under this framework, we propose DEFT *(Doob's h-transform Efficient FineTuning)*, a new approach for conditional generation that simply fine-tunes a very small network to quickly learn the conditional $h$-transform, while keeping the larger unconditional network unchanged. DEFT is much faster than existing baselines while achieving state-of-the-art performance across a variety of linear and non-linear benchmarks. On *image reconstruction* tasks, we achieve speedups of up to $1.6\times$, while having the best perceptual quality on natural images and reconstruction performance on medical images. Further, we also provide initial experiments on protein motif scaffolding and outperform reconstruction guidance methods.

## 1 Introduction

Denoising diffusion models are a powerful class of generative models where noise is gradually added to data samples until they converge to pure noise. The time reversal of this noising process then

---

*equal contributions

38th Conference on Neural Information Processing Systems (NeurIPS 2024).

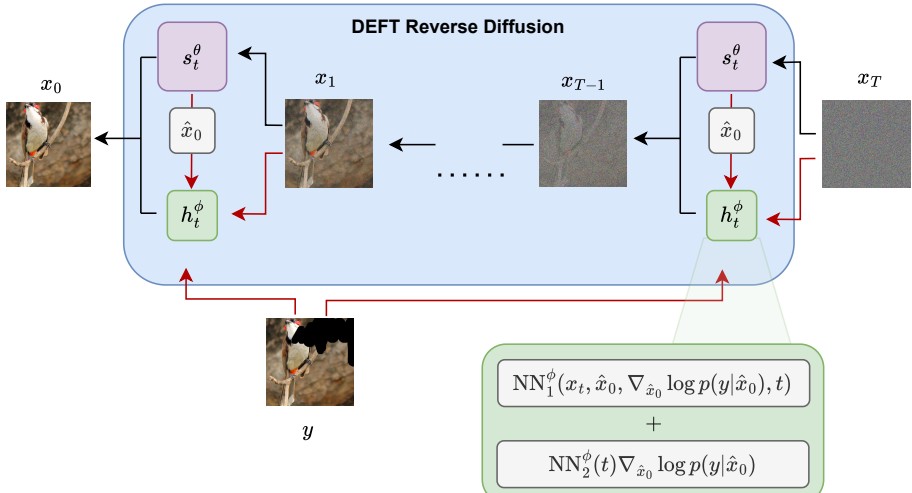

Figure 1: DEFT reverse diffusion setup. The pre-trained unconditional diffusion model $s_t^\theta$ and the fine-tuned $h$-transform $h_t^\phi$ are combined at every sampling step. We propose a special network to parametrise the $h$-transform including the guidance term $\nabla_{\hat{\boldsymbol{x}}_0} \ln p(\boldsymbol{y}|\hat{\boldsymbol{x}}_0)$ as part of the architecture. Here $\hat{\boldsymbol{x}}_0$ denotes the unconditional denoised estimate given $s_t^\theta(\boldsymbol{x}_t)$. During training, we only need to fine-tune $h_t^\phi$ (usually 4-9% the size of $s_t^\theta$) using a small dataset of paired measurements, keeping $s_\theta^t$ fixed. During sampling, we do not need to backpropagate through either model, resulting in speed-ups during evaluation.

allows noise to be transformed into samples. This process has been widely successful in generating high-quality images [28] and has more recently shown promise in designing protein backbones that have been validated in experimental protein design workflows [77]. Recently, there has been much interest in *conditioning* the time reversal process, in order to generate samples that are subject to an observed condition. Conditional sampling requires the posterior score $\nabla_{\boldsymbol{x}} \ln p_t(\boldsymbol{x}|\boldsymbol{Y} = \boldsymbol{y})$, given some observation $\boldsymbol{y}$. As diffusion models typically approximate the score of the underlying distribution, i.e., $s_t^{\theta^*}(\boldsymbol{x}) \approx \nabla_{\boldsymbol{x}} \ln p_t(\boldsymbol{x})$, a pre-trained diffusion model can be leveraged using Bayes' theorem

$$\nabla_{\boldsymbol{x}} \ln p_t(\boldsymbol{x}|\boldsymbol{Y} = \boldsymbol{y}) \approx s_t^{\theta^*}(\boldsymbol{x}) + \nabla_{\boldsymbol{x}} \ln p_t(\boldsymbol{Y} = \boldsymbol{y}|\boldsymbol{x}), \tag{1}$$

to approximate the posterior score. The time-dependent likelihood $\nabla_{\boldsymbol{x}} \ln p_t(\boldsymbol{Y} = \boldsymbol{y}|\boldsymbol{x})$ is often termed *guidance* due to its interpretation to guide the reverse process to the conditioned inputs, and is unfortunately analytically intractable. To tackle this problem, several approximations for the *guidance* have been proposed; see, for example, [12, 22, 29, 56, 65, 69] and further discussion in Appendix B. Instead of relying on the decomposition (1), another line of work aims to learn the posterior score directly [5, 27], which requires expensive training for new conditional sampling tasks, and access to large amounts of paired data points.

In the setting of conditional generation with diffusion models, our primary goal is to leverage large pre-trained foundation models which are prevalent in applications, but which typical front-end users are not able to backpropagate through, making approaches like [12, 22] infeasible. This might be due to their prohibitive computation times or because they lie behind an API preventing the usage of autodiff frameworks.

In this work, we propose a unified framework for conditional generation using Doob's $h$-transform, a well-known result in the stochastic differential equations (SDE) literature [14, 55, 61, 78]. Under this framework, we propose DEFT *(Doob's h-transform Efficient FineTuning)*, an algorithm that estimates the time-dependent likelihood directly from data, i.e., $h^* = \nabla_{\boldsymbol{x}} \ln p_t(\boldsymbol{Y} = \boldsymbol{y}|\boldsymbol{x})$, while being able to leverage an existing pre-trained unconditional model. We learn the guidance term ($h$-transform) efficiently using 1) smaller networks, and 2) a small training dataset of paired data points and corresponding observations. Furthermore, through connections to stochastic control, we propose a novel network architecture for general-purpose fine-tuning, which, in conjunction with our

proposed loss, achieves competitive results across a series of inverse problems in imaging and protein design, while having a much lower computational cost.

## 2 Conditioning diffusions via the $h$-transform

In this section, we explore the formal mechanism to condition the boundary points of an SDE mathematically, and connect it to existing methodologies for conditioning diffusions in generative modelling. For a more rigorous background to denoising diffusion models, see Appendix A. Let us first recap the score-based generative modelling framework of [68]; we start with a forward SDE, which progressively transforms the target distribution $\mathcal{P}_0$ (e.g. $\mathcal{P}_0 = p_{\text{data}}$)

$$\mathrm{d}\boldsymbol{X}_t = f_t(\boldsymbol{X}_t)\,\mathrm{d}t + \sigma_t\overrightarrow{\mathrm{d}\mathbf{W}}_t, \quad \boldsymbol{X}_0 \sim \mathcal{P}_0, \tag{2}$$

with drift $f_t$ and diffusion $\sigma_t$. Under some regular assumptions, there exists a corresponding reverse SDE with corresponding drift $\bar{b}_t$ [2], that allows us to take samples from $\mathcal{P}_T$ (typically $\mathcal{N}(0, \mathbf{I})$) and denoise them to generate samples from $\mathcal{P}_0$,

$$\mathrm{d}\boldsymbol{X}_t = \left(f_t(\boldsymbol{X}_t) - \sigma_t^2 \nabla_{\boldsymbol{X}_t} \ln p_t(\boldsymbol{X}_t)\right)\mathrm{d}t + \sigma_t\overleftarrow{\mathrm{d}\mathbf{W}}_t, \quad \boldsymbol{X}_T \sim \mathcal{P}_T, \tag{3}$$

where the time flows backwards, and $\bar{b}_t = f_t(\boldsymbol{X}_t) - \sigma_t^2 \nabla_{\boldsymbol{X}_t} \ln p_t(\boldsymbol{X}_t)$. The goal of conditional sampling is to condition the reverse SDE on a particular observation, i.e., to produce samples that satisfy constraints. For example, we might want to use (3) to generate samples where we already know some dimensions of the sample (e.g. knowing some pixels of the image a-priori in image inpainting). Doob's $h$-transform [55, 14] provides a formal mechanism for conditioning an SDE to hit an event at a given time. We will show that existing methods for conditional generative modelling arise as approximate instances of this proposed framework. Formally, we have:

**Proposition 2.1.** *(Doob's $h$-transform [55]) Consider the reverse SDE in Eqn. (3). The conditioned process $\boldsymbol{X}_t | \boldsymbol{X}_0 \in B$ is a solution of*

$$\mathrm{d}\boldsymbol{H}_t = \left(\bar{b}_t(\boldsymbol{H}_t) - \sigma_t^2 \boxed{\nabla_{\boldsymbol{H}_t} \ln \overleftarrow{p}_{0|t}(\boldsymbol{X}_0 \in B | \boldsymbol{H}_t)}\right)\mathrm{d}t + \sigma_t\overleftarrow{\mathrm{d}\mathbf{W}}_t, \quad \boldsymbol{H}_T \sim \mathcal{P}_T, \tag{4}$$

*with a backward drift $\bar{b}_t(\boldsymbol{H}_t) = f_t(\boldsymbol{H}_t) - \sigma_t^2 \nabla_{\boldsymbol{H}_t} \ln p_t(\boldsymbol{H}_t)$, such that $\mathrm{Law}\,(\boldsymbol{H}_s | \boldsymbol{H}_t) = \overleftarrow{p}_{s|t,0}(\boldsymbol{x}_s | \boldsymbol{x}_t, \boldsymbol{x}_0 \in B)$ and $\mathbb{P}(\boldsymbol{X}_0 \in B) = 1$.*

Note, that we will refer to the conditional process with $\boldsymbol{H}_t$ and to the unconditional process with $\boldsymbol{X}_t$. Doob's $h$-transform shows that by conditioning a diffusion process to hit a particular event $\boldsymbol{X}_0 \in B$ at a boundary time, the resulting conditional process is itself an SDE with an *additional drift term* (shown in the blue box above). Furthermore, the resulting SDE will hit the specified event within a finite time $T$. The function $h(t, \boldsymbol{H}_t) \triangleq \overleftarrow{P}_{0|t}(\boldsymbol{X}_0 \in B \mid \boldsymbol{H}_t)$ is referred to as the *h-transform* [55, 14]. See also Appendix C.3 for a discussion about the connection to reconstruction guidance methods.

Rather than conditioning an SDE on a deterministic event, one is often interested in a posterior arising from noisy observations (e.g. noisy inverse problems)

$$\boldsymbol{Y} = \mathrm{noisy}(\mathcal{A}(\boldsymbol{X}_0)), \; \boldsymbol{X}_0 \sim p_{\text{data}}, \tag{5}$$

where $\mathcal{A}$ is a forward operator, "noisy" describes a noise process and unlike the classical $h$-transform, we are not enforcing a deterministic condition such as $\mathcal{A}(\boldsymbol{X}_0) = \boldsymbol{Y}$. We typically assume we can evaluate and sample from the likelihood $p(\boldsymbol{y} | \boldsymbol{X} = \boldsymbol{x}_0)$. Our goal is to sample from the posterior $p(\boldsymbol{x}_0 | \boldsymbol{Y} = \boldsymbol{y}) = p(\boldsymbol{y} | \boldsymbol{x}_0) p_{\text{data}}(\boldsymbol{x}_0) / p(\boldsymbol{y})$. Sampling from the posterior $p(\boldsymbol{x}_0 | \boldsymbol{Y} = \boldsymbol{y})$ can be achieved by a generalisation of the $h$-transform that build on results in [75], given as follows:

**Proposition 2.2.** *(Generalised $h$-transform) Given the following backwards SDE with marginals $p_t$*

$$\mathrm{d}\boldsymbol{X}_t = \bar{b}_t(\boldsymbol{X}_t)\,\mathrm{d}t + \sigma_t\overleftarrow{\mathrm{d}\mathbf{W}}_t, \quad \boldsymbol{X}_T \sim \mathbb{P}_T, \tag{6}$$

*then it follows that the backward SDE*

$$\boldsymbol{H}_T \sim Q_T^{f_t}[p(\boldsymbol{x}_0 | \boldsymbol{y})] = \int \overleftarrow{p}_{T|0}(\boldsymbol{x} | \boldsymbol{x}_0) p(\boldsymbol{x}_0 | \boldsymbol{y})\mathrm{d}\boldsymbol{x}_0$$

$$\mathrm{d}\boldsymbol{H}_t = \left(\bar{b}_t(\boldsymbol{H}_t) - \sigma_t^2 \boxed{\nabla_{\boldsymbol{H}_t} \ln p_{y|t}(\boldsymbol{Y} = \boldsymbol{y} | \boldsymbol{H}_t)}\right)\mathrm{d}t + \sigma_t\overleftarrow{\mathrm{d}\mathbf{W}}_t, \tag{7}$$

*satisfies* Law $(\boldsymbol{H}_0) = p(\boldsymbol{x}_0|\boldsymbol{Y} = \boldsymbol{y})$ *with* $p_{y|t}(\boldsymbol{Y} = \boldsymbol{y}|\cdot) = \int p(\boldsymbol{Y} = \boldsymbol{y}|\boldsymbol{x}_0)\bar{p}_{0|t}(\boldsymbol{x}_0|\cdot)\mathrm{d}\boldsymbol{x}_0$. *We recover guidance based diffusions via* $\bar{b}_t(\boldsymbol{H}_t) = f_t(\boldsymbol{H}_t) - \sigma_t^2 \nabla_{\boldsymbol{H}_t} \ln p_t(\boldsymbol{H}_t)$.

Here $Q_T^{f_t}[\pi(\boldsymbol{x}_0)] = \int \bar{p}_{T|0}(\boldsymbol{x}|\boldsymbol{x}_0)\pi(\boldsymbol{x}_0)\mathrm{d}\boldsymbol{x}_0$ is the transition operator of the forward process. Note, that the initial distribution $Q_T^{f_t}[p(\boldsymbol{x}_0|\boldsymbol{y})]$ of the controlled SDE differs from the unconditional SDE. However, in Proposition G.2 we show that for the VP-SDE the difference between them gets exponentially small for increasing $T$. To summarise, the above result gives a generalisation of the $h$-transform that allows sampling from posteriors; notice that it recovers the traditional $h$-transform in the no-noise setting. Whilst this more general formulation of the $h$-transform has been explored in unconditional generative modelling [78], this is the first work to cast conditional generative modelling in this light. We refer to the term in blue as the *generalised h-transform* henceforth.

Proposition 2.2 provides theoretical backing to methodologies such as DPS [12] or ΠGDM [65], in which the reverse SDE (7) is used to solve noisy inverse problems. For a careful derivation of Proposition 2.2 see Appendix D. While prior works have explored using Bayes' rule to decompose the conditional score, we provide rigorous arguments for intermediate steps, and carefully formalise the connection between conditional generative modelling and the $h$-transform, providing a concise result. This framework is flexible enough to also encompass prior work on conditional score matching, see e.g., [5, 27], and the discussion Appendix H.

## 3 Learning the generalised $h$-transform

Prior works either learn the posterior score from scratch, see e.g. [5, 27], or use approximations to the generalised $h$-transform, see e.g. [12, 32]. Instead, we propose a method to learn the generalised $h$-transform. We refer to this process as fine-tuning, as the pre-trained unconditional network remains unchanged and only the approximation to the generalised $h$-transform is learned. Our main result is given in the following theorem, where we give several representations of the generalised $h$-transform.

---

**Theorem 3.1.** *(Representations of conditional SDE sampling) For a given* $\boldsymbol{y} \sim noisy(\mathcal{A}(\boldsymbol{x}_0))$*, let* $\mathbb{Q}$ *be the path measure of the conditional SDE*

$$\mathrm{d}\boldsymbol{H}_t = \left(f_t(\boldsymbol{H}_t) - \sigma_t^2 \left(\nabla_{\boldsymbol{H}_t} \ln p_t(\boldsymbol{H}_t) + h_t(\boldsymbol{H}_t)\right)\right)\mathrm{d}t + \sigma_t \overleftarrow{\mathrm{d}\mathbf{W}}_t, \tag{8}$$

*where* $\boldsymbol{H}_T \sim Q_T^{f_t}[p(\boldsymbol{x}_0|\boldsymbol{y})]$*. The generalised h-transforms admits the following representations:*

1) *The path measure induced by the h-transformed SDE satisfies* $\mathrm{d}\mathbb{Q}^* = \mathrm{d}\mathbb{P}\frac{\mathrm{d}p(\boldsymbol{x}_0|\boldsymbol{y})}{\mathrm{d}\mathbb{P}_0}$*, where* $\mathbb{P}$ *is the path measure of the unconditioned SDE and* $\mathbb{P}_0$ *is it's time* $0$ *marginal.*

2) *The h-transform admits a **denoising score matching** representation*

$$h_t^* = \underset{h_t \in \mathcal{H}}{\arg\min} \mathcal{L}_{SM}^{\boldsymbol{y}}(h_t)$$

$$\mathcal{L}_{SM}^{\boldsymbol{y}}(h_t) := \underset{\substack{\boldsymbol{X}_0 \sim p(\boldsymbol{x}_0|\boldsymbol{y}) \\ t \sim \mathrm{U}(0,T), \boldsymbol{H}_t \sim p_{t|0}(\boldsymbol{x}_t|\boldsymbol{x}_0)}}{\mathbb{E}} \left[ \left\| \left(h_t(\boldsymbol{H}_t) + \nabla_{\boldsymbol{H}_t} \ln p_t(\boldsymbol{H}_t)\right) - \nabla_{\boldsymbol{H}_t} \ln \bar{p}_{t|0}(\boldsymbol{H}_t|\boldsymbol{X}_0) \right\|^2 \right]$$

3) *The h-transform admits the following **stochastic control** formulation*

$$h_t^* = \underset{h_t \in \mathcal{H}}{\arg\min} \left\{ \mathcal{L}_{SC}^{\boldsymbol{y}}(h_t) := \mathbb{E}_{\mathbb{Q}} \left[ \frac{1}{2} \int_0^T \sigma_t^2 ||h_t(\boldsymbol{H}_t)||^2 \mathrm{d}t \right] - \mathbb{E}_{\boldsymbol{H}_0 \sim \mathbb{Q}_0}[\ln p(\boldsymbol{y}|\boldsymbol{H}_0)] \right\},$$

*where* $\mathbb{Q}$ *is the path measure for the conditional SDE being controlled.*

4) *The path measure induced by the h-tranformed SDE solves the a Schrödinger bridge problem with boundary conditions* $\mathbb{Q}_0 = Q_T^{f_t}[p(\boldsymbol{x}_0|\boldsymbol{y})] \approx \mathcal{N}(0, I)$*,* $\mathbb{Q}_T = p(\boldsymbol{x}_0|\boldsymbol{y})$ *and with the unconditional process* $\mathbb{P}$ *as its reference.*

---

Here, 4) and 1) follow directly from [7, 73]. For the proof 2) see Appendix D.2 and for 3) see Appendix G. Under appropriate conditions on the likelihood, the space of admissible controls $\mathcal{H}$

can be taken to be the set of $C_1$-vector fields with linear growth in space; see [48]. In the following sections, we will discuss the representations in 2) and 3) in more detail.

## 3.1 DEFT: Fine-tuning by score matching

The score matching objective in Theorem 3.1 2) offers a simulation-free loss function to estimate the generalised $h$-transform. While the theorem's formulation focuses on learning the $h$-transform for a specific measurement $\boldsymbol{y}$, this loss function can naturally be extended and amortized over the full range of measurements, i.e.,

$$\min_{h \in \mathcal{H}} \mathbb{E}_{\boldsymbol{y} \sim \boldsymbol{Y}}[\mathcal{L}_{\mathrm{SM}}^{\boldsymbol{y}}(h)], \tag{9}$$

to obtain $h_t^*(\boldsymbol{x}, \boldsymbol{y}) = \nabla_{\boldsymbol{x}} \ln p_t(\boldsymbol{y}|\boldsymbol{x})$. Further, for settings where the operator may vary, we can additionally amortise over the forward operator $\mathcal{A} \sim p$ and learn $h_t^*(\boldsymbol{x}, \boldsymbol{y}, \mathcal{A}) = \nabla_{\boldsymbol{x}} \ln p_t(\boldsymbol{y}|\boldsymbol{x}, \mathcal{A})$. We exploit this to amortise over inpainting masks, see Section 4.1, and motif scaffolding, see Section 4.3. For the DDPM [28] discretisation of the SDE and a pre-trained epsilon matching model $\epsilon_t^{\theta^*}$, the fine-tuning objective (9) reduces to

$$\min_{\phi} \mathbb{E}_{(\boldsymbol{X}_0,\boldsymbol{Y}),\epsilon,t} \left[ \|(h_t^{\phi}(\boldsymbol{H}_t, \boldsymbol{Y}) + \epsilon_t^{\theta^*}(\boldsymbol{H}_t)) - \epsilon\|^2 \right], \tag{10}$$

with $\boldsymbol{H}_t = \sqrt{\bar{\alpha}_t}\boldsymbol{X}_0 + \sqrt{1 - \bar{\alpha}_t}\epsilon$, $(\boldsymbol{X}_0, \boldsymbol{Y}) \sim p(\boldsymbol{x}_0, \boldsymbol{y}), \epsilon \sim \mathcal{N}(0, \mathbf{I})$, where $h_t^{\phi}$ represents the neural network used to approximate the generalised $h$-transform. Note that the loss function (10) only requires evaluation of the pre-trained model, without needing to backpropagate through the weights $\theta^*$, which is often quite expensive and sometimes impossible in closed APIs. Training under the DDPM discretisation can be performed according to Algorithm 5. Sampling with DEFT is further explained in Algorithm 6, and pictorially represented in Figure 1. As an additional insight into the behaviour of the $h$-transform that makes it more flexible and capable of modelling non-linear tasks than standard reconstruction guidance methods, we show that the $h$-transform can be interpreted as a correction term for the Tweedie estimate [20]. We can express the conditional Tweedie's estimate as

$$\mathbb{E}[\boldsymbol{x}_0|\boldsymbol{x}_t, \boldsymbol{y}] \approx \hat{\boldsymbol{x}}_0(\boldsymbol{x}_t, \boldsymbol{y})$$
$$= \frac{\boldsymbol{x}_t - \sqrt{1 - \bar{\alpha}_t} \left( h_t^{\phi^*}(\boldsymbol{x}_t, \boldsymbol{y}) + \epsilon_t^{\theta^*}(\boldsymbol{x}_t) \right)}{\sqrt{\bar{\alpha}_t}} = \hat{\boldsymbol{x}}_0(\boldsymbol{x}_t) - \frac{\sqrt{1 - \bar{\alpha}_t}}{\sqrt{\bar{\alpha}_t}} h_t^{\phi^*}(\boldsymbol{x}_t, \boldsymbol{y}), \tag{11}$$

where $\hat{\boldsymbol{x}}_0(\boldsymbol{x}_t)$ is the unconditional Tweedie estimate. Equation (11) highlights that the $h$-transform can also be interpreted as a correction factor to the unconditional denoised estimate, similar to [52, 80].

## 3.2 Connections to variational inference and stochastic control

A limitation to the fine-tuning objective with DEFT is that it requires a small dataset of paired datapoints and measurements. In this section, we propose an alternative approach by expressing the solution to the conditional sampling problem as a stochastic optimal control objective, which is highlighted in Theorem 3.1 3). This allows us to learn the $h$-transform by optimising a variational inference-type problem. Importantly, this stochastic control objective only requires the availability of a single noisy observation $\boldsymbol{y}$ instead of a paired fine-tuning dataset. Further, the stochastic control objective can even be used in other conditional sampling tasks, for example in reward tilted distributions, i.e. where the goal is to sample from $\pi(\boldsymbol{x}) \propto e^{r(\boldsymbol{x})} p_{\mathrm{data}}(\boldsymbol{x})$. Here $e^{r(\boldsymbol{x})}$ serves the same purpose as the likelihood, but there is no explicit measurement $\boldsymbol{y}$ [18].

However, the stochastic control objective is not directly applicable for high-dimensional training, as the complete chain $\{\boldsymbol{H}_t\}_t$ must be kept in memory and backpropagated through or adjoint methods have to be used [36]. In Appendix G.3, we discuss several alternatives and present experiments for scaling up the above objective, e.g., methods like VarGrad [53] and Trajectory Balance [42]. VarGrad allows to detach the trajectory from the gradient computation, drastically reducing the memory footprint. We discuss concurrent work in G.1 and G.2. Further, we show initial experiments for conditional sampling in G.4. The stochastic control objective serves as a conceptual bridge between sampling from unnormalised densities using diffusion models [74, 75, 81] and conditional score-based generative modelling.

### 3.3 Likelihood-informed inductive bias

If the likelihood is differentiable, we can impose an inductive bias on the $h$-transform approximation. Specifically, the generalized $h$-transform can be expressed as an expectation, and we can apply the DPS approximation [12] as follows

$$\nabla_{\boldsymbol{x}_t} \ln p_{y|t}(\boldsymbol{y}|\boldsymbol{x}_t) = \nabla_{\boldsymbol{x}_t} \ln \mathbb{E}_{\boldsymbol{x}_0 \sim p(\boldsymbol{x}_0|\boldsymbol{x}_t)}[p(\boldsymbol{y}|\boldsymbol{x}_0)] \approx \nabla_{\boldsymbol{x}_t} \ln p(\boldsymbol{y}|\mathbb{E}[\hat{\boldsymbol{x}}_0|\boldsymbol{x}_t])$$
$$\approx \nabla_{\boldsymbol{x}_t} \ln p(\boldsymbol{y}|\hat{\boldsymbol{x}}_0(\boldsymbol{x}_t)),$$

where we use Tweedie's estimate based on the pre-trained unconditional diffusion model in the last step. The DPS approximation has been validated in many different conditional sampling tasks, so it would make for a good initialisation of the learned $h$-transform. However, the DPS approximation requires the Jacobian of the unconditional model, which is expensive to compute and known to be poorly conditioned. Further, in applications where we only have access to the forward pass of the unconditional model, the Jacobian is infeasible to compute. Similar to [50], we found that omitting this term still leads to an expressive architecture, while greatly reducing the computational cost. Thus, we propose the following network architecture

$$h_t^\phi(\boldsymbol{x}_t, \boldsymbol{y}) = \text{NN}_1^\phi(\boldsymbol{x}_t, \hat{\boldsymbol{x}}_0(\boldsymbol{x}_t), \nabla_{\hat{\boldsymbol{x}}_0} \ln p(\boldsymbol{y}|\hat{\boldsymbol{x}}_0(\boldsymbol{x}_t)), t) + \text{NN}_2^\phi(t) \nabla_{\hat{\boldsymbol{x}}_0} \ln p(\boldsymbol{y}|\hat{\boldsymbol{x}}_0(\boldsymbol{x}_t)), \quad (12)$$

to parametrise the $h$-transform, where the last layer of $\text{NN}_1^\phi$ is initialised with $\mathbf{0}$ and $\text{NN}_2^\phi$ is initialised to output $1$. This initialisation provides a computationally efficient approximation to the $h$-transform, which still guides the sampling.

This type of network architecture has been proposed within the sampling community to apply diffusion models to normalising constant estimation [49, 74, 81]. The theoretical connection to stochastic control in Section 3.2, motivates us further to adapt the architectures from the sampling field to the conditional generative modelling setting. We ablate the different components of our proposed architecture in Appendix F.1 and find that the additional components greatly improve performance empirically.

## 4 Experiments

We evaluate the DEFT framework from Section 3.1 on both linear and non-linear natural and medical image reconstruction tasks, as well as the motif scaffolding problem in protein design. Further, in Appendix H.2 we provide a comparison of the conditional training framework with DEFT on the FLOWERS [47] image dataset. We provide our code `https://github.com/alexdenker/DEFT`.

### 4.1 Image reconstruction

We test a wide variety of both linear and non-linear image reconstruction tasks on the $256 \times 256\text{px}$ ImageNet dataset [58]. We make use of a pre-trained unconditional diffusion model with $\sim 500\text{M}$ parameters [16][2]. We perform all our evaluations on a $1k$ subset of the validation set[3]. For all inverse problems under consideration, the $h$-transform was trained on a separate 1k subset of the validation set. For linear inverse problems, we compare against $\Pi$GDM [65], DDRM [32], DPS [12] and RED-diff [44]. Additionally, we evaluate $\text{I}^2\text{SB}$ [39]. The performance of $\text{I}^2\text{SB}$ can be seen as an upper-bound to DEFT, as it is a conditional diffusion trained on the complete ImageNet dataset. For non-linear tasks, we only compare against DPS and RED-diff as both $\Pi$GDM and DDRM are not directly applicable to non-linear forward operators. For DEFT we make use of the DDIM sampling scheme with 100 time steps [64]. For the comparison methods we used the same hyperparameters as in [44] without further tuning, including the number of sampling steps (1000 for DPS and RED-Diff, 20 for DDRM and 100 for $\Pi$GDM).

We compute PSNR and SSIM, which are commonly used distortion measures, along with perceptual metrics such as Learned Perceptual Image Patch Similarity (LPIPS) [83], Kernel Inception Distance (KID) [8], and top-1 classifier accuracy of a pre-trained ResNet50 model [26]. There is a well-known tradeoff between optimising distortion metrics versus perceptual quality, and depending on the task, one may wish for better performance along one axis at the cost of the other. For natural image tasks

---

[2]Checkpoints are available at `https://github.com/openai/guided-diffusion`
[3]`https://bit.ly/eval-pix2pix`

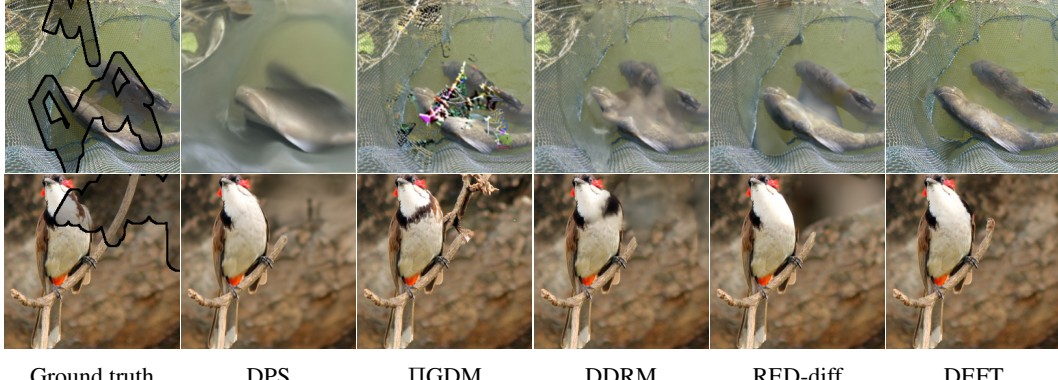

| Ground truth | DPS | ΠGDM | DDRM | RED-diff | DEFT |

Figure 2: Results for inpainting. We show the ground truth with the inpainting mask superimposed.

Table 1: Results on inpainting and 4x super-resolution. Best values are shown in **bold**, second best values are underlined. We report both the total time to sample 1k images, and the time per sample in seconds. The time to sample includes the training time for DEFT. These tasks aim to generate "natural"-looking images and therefore perceptual similarity metrics (KID, LPIPS and top-1) are more relevant. $I^2SB$ (grey column) can be considered an upper bound on performance.

| | Inpainting | | | | | | Super-resolution | | | | | |
| | DPS | ΠGDM | DDRM | RED-diff | DEFT | $I^2$SB | DPS | ΠGDM | DDRM | RED-diff | DEFT | $I^2$SB |
|---|---|---|---|---|---|---|---|---|---|---|---|---|
| PSNR (↑) | 21.27 | 20.30 | 20.72 | **23.29** | 22.18 | 23.26 | 24.83 | 25.25 | 25.32 | **25.95** | 24.92 | 23.95 |
| SSIM (↑) | 0.67 | 0.82 | 0.83 | **0.87** | 0.85 | 0.86 | 0.71 | 0.73 | 0.72 | **0.75** | 0.71 | 0.64 |
| KID (↓) | 15.28 | 4.50 | 2.50 | 0.86 | **0.29** | 0.238 | 10.01 | 10.9 | 14.0 | 10.0 | **1.78** | 0.004 |
| LPIPS (↓) | 0.26 | 0.12 | 0.14 | 0.10 | **0.09** | 0.068 | 0.16 | 0.15 | 0.23 | 0.25 | **0.12** | 0.11 |
| top-1 (↑) | 58.2 | 67.8 | 68.6 | **72.0** | 71.7 | 74.5 | 71.5 | 71.02 | 63.9 | 66.7 | **71.9** | 71.6 |
| Time (hrs) (↓) | 30.72 | 2.83 | **0.33** | 7.86 | 5.2 | N/A | 30.72 | 2.83 | **0.33** | 7.86 | 5.2 | N/A |
| Time per sample (s) (↓) | 100.6 | 10.2 | 1.22 | 28.3 | 4.36 | N/A | 100.6 | 10.2 | 1.22 | 28.3 | 4.36 | N/A |

involving in-painting and super-resolution, it is common to prefer "natural"-looking images, which score better on perceptual similarity, whereas for tasks involving (medical) image reconstruction it is standard to prefer a lower distortion metric [9]. Further, we calculate the total time (including training for DEFT) for evaluation 1k validation images in the "Time (hrs)" row. Furthermore, we report the effective time taken to sample a single image in the "Time per sample (s)" row. This time is calculated by fitting the largest batch size of validation images that fit on a single A100 GPU and dividing the time taken for the batch by the batch size.

**Inpainting** First, we evaluate DEFT on the linear inverse problem of image inpainting. We make use of the inpainting masks for the 1k subset used by [59][2], which includes masks that obscure $20\% - 30\%$ of the image. Results are shown in Table 1, including the computational time for sampling all 1000 validation images. For DEFT, this computational time additionally includes the 3.9 hrs of training time of the $h$-transform additionally with the 1.2 hrs of evaluation. Even with the added training time, we reduce the overall computational time for DEFT, compared to DPS and RED-diff. A visual comparison is provided in Figure 2. Further, in Figure 8 in the Appendix, we show the diversity of samples using different initial seeds. Even though ΠGDM and DDRM are faster methods, they perform significantly worse, and are only applicable for linear inverse problems. Inpainting is a task that prefers "natural"-looking image samples, and DEFT outperforms all other methods on perceptual metrics such as LPIPS and KID, being a close second on top-1 accuracy.

**Super-resolution** For another linear inverse problem, we evaluate 4x noiseless super-resolution. Here, the forward operator is given by a bicubic downsampling. The results are presented in Table 1. While DEFT has a lower PSNR compared to the baseline methods, we see significant improvement on perceptual quality metrics (KID, LPIPS, and top-1 accuracy). We show visual results in Figure 9.

**High dynamic range** For the first non-linear tasks, we make use of the high dynamic range (HDR) task described in [44]. Here, the forward operator is given by $\mathcal{A}(\boldsymbol{x}) = \text{clip}(2\boldsymbol{x}; -1, 1)$, where $\boldsymbol{x}$ denotes the RGB image scaled to the range $[-1, 1]$. The results are presented in Table 2. We observe

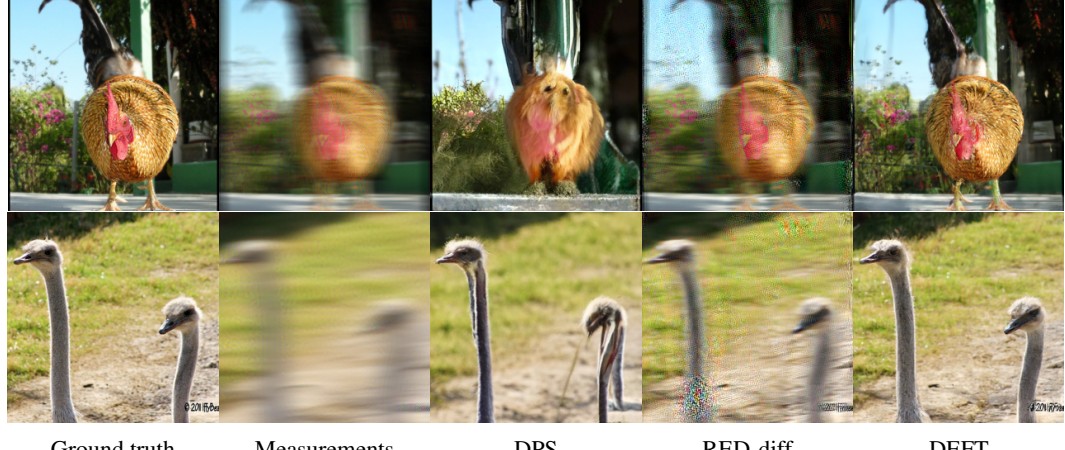

| Ground truth | Measurements | DPS | RED-diff | DEFT |

Figure 3: Results for non-linear deblurring. We show both the ground truth, the measurements and samples for DPS, RED-diff and DEFT. DEFT is able to reconstruct high-quality images.

Table 2: Results on different non-linear image reconstruction tasks. Best values are shown in **bold**, second best values are underlined.

| | **HDR** | | | **Phase retrieval** | | | **Non-linear Deblurring** | | |
|---|---|---|---|---|---|---|---|---|---|
| | DPS | RED-diff | DEFT | DPS | RED-diff | DEFT | DPS | RED-diff | DEFT |
| PSNR ($\uparrow$) | 7.94 | 25.23 | **28.51** | 9.99 | 10.53 | **13.03** | 17.57 | 21.21 | **25.16** |
| SSIM ($\uparrow$) | 0.21 | 0.79 | **0.89** | 0.12 | 0.17 | **0.32** | 0.39 | 0.53 | **0.79** |
| KID ($\downarrow$) | 272.5 | 1.2 | **0.10** | 93.2 | 114.0 | **80.89** | 12.89 | 66.8 | **0.34** |
| LPIPS ($\downarrow$) | 0.72 | 0.1 | **0.04** | 0.66 | 0.6 | **0.52** | 0.42 | 0.42 | **0.09** |
| top-1 ($\uparrow$) | 4.0 | 68.5 | **74.0** | 1.5 | 7.2 | **13.1** | 30.2 | 23.5 | **69.9** |
| Time (hrs) ($\downarrow$) | 30.7 | 7.9 | **5.2** | 30.7 | 7.9 | **5.2** | 30.9 | 8.1 | **5.2** |
| Time per sample (s) ($\downarrow$) | 100.4 | 28.3 | **4.4** | 100.6 | 28.4 | **4.4** | 101.2 | 30.4 | **4.6** |

that DPS struggles with this specific non-linear tasks, while DEFT achieves good results. We show a visual comparison in the Appendix, see Figure 10.

**Phase retrieval** The goal in phase retrieval is to recover the image from intensity measurements only, i.e., the forward operator is given by $\mathcal{A}(\boldsymbol{x}) = |\mathcal{F}\boldsymbol{x}|$, with $\mathcal{F}$ as the Fourier transform. We study the same 2x oversampling setting as in [12, 44]. Phase retrieval is a challenging non-linear inverse problem, as the forward operator is invariant to translations, global phase shifts and complex conjugation. In Figure 7 in the appendix, we show samples for different initial seeds and observe a wide variety of image quality. We also observe this behaviour for the baseline methods. However, DEFT is able to achieve better performance compared to RED-diff and DPS, see also Table 2. However, there is room for further improvement to achieve good reconstructions on a consistent basis.

**Non-linear deblurring** The non-linear deblurring task was originally proposed by [12]. Here, the forward operator is defined by a trained neural network [70], resulting in a highly non-linear blur. Quantitative results are presented in Table 2. This non-linear reconstruction task was also evaluated for RED-diff in [44]. However, we found that the forward operator of the original implementation[4] leads to a nearly trivial reconstruction task. In Appendix F.2, we show results with the code from [44], while Table 2 shows the results with our implementation of the forward operator. Further, in Figure 3 we provide a visual comparison, where DEFT is able to recover the ground truth quite faithfully.

**Ablation: Size of fine-tuning dataset** As DEFT requires a dataset for fine-tuning, we ablate the number of training samples. We trained DEFT on a subset of 10, 100 and 200 ImageNet images for Inpainting. We see improvements of all metrics, when training on a larger dataset. The results are presented in Table 3. For the KID, we outperform RED-diff (KID: 0.86) even when trained on only 200 images. However, even with 10 images, we perform quite competitively, showcasing that our method is very sample-efficient when it comes to learning a conditional transform.

---

[4]https://github.com/NVlabs/RED-diff/tree/master

Table 3: Varying the size of the fine-tuning dataset for DEFT for Inpainting on ImageNet.

| DEFT on ImageNet for Inpainting | | | | |
|---|---|---|---|---|
| Number of images | 10 | 100 | 200 | 1000 |
| PSNR ($\uparrow$) | 20.87 | 20.99 | 22.11 | 22.18 |
| SSIM ($\uparrow$) | 0.83 | 0.84 | 0.847 | 0.85 |
| KID ($\downarrow$) | 1.85 | 0.978 | 0.401 | 0.29 |
| LPIPS ($\downarrow$) | 0.123 | 0.112 | 0.096 | 0.09 |
| top-1 ($\uparrow$) | 68.8 | 69.6 | 70.6 | 71.7 |

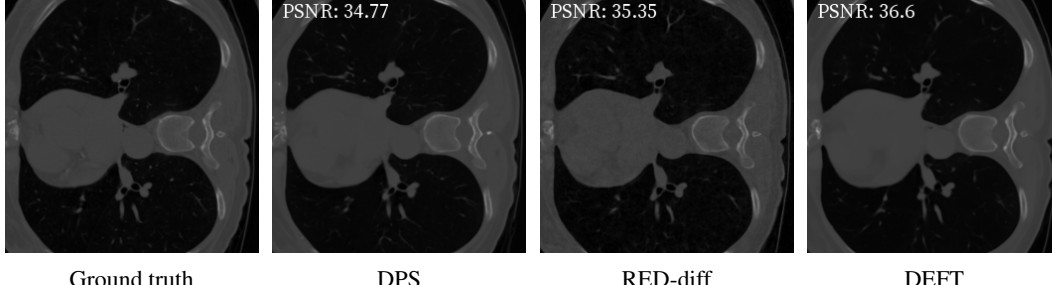

| Ground truth | DPS | RED-diff | DEFT |

Figure 4: Reconstructions for computed tomography on LoDoPab-CT

## 4.2 Computed tomography

We are evaluating DEFT both on the 2016 American Association of Physicists in Medicine (AAPM) grand challenge dataset [45], and the LoDoPab-CT dataset [35], for details see Appendix E.1. For the unconditional models we make use of the attention U-Net architecture [16]. For the model trained on AAPM, we use exactly the same architecture ($\approx 374$ params.) as in [11], while for LoDoPab-CT we use a smaller model ($\approx 133$M params.). For the forward operator, we use a parallel-beam radon transform with 60 angles and add Gaussian noise with $\sigma_y = 1.0$, which corresponds to approx. 3.5% relative noise. We compare against DPS [12] and RED-diff [44], where the parameters were obtained using a grid search on a subset of the validation set to maximise the PSNR. In Table 4 we present PSNR and SSIM, in addition to the sampling time, and provide a visual comparison Figure 4. For both datasets, we choose the same DEFT architecture with about 23M parameters. In the Appendix F, we perform an ablation regarding the parametrisation of DEFT, see Table 6. In particular, these results show the necessity of providing the unconditional Tweedie estimate $\hat{x}_0$ as input to the $h$-transform in (12). We observe almost a 8dB difference in PSNR for models without the Tweedie estimate and the log-likelihood term.

Table 4: Results for CT on AAPM and LODOPAB-CT and sampling time per image on a single GeForce RTX 3090. Best values are shown in **bold**, second best values are underlined. For DEFT we use 100 DDIM steps, while RED-diff and DPS use 1000 time steps.

| | AAPM | | | LoDoPab-CT | | |
|---|---|---|---|---|---|---|
| | DPS | RED-diff | DEFT | DPS | RED-diff | DEFT |
| PSNR | 33.11 | **34.85** | 34.73 | 34.16 | 34.95 | **35.81** |
| SSIM | 0.885 | 0.865 | **0.887** | 0.846 | 0.849 | **0.876** |
| Time (s) | 208.9 | 83.4 | **16.3** | 156.8 | 70.1 | **13.8** |

## 4.3 Conditional protein design: motif scaffolding

We evaluate DEFT on the contiguous motifs of the RFDiffusion benchmark [77]. In this motif scaffolding task, we sample protein $C_\alpha$ atom coordinates $x \in \mathbb{R}^d$ such that the generated backbone contains a targeted motif, i.e. a subset of $C_\alpha$ coordinates $y \in \mathbb{R}^n$, similar to an image outpainting task. The forward operator is therefore given by $y = \mathcal{A}(x) = Ax$, where $A \in \{0, 1\}^{n \times d}$ denotes a masking matrix which only selects the $n$ observed $C_\alpha$ coordinates.

We leverage the pretrained Genie diffusion model which is an unconditional model for protein backbone generation [37]. To apply DEFT to it, we use a downsized version of the unconditional base model as our $h$-transform model which only uses 200k instead of the original 4.1M parameters. To adopt this model to the DEFT algorithm, we modify the SE(3)-invariant encoder by adding additional conditional pair feature networks for the motif coordinates as well as the unconditional Tweedie estimate $\hat{x}_0$, similar to the setting in the previous experiments. As per Section 3.3, we add a time-dependent likelihood approximation term to the $h$-transform network. We train the $h$-transform

network on the same SCOPe dataset as in [17]. More details on the training details can be found in App. E.2. We compare DEFT against DPS [12] and a previously published version of Genie that was trained in an amortised fashion [17]. The guidance parameter of DPS was tuned over 5 different experiment runs. The amortised model serves here as an upper limit of how well DEFT can perform with Genie as a base model.

The overall in-silico success, defined by scRMSD $< 2$Å and motifRMSD $< 1$Å, is provided in Figure 5. In the Appendix, we provide a detailed breakdown of these results, see Figure 13. Further, in Figure 14 and Figure 15 we provide a comparison of the task 1YCR for the different methods. We observe that DEFT outperforms DPS, solving 10 out of the 12 tasks compared to only 5 tasks for DPS. While it has lower success rates than the amortised model, it still solves all but two tasks in that benchmark with only 9% of the parameter count and significantly shorter training time compared to the amortised model (800 epochs for DEFT vs 2100 epochs for amortised). The low performance of DPS indicates that the base Genie model is limiting the performance here and may partly explain the performance difference between DEFT and amortised training. Exploring DEFT with a more capable base model is therefore another promising avenue for research. Excitingly, the lower training time and data requirements of DEFT enable fine-tuning a model on specific protein families for particular applications, a task that is left for future work.

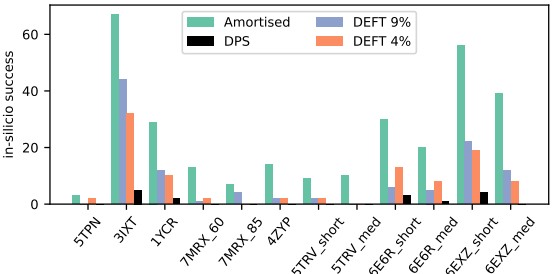

Figure 5: Comparison of DPS, DEFT and amortised training for motif scaffolding for 12 contiguous targets. 4% and 9% are the relative sizes of the h-transform compared to the unconditional model.

| METRIC | DPS | DEFT | AMORTISED |
|---|---|---|---|
| % Success ($\uparrow$) | 1.3 | 9.2 | 24.5 |
| % scRMSD $< 2$ Å($\uparrow$) | 44.3 | 28.9 | 42. |
| % mRMSD $< 1$ Å($\uparrow$) | 4.1 | 24.0 | 45.8 |

Table 5: RFDIFF benchmark metrics (averaged over the 11 targets, 100 samples each). Success: scRMSD $< 2$Å, motifRMSD $< 1$Å. Details in Sec. 4.3.

## 5 Conclusion

We presented a unified mathematical framework, based on Doob's $h$-transform, to better understand and classify different conditional diffusion methods. Under this framework, we proposed DEFT, a novel parameter-efficient conditional fine-tuning method that does not require backpropagation through large pre-trained score networks, resulting in efficient sampling. We evaluated DEFT on several image reconstruction tasks and showed that it reliably outperformed standard methods, both in time, reconstruction quality and perceptual similarity metrics. While DEFT requires additional training on a small dataset of paired measurements, we find that it is still faster than many existing baselines due to being able to use fewer sampling steps during evaluation, and not needing to backpropagate during evaluation.

**Limitations and future work** The DEFT framework uses a (small) fine-tuning dataset, in contrast to zero-shot conditional sampling approaches, e.g., DPS [12] or ΠGDM [65]. Fine-tuning on small datasets may have the risk of overfitting to biases inherent in the data. In contrast to zero-shot conditional sampling, DEFT assumes no knowledge of the forward operator. However, the forward operator can be incorporated as an inductive bias within the network architecture to improve performance. We also proposed a zero-shot approach through the optimal control loss in Section 3.2, which only needs a single observation $\boldsymbol{y}$ to learn the $h$-transform. Though we show promising results scaling this approach to the MNIST dataset in Appendix H, the computational burden of simulating the full SDE at each iteration is still high, which might make this optimal control loss infeasible for high-dimensional data. However, there is promising recent work on partial trajectory optimisation [79], which may reduce the computational burden of the stochastic control objective, making it competitive with existing methods.

## Acknowledgements

Alexander Denker acknowledges support by the EPSRC programme grant EP/V026259/1. Shreyas Padhy is funded by the University of Cambridge Harding Distinguished Postgraduate Scholars Programme.

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

# A   Background on diffusion formulations

## A.1   Recap - continuous and discrete diffusion formulations

The discretised DDPM versions with various discrete time schedules amount to the time-dependent OU process

$$\mathrm{d}\mathbf{X}_t = -\frac{\beta(t)}{2}\boldsymbol{X}_t\mathrm{d}t + \sqrt{\beta(t)}\,\overrightarrow{\mathrm{d}\mathbf{W}}_t, \tag{13}$$

where choosing different time schedules amounts to choosing different functions $\beta(t)$. This process gives rise to the following transition probabilities

$$p(\boldsymbol{x}, t|\boldsymbol{x}_0, 0) = \vec{p}_{t|0}(\boldsymbol{x}|\boldsymbol{x}_0) \tag{14}$$

$$= \mathcal{N}\left(\boldsymbol{x}_0 e^{-\int_0^T \frac{\beta(s)}{2}\mathrm{d}s}, \boldsymbol{I}\int_0^T \beta(t)e^{-\int_0^{T-t}\beta(s)\mathrm{d}s}\mathrm{d}t\right) \tag{15}$$

$$= \mathcal{N}\left(\boldsymbol{x}_0 e^{-\int_0^T \frac{\beta(s)}{2}\mathrm{d}s}, \left(1 - e^{-\int_0^T \beta(s)\mathrm{d}s}\right)\boldsymbol{I}\right), \tag{16}$$

see also Appendix B in [67]. With $\bar{\alpha}(t) = e^{-\int_0^T \beta(s)\mathrm{d}s}$, this is the familiar form ([28]):

$$p(\boldsymbol{x}, t|\boldsymbol{x}_0, 0) = \vec{p}_{t|0}(\boldsymbol{x}|\boldsymbol{x}_0) = \mathcal{N}\left(\boldsymbol{x}_0\sqrt{\bar{\alpha}(t)}, (1 - \bar{\alpha}(t))\boldsymbol{I}\right), \tag{17}$$

with $\bar{\alpha}(t)$ time-dependent and we can therefore choose different functional forms for the noise schedule by either choosing the transition parameters $\beta(t)$ or the cumulative parameters $\alpha(t)$.

If we define the noise schedule in terms of $\beta(t)$, the time-dependent OU process is immediately apparent (see (13)). If we define the noise schedule in terms of $\bar{\alpha}(t)$, the mean and variance of the corresponding OU process can simply be obtained from

$$\beta(t) = -\frac{\mathrm{d}}{\mathrm{d}t}\left[\ln\bar{\alpha}(t)\right]. \tag{18}$$

## A.2   Score, noise and mean diffusion formulations

The score-based model used for generation at inference time can be parametrised to model different quantities. The three most common one are the score, the noise and the mean. Using the score based SDE formulation we parametrise the network as the score that is the network approximates the quantity:

$$\nabla_{\boldsymbol{x}}\ln p_t(\boldsymbol{x}_t) \approx s_t^\theta(\boldsymbol{x}_t) \tag{19}$$

Moving on to the DDPM formulation one typically trains a noise prediction network instead which is proportional to the score

$$s_t^\theta(\boldsymbol{x}_t) = -\frac{1}{\sqrt{1 - \bar{\alpha}(t)}}\epsilon_t^\theta(\boldsymbol{x}_t). \tag{20}$$

This formulation is typically preferable for training as it is known to learn a less stiff vector field [82].

Finally in its most naive form DDPM also admits a mean matching formulation

$$\mu_0^\theta(\boldsymbol{x}_t, t) = \frac{1}{\sqrt{\alpha_t}}\left(\boldsymbol{x}_t - \frac{1 - \alpha_t}{\sqrt{1 - \bar{\alpha}(t)}}\epsilon_t^\theta(\boldsymbol{x}_t)\right), \tag{21}$$

whilst not the ideal parametrisation for direct training, it is a useful expression/macro, for expressing the sampling updates more succinctly.

In this work we parametrise $\epsilon_t^\theta$ directly with our novel architecture for finetuning, using a noise matching objective as in DDPM, however we allude to and use the above parametrisations across our propositions and novel architectures. See Algorithm 1 for training and Algorithm 2 for sampling.

# B   Related Work Discussion

A common distinction to all the works we will discuss in this section is that they all either train the conditional network from scratch or they initialise the conditional network with a pretrained score and fully train a conditional network. This is in stark contrast to DEFT which completely freezes the unconditional score and trains a highly efficient network to learn the $h$-transform which ranges from $4-17\%$ in total parameter size of the pretrained unconditional score network.

**Classifier free guidance**   Methodologies such as classifier free guidance [27] do not model the forward operator explicitly. As a result, if these methods are applied to settings such as motif-scaffolding or image out-painting (where the conditioning is on a subset of the random variable), these methodologies would only denoise the scaffolding and the missing image patches. This is different to our approach which adds noise to both motif and scaffolding and then proceeds to denoise both jointly as part of the same space. In a way, one can view RFDiffusion's conditional training as an application of classifier-free guidance to this subset conditioning setting.

**Image 2 Image Schrödinger Bridges (I2SB [39])**   I2SB and more generally aligned Schrödinger Bridges [63] are a recently proposed class of conditional generative models based on ideas from Schrödinger bridges. The premise of these methods is that they aim to learn an interpolating diffusion between a clean data sample and a altered or corrupted data sample. This is in contrast to our framework: we consider an unconditioned SDE and condition it to hit an event at a particular time, thus learning an interpolating distribution between noise and an un-corrupted target distribution. This results in several algorithmic differences:

- At its core, I2SB treat $\boldsymbol{Y} = \mathcal{A}(\boldsymbol{X}_0) + \eta$ and $\boldsymbol{X}_0$ as source and target distributions respectively; thus, at sampling time, $\boldsymbol{Y}$ is provided to the learned SDE which generates approximate samples from $\mathrm{Law}\,(\boldsymbol{X}_0)$. However, in our approach, the source distribution is $\mathcal{N}(0, \mathbf{I})$ and we pass $\boldsymbol{Y}$ to the score network to then obtain approximate samples from $\mathrm{Law}\,(\boldsymbol{X}_0)$.

- The score network in I2SB is a function only of $\boldsymbol{X}_t$ and not $\boldsymbol{Y} = \mathcal{A}(\boldsymbol{X}_0) + \eta$. This means that in I2SB, the network is parametrised as $\epsilon_t^\theta(\boldsymbol{X}_t)$, whilst in our setting we parametrise as $\epsilon_t^\theta(\boldsymbol{X}_t, \mathcal{A}(\boldsymbol{X}_0) + \eta, \mathcal{A})$. In the case of completion tasks like motif-scaffolding or image out-painting, our paramerisation looks something like $\epsilon_t^\theta(\boldsymbol{X}_t, \boldsymbol{X}_0^{\mathrm{mask}}, \mathrm{mask})$. This makes the task much easier for the network as we effectively provide it with a binary variable indicating which parts of the image are conditioned and which are not. In I2SB, the network must learn this on its own. Furthermore, as we show in Prop. H.1, adding this to the network parametrisation is essential to allow recovering the true conditional score.

- The training procedure in I2SB uses the diffusion bridge $p(\boldsymbol{x}_t | \boldsymbol{x}_0, \boldsymbol{y})$ to add noise to both the source and target distributions, whilst our forward process is given by the transition density of an OU process $p(\boldsymbol{x}_t | \boldsymbol{x}_0)$ and is identical to standard DDPM/VP-SDE [28, 68] noise adding procedures.

- Finally and most importantly I2SB does full fine-tuning firstly initialising with a pretrained score and training all parameters of this large pretrained network to learn an unconditional network, this requires longer training times and significantly larger networks as they must be at least the same size as the unconditional score network.

To summarise: whilst both methodologies employ similar mathematical methodologies (e.g. Diffusion Bridges [14]), their ideations and resulting methods are fundamentally different: on one side, [39] learns an interpolating distribution between the unconditioned $p(\boldsymbol{x}_0)$ and conditioned $p(\boldsymbol{y} | \boldsymbol{x}_0)$ samples. On the other, we learn a denoising procedure that directly samples from the posterior $p(\boldsymbol{x}_0 | \boldsymbol{y})$; via this, we derive and explain most popular approaches for conditioning denoising diffusion models as part of our framework.

It's important to highlight that another akin approach to I2SB, also based on the h-transform but leveraging VP-SDEs, was recently proposed in [84].

**CDE**   [5] Similar to classifier free guidance. CDE [5] trains a conditional network from scratch without leveraging a pretrained unconditional score. For more details on CDE please see our detailed discussion in Appendix H.

**First Hitting Diffusions**   A line of generative modelling methods proposed in [40, 78] utilise the $h$-transform for unconditional generative modelling in the following settings:

- Hitting the target distribution $p_{\text{data}}$ in a finite amount of time $[0, T]$ via time reversing an h-transformed VP-SDE conditioned to hit $0$ at time $T$.
- Constraining a diffusion process at time $T$ to lie in a subset of the reals $\Omega \subseteq \mathbb{R}^d$.

Whilst the aforementioned work uses a similar methodology and theory the focus is more in line with unconditional generative modelling rather than our setting which seeks to sample from the posterior arising in an inverse problem / conditional generative modelling.

**RFDiffusion**   As highlighted in Alg. 3 and in contrast to AMORTISED TRAINING, RFDiffusion [77] does not noise the motif coordinates $\boldsymbol{X}_0^{[M]}$ with the forward OU-Process, instead it directly aims to sample from $p(\boldsymbol{X}_t^{[\backslash M]}|\boldsymbol{X}_0^{[M]})$ and estimate this score while keeping the motif fixed. We can relate this approach to our amortised learning of Doob's $h$-transform, by noting that RF diffusion can be understood as learning the marginal conditional score

$$p(\boldsymbol{x}_t^{[\backslash M]}|\boldsymbol{x}_0^{[M]}) = \int \overbrace{p(\boldsymbol{x}_t|\boldsymbol{x}_0^{[M]})}^{\propto h(t,\boldsymbol{x}_t)p_t(\boldsymbol{x}_t)} \ d\boldsymbol{x}_t^{[M]}. \tag{22}$$

This can be viewed as RFDiffusion estimating a marginal counterpart of our amortised $h$-transform approach. See Algs. 3 and 4 for more details on how these approaches differ in a pseudo-code implementation.

## C   Doob's $h$-transform

### C.1   Doob's $h$-transform intuition

Doob's transform provides a formal mechanism for conditioning a stochastic differential equation (SDE) to hit an event at a given time. The $h$-transform drift decomposes into two terms via Bayes rule, a conditional and a prior score

$$\nabla_{\boldsymbol{H}_t} \ln \overline{P}_{0|t}(\boldsymbol{X}_0 \in B \mid \boldsymbol{H}_t) = \nabla_{\boldsymbol{H}_t} \ln \overrightarrow{P}_{t|0}(\boldsymbol{H}_t \mid \boldsymbol{X}_0 \in B) - \nabla_{\boldsymbol{H}_t} \ln P_t(\boldsymbol{H}_t), \tag{23}$$

whereby the conditional score ensures that the event is hit at the specified boundary time, while the prior score ensures it is time-reversal of the correct forward process [14]. Doob's $h$-transform adds a new drift to the SDE which amounts to two terms (via Bayes Theorem), a conditional and an unconditional score

$$\nabla \ln \overline{P}_{0|t}(\boldsymbol{X}_0 \in B|\cdot) = \nabla \ln \overrightarrow{P}_{t|0}(\cdot|\boldsymbol{X}_0 \in B) - \nabla \ln P_t(\cdot). \tag{24}$$

Interestingly, these two terms provide for a unique intuition: the Doob's transform SDE is the time reversal of the forward SDE corresponding to (3), that is the time reversal of the forward SDE

$$\mathrm{d}\boldsymbol{X}_t = \vec{b}_t(\boldsymbol{X}_t)\,\mathrm{d}t + \sigma_t \overline{\mathrm{d}\mathbf{W}}_t, \quad \boldsymbol{X}_0 \sim \overrightarrow{P}_0(\cdot|\boldsymbol{X}_0 \in B), \tag{25}$$

coincides with the Doob transformed SDE (4) [14]. Thus we can view Doob's transform as the following series of steps:

1. Time reverse the SDE we want to condition ((4) to (25)).
2. Impose the condition via ancestral sampling from the conditioned distribution/posterior.
3. Time reverse once more to be in the same time direction as we started.

### C.2   Example: Truncated normal

Here for illustrative purposes we frame the problem of sampling from a truncated normal distribution as simulating an SDE that is given by Doob's h-transform.

Let's remind that a 1d truncated normal distribution had a density $p(x|a, b) \propto \mathbb{1}_{x \in (a,b)}(x)\mathcal{N}(x|\mu, \sigma^2)$. Now, let's assume a data distribution $p_0(x) = \mathcal{N}(\mu, \sigma^2)$ which is noised with an OU process (13). Thus we have that $p(x_0|x_t) = \mathcal{N}(x_0|\hat{\mu}_{0|t}(x_t), \hat{\sigma}_{0|t}(x_t)^2)$ is Gaussian, and so is

$p(x_t) = \mathcal{N}(x_t|\hat{\mu}_t, \hat{\sigma}_t^2)$. Let's add the constraint that the process hit at time $t = 0$ the event $\boldsymbol{X}_0 \in (a, b)$.

$$\mathrm{d}\boldsymbol{H}_t = \beta(t)\left(\frac{\boldsymbol{H}_t}{2} + \nabla_{\boldsymbol{H}_t}\ln\overrightarrow{P}_t(\boldsymbol{H}_t) - \nabla_{\boldsymbol{H}_t}\ln\overline{P}_{0|t}(\boldsymbol{X}_0 \in (a,b) \mid \boldsymbol{H}_t)\right)\mathrm{d}t + \sqrt{\beta(t)}\,\overline{\mathrm{d}\mathbf{W}}_t, \tag{26}$$

We have that the h-transform is given by

$$h(t, \boldsymbol{H}_t) = \overline{P}_{0|t}(\boldsymbol{X}_0 \in (a,b)|\boldsymbol{H}_t) = \int \mathbb{1}_{x\in(a,b)}(\boldsymbol{x}_0)\overline{p}_{0|t}(\boldsymbol{x}_0|\boldsymbol{H}_t)\mathrm{d}\boldsymbol{x}_0$$

$$= \int \mathbb{1}_{x\in(a,b)}(\boldsymbol{x}_0)\mathcal{N}(x|\hat{\mu}_{0|t}(\boldsymbol{H}_t), \hat{\sigma}_{0|t}(\boldsymbol{H}_t)^2)\mathrm{d}\boldsymbol{x}_0$$

$$= \frac{1}{\hat{\sigma}_{0|t}(\boldsymbol{H}_t)}\frac{\phi\left(\frac{\boldsymbol{H}_t - \hat{\mu}_{0|t}(\boldsymbol{H}_t)}{\hat{\sigma}_{0|t}(\boldsymbol{H}_t)}\right)}{\Phi\left(\frac{b - \hat{\mu}_{0|t}(\boldsymbol{H}_t)}{\hat{\sigma}_{0|t}(\boldsymbol{H}_t)}\right) - \Phi\left(\frac{a - \hat{\mu}_{0|t}(\boldsymbol{H}_t)}{\hat{\sigma}_{0|t}(\boldsymbol{H}_t)}\right)} \tag{27}$$

where $\phi(\xi) = \frac{1}{\sqrt{2\pi}}\exp\left(-\frac{1}{2}\xi^2\right)$ is the pdf of a standard normal distribution, $\Phi(\xi) = \frac{1}{2}\left(1 + \mathrm{erf}(\xi/\sqrt{2})\right)$ its cumulative function. The corrective drift term due to the h-transform can then be computed via autograd. The unconditional score term can be computed in closed form.

### C.3 Doob's $h$-transform classical noiseless setting

We now consider events of the form $\boldsymbol{X}_0 \in B$ which are described by an equality constraint $\mathcal{A}(\boldsymbol{X}_0) = \boldsymbol{y}$ with $\mathcal{A}$ a known *measurement* (or *forward*) operator and $\boldsymbol{y}$ an observation, which is a common setup in inverse problems such as inpainting or super-resolution.

**Corollary C.1.** *Consider the reverse SDE* (3)*, then it follows that*

$$\mathrm{d}\boldsymbol{H}_t = (\breve{b}_t(\boldsymbol{H}_t) - \sigma_t^2\nabla_{\boldsymbol{H}_t}\ln\overline{P}_{0|t}(\mathcal{A}(\boldsymbol{X}_0) = \boldsymbol{y} \mid \boldsymbol{H}_t))\,\mathrm{d}t + \sigma_t\overline{\mathrm{d}\mathbf{W}}_t, \tag{28}$$

*with* $\breve{b}_t(\boldsymbol{H}_t) = f_t(\boldsymbol{H}_t) - \sigma_t^2\nabla_{\boldsymbol{H}_t}\ln p_t(\boldsymbol{H}_t)$ *satisfies* $\mathrm{Law}\,(\boldsymbol{H}_s|\boldsymbol{H}_t) = \mathrm{Law}\,(\boldsymbol{X}_s|\boldsymbol{X}_t, \mathcal{A}(\boldsymbol{X}_0) = \boldsymbol{y})$ *thus* $\mathrm{Law}\,(\boldsymbol{H}_0) = \mathrm{Law}\,(\boldsymbol{X}_0|\mathcal{A}(\boldsymbol{X}_0) = \boldsymbol{y})$.

Sampling from (28) directly provides samples $\boldsymbol{x} \sim p_{\mathrm{data}}$ which also satisfy $\mathcal{A}(\boldsymbol{x}) = \boldsymbol{y}$. Crucially, this SDE is guaranteed to hit the conditioning in finite time, unlike prior equilibrium-motivated approaches [12, 19, 22, 24, 46, 65].

**Reconstruction guidance**  To get better insight into the challenge of sampling via Doob's $h$-transform in (28) let us re-express the $h$-transform as

$$\overline{P}_{0|t}(\mathcal{A}(\boldsymbol{X}_0) = \boldsymbol{y} \mid \boldsymbol{H}_t) = \int \mathbb{1}_{\mathcal{A}(\boldsymbol{x}_0)=\boldsymbol{y}}(\boldsymbol{x}_0)\overline{p}_{0|t}(\boldsymbol{x}_0|\boldsymbol{H}_t)\mathrm{d}\boldsymbol{x}_0, \tag{29}$$

where in the case of denoising diffusion models $\overline{p}_{0|t}(\boldsymbol{x}_0|\cdot)$ is the transition density of the reverse SDE (3). In practice, one does not have access to this transition density – i.e. we can sample from this distribution, but we cannot easily get its value at a certain point. This makes it difficult to approximate the integral. To alleviate this, a strand of recent works [12, 22, 57, 65] have proposed to apply a Gaussian approximation of $\overline{p}_{0|t}(\boldsymbol{x}_0|\cdot) \approx \mathcal{N}(\boldsymbol{x}_0 \mid \mathbb{E}[\boldsymbol{X}_0|\boldsymbol{X}_t = \cdot], \Gamma_t)$ leveraging Tweedie's formula and the pre-trained score network. This line of work is referred as *reconstruction guidance*. We note that whilst proposing to approximate the quantity $\overline{P}_{0|t}(\mathcal{A}(\boldsymbol{X}_0) = \boldsymbol{y}|\cdot)$, they do not make the connection to Doob's transform and thus are unable to provide guarantees on the conditional sampling that Cor. C.1 provides. Overall, the Gaussian-based approximations of Doob's $h$-transform lead to reconstruction guidance-based approaches

$$\mathrm{d}\boldsymbol{H}_t = \left(\breve{b}_t(\boldsymbol{H}_t) + \sigma_t^2\nabla_{\boldsymbol{H}_t}||\boldsymbol{y} - \mathrm{A}\mathbb{E}[\boldsymbol{X}_0|\boldsymbol{X}_t = \boldsymbol{H}_t]||_{\Gamma_t}^2\right)\mathrm{d}t + \sigma_t\overline{\mathrm{d}\mathbf{W}}_t, \boldsymbol{X}_T \sim \mathcal{P}_T, \tag{30}$$

where $\Gamma_t$ acts as a guidance scale [57, 62], and A is a matrix if $\mathcal{A}$ is linear otherwise $\mathrm{A} = \mathrm{d}\mathcal{A}(\mathbb{E}[\boldsymbol{X}_0|\boldsymbol{X}_t = \boldsymbol{H}_t])$.

# D  Proofs

## D.1  Proof of Proposition 2.2

*Proof.* Starting from the unconditioned reverse SDE

$$d\boldsymbol{X}_t = \bar{b}_t(\boldsymbol{X}_t)\, dt + \sigma_t\, \overline{d\mathbf{W}}_t, \quad \boldsymbol{X}_T \sim \mathbb{P}_T = Q_T^{f_t}[p(\boldsymbol{x}_0)], \tag{31}$$

we consider its reversal, the forward SDE, but we change its initial from distribution $p(\boldsymbol{x}_0)$ to the target posterior $p(\boldsymbol{x}_0|\boldsymbol{y})$, i.e.,

$$d\boldsymbol{H}_t = f_t(\boldsymbol{H}_t)\, dt + \sigma_t\, \overrightarrow{d\mathbf{W}}_t, \quad \boldsymbol{H}_0 \sim \frac{p(\boldsymbol{y}|\boldsymbol{x}_0)p_{\text{data}}(\boldsymbol{x}_0)}{p(\boldsymbol{y})}, \tag{32}$$

where $\bar{b}_t(\boldsymbol{H}_t) = f_t(\boldsymbol{H}_t) - \sigma_t^2 \nabla_{\boldsymbol{H}_t} \ln p_t(\boldsymbol{H}_t)$.

Now let us use $p_{t|y}(\boldsymbol{x}|\boldsymbol{y}) = \int p(\boldsymbol{x}_t|\boldsymbol{x}_0)\mathrm{d}p(\boldsymbol{x}_0|\boldsymbol{y})$ to denote the marginal of the above SDE and as before $p_t$ to denote the marginal of the reference starting at the data distribution. Then it follows that

$$p_{t|y}(\boldsymbol{x}|\boldsymbol{y}) = p_t(\boldsymbol{x})p(\boldsymbol{y})^{-1} \int \frac{\bar{p}_{t|0}(\boldsymbol{x}|\boldsymbol{x}_0)}{p_t(\boldsymbol{x})} p(\boldsymbol{y}|\boldsymbol{x}_0)p_{\text{data}}(\boldsymbol{x}_0)d\boldsymbol{x}_0 \tag{33}$$

$$= p_t(\boldsymbol{x})p(\boldsymbol{y})^{-1} \int \bar{p}_{0|t}(\boldsymbol{x}_0|\boldsymbol{x})p(\boldsymbol{y}|\boldsymbol{x}_0)d\boldsymbol{x}_0 \tag{34}$$

and thus the score of the reference starting at the posterior is given by:

$$\nabla_{\boldsymbol{x}} \ln p_{t|y}(\boldsymbol{x}|\boldsymbol{y}) = \nabla_{\boldsymbol{x}} \ln p_t(\boldsymbol{x}) + \nabla_{\boldsymbol{x}} \ln \int \bar{p}_{0|t}(\boldsymbol{x}_0|\boldsymbol{x})p(\boldsymbol{y}|\boldsymbol{x}_0)d\boldsymbol{x}_0 \tag{35}$$

Now that we have the score of the SDE in Equation 32 we can reverse it yet another time to obtain the conditional backwards SDE:

$$\boldsymbol{H}_T \sim Q_T^{f_t}[p(\boldsymbol{x}_0|\boldsymbol{y})] = \int \bar{p}_{T|0}(\boldsymbol{x}|\boldsymbol{x}_0)p(\boldsymbol{x}_0|\boldsymbol{y})\mathrm{d}\boldsymbol{x}_0$$

$$d\boldsymbol{H}_t = \left(f_t(\boldsymbol{H}_t) - \sigma_t^2\left(\nabla_{\boldsymbol{H}_t} \ln p_t(\boldsymbol{H}_t) + \nabla_{\boldsymbol{H}_t} \ln p_{y|t}(\boldsymbol{Y} = \boldsymbol{y}|\boldsymbol{H}_t)\right)\right)\, dt + \sigma_t\, \overline{d\mathbf{W}}_t,$$

$$d\boldsymbol{H}_t = \left(\bar{b}_t(\boldsymbol{H}_t) - \sigma_t^2 \nabla_{\boldsymbol{H}_t} \ln p_{y|t}(\boldsymbol{Y} = \boldsymbol{y}|\boldsymbol{H}_t)\right)\, dt + \sigma_t\, \overline{d\mathbf{W}}_t, \tag{36}$$

Where it is very important to notice that the backward SDE starts a the terminal distribution of the conditional forward SDE $Q_T^{f_t}[p(\boldsymbol{x}_0|\boldsymbol{y})]$ and not $\mathbb{P}_T = Q_T^{f_t}[p(\boldsymbol{x}_0)]$, for a VP-SDE this happens to approximately be $\mathcal{N}(0, I)$ in both cases (i.e. $Q_T^{f_t}[p(\boldsymbol{x}_0|\boldsymbol{y})] \approx Q_T^{f_t}[p(\boldsymbol{x}_0)] \approx \mathcal{N}(0, I)$) however more generally it is not and one needs to be careful.

Note that this remark highlights that the score used in DPS [12] (i.e. $\nabla_{\boldsymbol{x}} \ln p_{t|y}(\boldsymbol{x}|\boldsymbol{y})$) is in fact the score of an OU process starting at $p(\boldsymbol{x}_0|\boldsymbol{y})$ notice the cancellation going from Equations (33) to (34) was only possible since the prior in our target posterior is the initial distribution for the forward SDE (in their case an OU-process) with marginal $p_t$, these considerations are subtle yet important and omitted in prior works. Whilst this is akin the relationship motivated in DPS as $\nabla_{\boldsymbol{x}} \ln p_{t|y}(\boldsymbol{x}|\boldsymbol{y}) = \nabla_{\boldsymbol{x}} \ln p_t(\boldsymbol{x}) + \nabla_{\boldsymbol{x}} \ln p_t(\boldsymbol{y}|\boldsymbol{x})$, DPS fails to convey that this is in fact the score of a VP-SDE with the posterior $p(\boldsymbol{x}_0|\boldsymbol{y})$ as its initial distribution.

$\square$

## D.2  Proof of Theorem 3.1 2)

*Proof.* The idea of the proof is similar to Proposition H.1. Here, we directly proof the amortised variation from Equation (9). First, we state the well known score matching identity for the posterior score:

$$\nabla_{\boldsymbol{x}_t} \ln p_t(\boldsymbol{x}|\boldsymbol{y}) = \int \nabla_{\boldsymbol{x}_t} \ln \bar{p}_{t|0}(\boldsymbol{x}_t|\boldsymbol{x}_0)\bar{p}_{0|t}(\boldsymbol{x}_0|\boldsymbol{x}_t, \boldsymbol{y})\mathrm{d}\boldsymbol{x}_0.$$

Via the mean squared error property of the conditional expectation we get the minimiser as

$$\mathbb{E}\left[\nabla_{\boldsymbol{H}_t} \ln \bar{p}_{t|0}(\boldsymbol{H}_t|\boldsymbol{X}_0) - \nabla_{\boldsymbol{H}_t} \ln p_t(\boldsymbol{H}_t)\Big| \boldsymbol{Y} = \boldsymbol{y}, \boldsymbol{H}_t = \boldsymbol{x}\right]$$

Then

$$h_t^*(\boldsymbol{x}, \boldsymbol{y}) = \int (\nabla_{\boldsymbol{x}} \ln \vec{p}_{t|0}(\boldsymbol{x}|\boldsymbol{x}_0) - \nabla_{\boldsymbol{x}} \ln p_t(\boldsymbol{x})) \bar{p}_{0|t}(\boldsymbol{x}_0|\boldsymbol{H}_t = \boldsymbol{x}, \boldsymbol{Y} = \boldsymbol{y}) \mathrm{d}\boldsymbol{x}_0$$

$$= \int \nabla_{\boldsymbol{x}} \ln \vec{p}_{t|0}(\boldsymbol{x}|\boldsymbol{x}_0) \bar{p}_{0|t}(\boldsymbol{x}_0|\boldsymbol{H}_t = \boldsymbol{x}, \boldsymbol{Y} = \boldsymbol{y}) \mathrm{d}\boldsymbol{x}_0 - \nabla_{\boldsymbol{x}} \ln p_t(\boldsymbol{x})$$

$$= \nabla_{\boldsymbol{x}} \ln p_t(\boldsymbol{x}|\boldsymbol{y}) - \nabla_{\boldsymbol{x}} \ln p_t(\boldsymbol{x}) = \nabla_{\boldsymbol{x}} \ln p_t(\boldsymbol{y}|\boldsymbol{x}).$$

□

# E  Experimental details

## E.1  Image Experiments

In the image experiment, we use the DDPM [28] formulation for the diffusion model with $N = 1000$ steps, a linear $\beta$-schedule with $\beta_0 = 10^{-4}$ and $\beta_N = 2 \cdot 10^{-2}$. In all our imaging experiments the $h$-transform was implemented according to the parametrisation in Section 3.3 using an attention U-Net [16] for $\mathrm{NN}_1^\phi$. The network $\mathrm{NN}_2^\phi$ in the residual pathway was implemented as a small three layer fully connected network with SiLU activation functions, predicting a single scalar value. The final layer in $\mathrm{NN}_1^\phi$ was initialised with zeros. All weights in $\mathrm{NN}_2^\phi$ where initialised with zeros, expect for the bias in the last layer, which was initialised to 0.01. We found that this initialisation was stable over all imaging tasks, i.e., close to the unconditional model. All tasks on ImageNet were noiseless, i.e., the log-likelihood $p(\boldsymbol{x}|\boldsymbol{y})$ would be a Delta function. However, for the guidance term $\nabla_{\hat{\boldsymbol{x}}_0} \ln p(\boldsymbol{y}|\hat{\boldsymbol{x}}_0)$ we always approximated the log-likelihood using a Gaussian with $\sigma_y = 1.0$.

**Computed Tomography**  The 2016 American Association of Physicists in Medicine (AAPM) grand challenge dataset [45] contains CT scans of 10 patients. We only make use of the abdomen scans. We use the scans of 9 patients to train both the unconditional model and the $h$-transform. The remaining patient, i.e., id L035 with 87 slices, is used for testing only. We used a bilinear interpolation to resize the images to $256 \times 256$px. The unconditional model follows the same architecture as in [11] with about 374M parameters. The model was trained for 300 epochs using the Adam optimiser [33]. We used exponential moving average on the weights as in [66] with a parameter of 0.999.

The LoDoPab-CT dataset [35] is a collection of human chest CT scans. We resize the images to $256 \times 256$px using a bilinear interpolation. The training set contains 35820 images and was used to train the unconditional diffusion model. The validation set contains 3522 images and was used to train the $h$-transform model and for a hyperparameter search for DPS and RED-diff. Finally, we take 178 images (every 20th) from the test set to compute the final results. The unconditional model is an attention U-Net [16] with about 133M parameters. The unconditional model was trained for 75 epochs, corresponding to about 671625 gradient steps using the Adam optimiser [33]. We used exponential moving average on the weights as in [66] with a parameter of 0.999.

For both datasets we used the same parametrisation for the $h$-transform, with an attention U-Net [16] and a residual path as in Section 3.3. The $h$-transform has about 23M parameters. Thus, the $h$-transform is about 6% and 17% the size of the unconditional model for AAPM and LoDoPab-CT, respectively. For LoDoPab-CT, the $h$-transform was trained on a set of 3522 images, while for AAPM we trained the $h$-transform on the same dataset as the unconditional model. The forward operator $A$ is linear and we added Gaussian noise with standard deviation $\sigma_y = 1.0$. Thus, the gradient of the log likelihood reduces to

$$\nabla_{\hat{\boldsymbol{x}}_0} \ln p(\boldsymbol{y}|\hat{\boldsymbol{x}}_0) = -\frac{1}{2\sigma_y^2} A^*(A\hat{\boldsymbol{x}}_0 - \boldsymbol{y}),$$

where $A^*$ denotes the backprojection. Here, instead of the backprojection, we made use of the filtered backprojection, which is a common technique for computed tomography [25].

**Inpainting**  We made use of the pre-trained ImageNet dataset. For this task, we also amortised over the different sampling masks. We implemented the $h$-transform using an attention U-Net [16] as the base network $\mathrm{NN}_1$ with an initial convolution with 13 channels, i.e., the noisy image $\boldsymbol{x}_t$, the denoised estimate $\hat{\boldsymbol{x}}_0$, the observation $\boldsymbol{y}$ (where the missing pixels filled with zeros), the mask $M$ and the cheap

guidance term $\nabla_{\hat{\boldsymbol{x}}_0} \ln p(\boldsymbol{y}|\hat{\boldsymbol{x}}_0)$. The pre-trained unconditional score model has 550M parameters, and we use a fine-tuning network with 23M parameters (4.2% of the size). The fine-tuning network was trained for 200 epochs using a batch size of 16, and the Adam optimiser with a learning rate of $5e^{-4}$ and annealing.

**Super-resolution**  We made use of the pre-trained ImageNet dataset. We implemented the $h$-transform using an attention U-Net [16] as the base network $\text{NN}_1$ with an initial convolution with 9 channels, i.e., the noisy image $\boldsymbol{x}_t$, the denoised estimate $\hat{\boldsymbol{x}}_0$, the observation $\boldsymbol{y}$ bilinear upsampled to the original size and the cheap guidance term $\nabla_{\hat{\boldsymbol{x}}_0} \ln p(\boldsymbol{y}|\hat{\boldsymbol{x}}_0)$. The pre-trained unconditional score model has 550M parameters, and we use a fine-tuning network with 23M parameters (4.2% of the size). The fine-tuning network was trained for 200 epochs using a batch size of 16, and the Adam optimiser with a learning rate of $5e^{-4}$ and annealing.

**HDR**  For HDR the forward operator is given by $\mathcal{A}(\boldsymbol{x}) = \text{clip}(2x; -1, 1)$ for the RGB image scaled to $[-1, 1]$. This leads to a reduction of high intensity values. We approximated the cheap guidance term

$$\nabla_{\hat{\boldsymbol{x}}_0} \ln p(\boldsymbol{y}|\hat{\boldsymbol{x}}_0) \approx 0.5(\mathcal{A}(\hat{\boldsymbol{x}}_0) - \boldsymbol{y}),$$

and found this to achieve good results. The pre-trained unconditional score model has 550M parameters, and we use a fine-tuning network with 23M parameters (4.2% of the size). The fine-tuning network was trained for 200 epochs using a batch size of 16, and the Adam optimiser with a learning rate of $5e^{-4}$ and annealing.

**Phase retrieval**  The $h$-transform was again implemented as an attention U-Net [16] as the base network $\text{NN}_1$. The observations in phase retrieval corresponds to the magnitude values of the Fourier transform. In our initial experiments, we feed these observation directly into the model. However, this model failed to create convincing reconstruction. Instead we used two rough reconstruction. For the first initial reconstruction, we used the phase of the unconditional Tweedie estimate $\hat{\boldsymbol{x}}_0$ and the magnitude of the observation to construct an image. For the second initial reconstruction, we ran 350 steps of an ER algorithm [21] with a random initialisation. Further, we used the cheap guidance term $\nabla_{\hat{\boldsymbol{x}}_0} \|\boldsymbol{y} - \mathcal{A}(\hat{\boldsymbol{x}}_0)\|_2^2$, calculated using torch autograd. The $h$-transform had about 23M parameter, i.e., 4% of the size of the unconditional model.

**Non-linear Deblurring**  For non-linear deblurring we found that there was little improvement when using the cheap guidance term $\nabla_{\hat{\boldsymbol{x}}_0} \|\boldsymbol{y} - \mathcal{A}(\hat{\boldsymbol{x}}_0)\|_2^2$, calculated using torch autograd. As the forward operator is defined by a trained neural network this additional autograd term adds to the computational expense. We found that the DEFT provided results of a similar quality, with a reduced computational burden, by using $\mathcal{A}(\mathcal{A}(\hat{\boldsymbol{x}}_0) - \boldsymbol{y})$ as a rough approximation. The pre-trained unconditional score model has 550M parameters, and we use a fine-tuning network with 23M parameters (4.2% of the size). The fine-tuning network was trained for 200 epochs using a batch size of 16, and the Adam optimiser with a learning rate of $5e^{-4}$ and annealing.

### E.2  Protein Motif Scaffolding

**Diffusion process**  We use a discrete-time DDPM [28] formulation for the diffusion model with $N = 1000$ steps and cosine $\beta$-schedule [16].

**Noise model**  The denoising model $\varepsilon_\theta$ is adapted from the Genie diffusion model [37]. In Genie, the denoiser architecture consists of an SE(3)-invariant encoder and an SE(3)-equivariant decoder. While the network uses Frenet-Serret frames as intermediate representations, the diffusion process itself is defined in Euclidean space over the $C_\alpha$ coordinates. Similar to AlphaFold2, the denoiser network consists of a single representation track that is initialised via a single feature network and a pair representation track that is initialised via a pair feature network. These two representations are further transformed via a pair transform network and are used in the decoder for noise prediction via IPA [30].

To evaluate unconditional sampling-based methods, we used a pre-trained version of the unconditional Genie model.

To evaluate the AMORTISED approach (Alg. 4), we perform a minor modification to the unconditional Genie model as described in [17]: we add an additional conditional pair feature network that takes the motif frames as input with the ground truth coordinates for the motif and 0 as values for all other coordinates that are not part of the motif. The output of this motif-conditional pair feature network is concatenated with the output of the unconditional pair feature network to form an intermediate dimension of twice the channel size compared to the unconditional model before being linearly projected down to the channel size of the unconditional model. From then onward the output is processed by the remaining Genie components as in the unconditional model. The implementation is therefore similar to the image case, where the motif features are presented as additional input and the model learns to use these for reconstructing the motif. This minor alteration of the Genie architecture means that the amortised network has $4.162M$ parameters while the unconditional Genie networks have $4.087M$ parameters ($\sim 1.8\%$ fewer). In 80% of the training steps for the amortised model, we pass a condition to the network. The other 20% contains an empty mask consisting of only 0's.

For the DEFT implementation, we follow a similar way of feeding in the additional inputs via conditional pair feature networks, but as part of the downsized h-transform model as described in the main text.

**Metrics**   We measure the performance of the methods across two axes: designability and success rate. To assess whether a particular protein scaffold is *designable*, we run the same pipeline as [37], consisting of an inverse folding generated $C_\alpha$ backbones with ProteinMPNN and then re-folding the designed sequences via ESMFold. The considered metrics and their corresponding thresholds are the following:

- scTM > 0.5: This refers to the TM-score between the structure that's been designed and the predicted structure based on self-consistency as previously described. The scTM-score ranges from 0 to 1. Higher scores indicate a higher likelihood that the input structure can be designed.
- scRMSD < 2 Å : The scRMSD metric is akin to the scTM metric. However, it uses the RMSD (Root Mean Square Deviation) to measure the difference between the designed and predicted structures, instead of the TM-score. This metric is more stringent than scTM as RMSD, being a local metric, is more sensitive to minor structural variances.
- pLDDT > 70 and pAE < 5: Both scTM and scRMSD metrics depend on a structure prediction method like AlphaFold2 or ESMFold to be reliable. Hence, additional confidence metrics such as pLDDT and pAE are employed to ascertain the reliability of the self-consistency metrics.

In addition, we want to judge whether the motif scaffolding was successful or not. Therefore, similar to previous work by [77], we calculate the motifRMSD between the predicted design structure and the original input motif and judge samples with < 1 Å motifRMSD as a successful motif scaffold.

We follow previous work and call a sample a "success" if scRMSD < 2 Å  and motifRMSD < 1 Å . Similar to previous work we call a task "solved" if among 100 samples for this task at least 1 sample is a success.

# F   Additional Results

## F.1   Ablation of the DEFT parametrisation

The parametrisation of the $h$-transform is motivated by the sampling theory in Section 3.3. We evaluate different parametrisations of this choice for the CT experiment on LoDoPab-CT. For all architecture choices, we used the same training setup. For quantitative results, see Table 6. In particular, we see that the naive choice $\mathrm{NN}_1^\phi(\boldsymbol{x}, A^*\boldsymbol{y}, t)$ only achieves a PSNR of 26.62dB. Note, that we do not input the observations directly into the network, but first transform them using the backprojection, which is a standard technique for computed tomography reconstruction [3]. In CT the observations correspond to sinograms and have a different geometry to the images. In difference, if we add the unconditional Tweedie estimate $\hat{\boldsymbol{x}}_0$ to the model architecture, we get an improvement to 34.04dB. This shows that it is beneficial to supply the $h$-transform with the information of the unconditional diffusion model. Further, given our architecture, i.e., adding the additional information

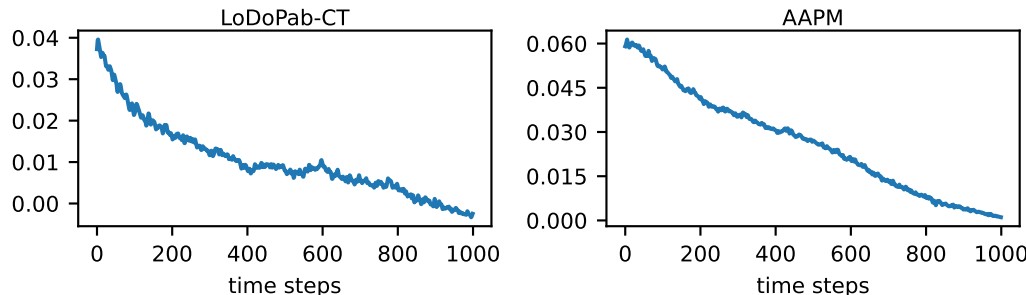

Figure 6: The trained residual scaling network $\text{NN}_2^\phi$ in the DEFT architecture (see Section 3.3) for computed tomography reconstruction on LoDoPab-CT and AAPM.

Table 6: PSNR and SSIM for computed tomography on LoDoPab-CT with different parametrisation of the $h$-transform.

| Parametrisation | PSNR | SSIM |
|---|---|---|
| $\text{NN}_1^\phi(\boldsymbol{x}, \hat{\boldsymbol{x}}_0, \nabla_{\hat{\boldsymbol{x}}_0} \ln p(\boldsymbol{y}\|\hat{\boldsymbol{x}}_0), t) + \text{NN}_2^\phi(t)\nabla_{\hat{\boldsymbol{x}}_0} \ln p(\boldsymbol{y}\|\hat{\boldsymbol{x}}_0)$ | 35.81 | 0.876 |
| $\text{NN}_1^\phi(\boldsymbol{x}, \nabla_{\hat{\boldsymbol{x}}_0} \ln p(\boldsymbol{y}\|\hat{\boldsymbol{x}}_0), t) + \text{NN}_2^\phi(t)\nabla_{\hat{\boldsymbol{x}}_0} \ln p(\boldsymbol{y}\|\hat{\boldsymbol{x}}_0)$ | 35.74 | 0.875 |
| $\text{NN}_1^\phi(\boldsymbol{x}, \hat{\boldsymbol{x}}_0, A^*\boldsymbol{y}, t)$ | 34.04 | 0.851 |
| $\text{NN}_1^\phi(\boldsymbol{x}, A^*\boldsymbol{y}, t)$ | 26.62 | 0.724 |

| Method | PSNR ($\uparrow$) | SSIM ($\uparrow$) | LPIPS ($\downarrow$) | Time in hrs ($\downarrow$) |
|---|---|---|---|---|
| RED-diff | 45.00 | 0.98 | $1.2e^{-3}$ | 7.9 |
| DEFT | 64.64 | 0.99 | $1.0e^{-5}$ | 5.2 |

Table 7: Results on the non-linear blurring operator from [44], where both methods achieve very high reconstruction and perceptual clarity due to the operator resulting in a trivial forward operation, rather than a non-linear blur. On this task, we still find DEFT to outperform RED-diff, with less time taken overall.

of the log-likelihood term, achieves a PSNR of $35.81$dB. Further, we show the learning residual scaling network $\text{NN}_2^\phi$ in Figure 6. We observe a similar behaviour for all tasks,i.e., at the start of sampling $t \approx 1000$ the scaling network assigned a small weighting to the guidance part, which increases during sampling ($t \to 0$).

### F.2  Non-linear Deblurring: Implementation from [44]

We find that the original implementation of the non-linear forward operator from the codebase provided in [44] results in an almost trivial reconstruction task, due to incorrect loading of weights for the neural network used for the non-linear deblurring. As a result, both RED-diff and DEFT can achieve highly performant results on this trivial task, as shown in Table 7. For the non-linear deblurring tasks in the main text in Table 2, we load the weights for the neural network correctly, and see much lower performance, corresponding to the difficulty of the non-linear task.

## G  Generalised $h$-transform and Stochastic Control

Thanks to our formal framework in this section we develop a new VI objective for learning the conditional score in the noisy inverse problems setting. That is by minimising the following ELBO

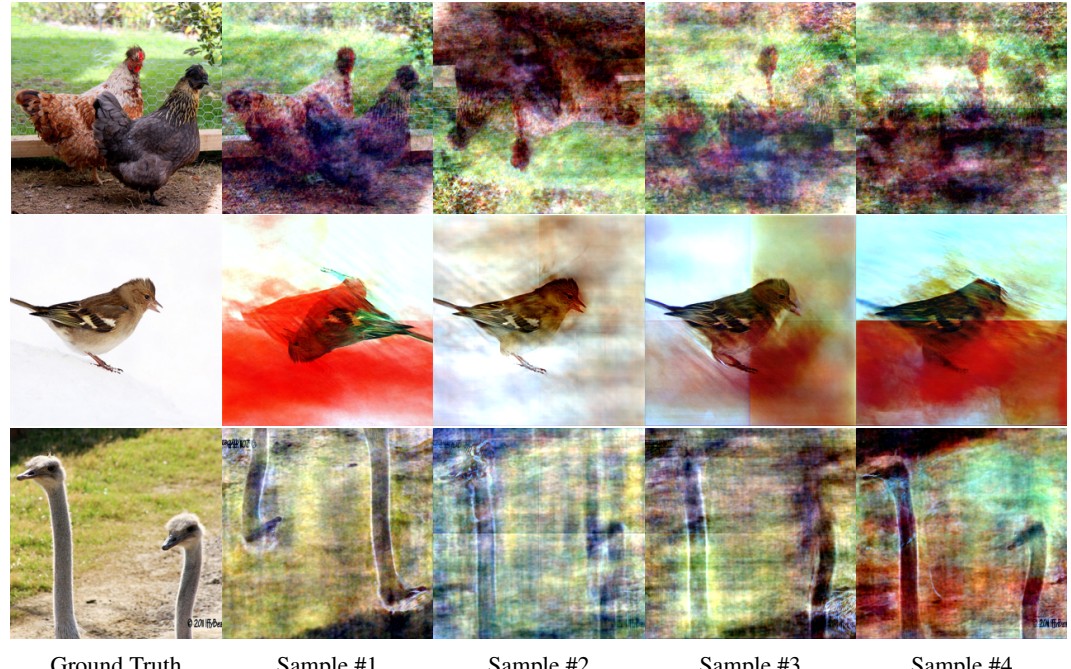

| Ground Truth | Sample #1 | Sample #2 | Sample #3 | Sample #4 |

Figure 7: Example reconstructions for phase retrieval. Phase retrieval has many local minima, which fully satisfy the data consistency constraints (e.g., complex conjugate, global phase sign). We often see flipped (and perturbed) images as our reconstruction. In some instances, even only one colour channel is flipped. This leads to a strong diversity of samples.

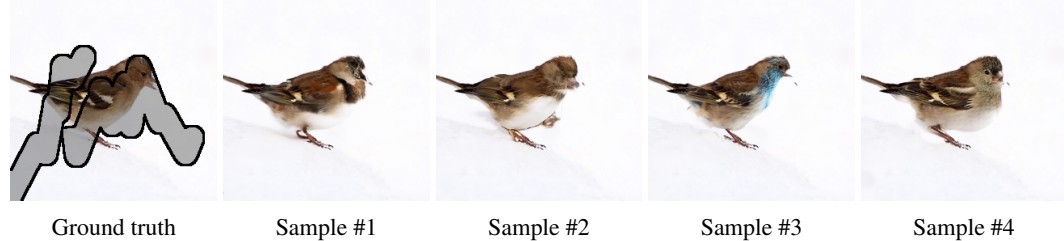

| Ground truth | Sample #1 | Sample #2 | Sample #3 | Sample #4 |

Figure 8: Diversity of samples for inpainting. On the left, we show the inpainting mask as a overlay over the ground truth.

with respect to an additional fine-tuning network, one can learn the conditional score

$$h^* = \arg\min_f \mathbb{E}_{\mathbb{Q}} \left[ \frac{1}{2} \int_0^T \sigma_t^2 ||h(\boldsymbol{H}_t)||^2 \mathrm{d}t \right] - \mathbb{E}_{\boldsymbol{H}_0 \sim \mathbb{Q}_0}[\ln p(\boldsymbol{y}|\boldsymbol{H}_0)], \tag{37}$$

where $\boldsymbol{H}_t$ follows the unconditioned score SDE with an added control $h$. This objective provides a way to learn the conditioned SDE from the unconditioned one, without making Gaussian approximations. We formalise this connection in the following proposition.

**Proposition G.1.** *The following stochastic control problem*

$$h^* = \arg\min_f \mathbb{E}_{\mathbb{Q}} \left[ \frac{1}{2} \int_0^T \sigma_t^2 ||h(\boldsymbol{H}_t)||^2 \mathrm{d}t \right] - \mathbb{E}_{\boldsymbol{H}_0 \sim \mathbb{Q}_0}[\ln p(\boldsymbol{y}|\boldsymbol{H}_0)] \tag{38}$$

*with*

$$\boldsymbol{H}_T \sim Q_T^{f_t}[p(\boldsymbol{x}_0|\boldsymbol{y})]$$
$$\mathrm{d}\boldsymbol{H}_t = \left( f_t(\boldsymbol{H}_t) - \sigma_t^2(\nabla_{\boldsymbol{H}_t} \ln p_t(\boldsymbol{H}_t) + h_t(\boldsymbol{H}_t)) \right) \mathrm{d}t + \sigma_t \overline{\mathrm{d}\mathbf{W}}_t, \tag{39}$$

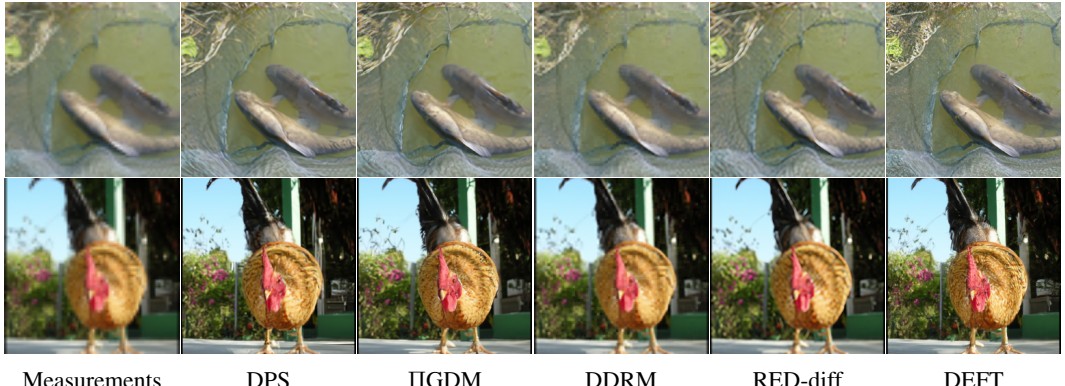

| Measurements | DPS | ΠGDM | DDRM | RED-diff | DEFT |

Figure 9: Results for 4x super-resolution.

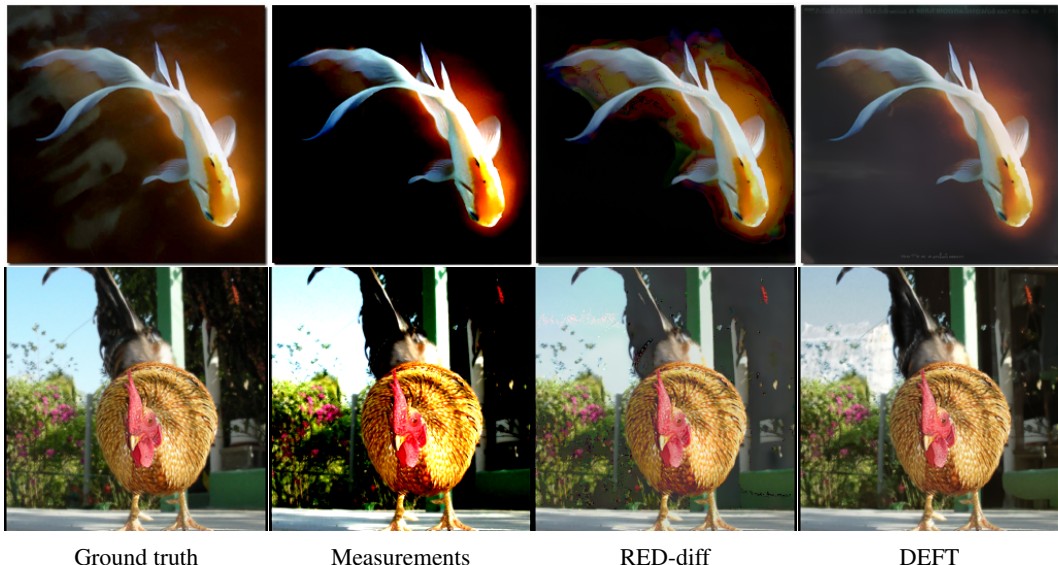

| Ground truth | Measurements | RED-diff | DEFT |

Figure 10: Results for non-linear HDR. Similar to [44], we found that DPS does not converge to a good solution, returning often black or images not consistent with the ground truth. Given that the forward operator $\mathcal{A}(\boldsymbol{x}) = \mathrm{clip}(2\boldsymbol{x} - 1, 1)$ has a zero gradient for $|\boldsymbol{x}| \geq 0.5$ the guidance term is often not informative.

*is minimised by the conditional score SDE in Equation* (28)*, that is*

$$h_t^*(\boldsymbol{x}) = \nabla_{\boldsymbol{x}} \ln \mathbb{E}_{\boldsymbol{X}_0 \sim p_{0|t}(\cdot|\boldsymbol{x})}[p(\boldsymbol{y}|\boldsymbol{X}_0)] = \nabla_{\boldsymbol{x}} \ln p_{y|t}(\boldsymbol{y}|\boldsymbol{x}). \tag{40}$$

*Furthermore, $h^*$ solves an associated half-bridge problem [7] with the SDE in Eqn.* (3) *as its reference process and $p(\boldsymbol{x}_0|\boldsymbol{y})$ as its source distribution.*

*Proof.* The derivation for this objective is inspired by the sequential Bayesian learning scheme proposed in Lemma 1, Appendix B of [75].

Let $\mathbb{P}$ denote the distribution for the forward SDE in Eqn. (25). Now consider the following variational problem termed a half-bridge [7, 15, 73].

$$\mathbb{Q}^* = \underset{\mathbb{Q}:\mathbb{Q}_0 = p(\boldsymbol{x}_0|\boldsymbol{y})}{\arg\min} D_{\mathrm{KL}}(\mathbb{Q}||\mathbb{P}) \tag{41}$$

where the constraint enforces that at time 0 we hit the target posterior $p(\boldsymbol{x}|\boldsymbol{y})$ then via standard results in half bridges we know that the above optimisation problem has an unconstrained formulation (e.g.

see [74]) that is $d\mathbb{Q}^* = d\mathbb{P}\frac{dp(\boldsymbol{x}|\boldsymbol{y})}{d\mathbb{P}_0}$ Now following [75] we notice that we can cancel the $p_{\text{data}}$ prior in the posterior term:

$$d\mathbb{P}\frac{dp(\boldsymbol{x}|\boldsymbol{y})}{d\mathbb{P}_0} = d\mathbb{P}\frac{dp(\boldsymbol{x}|\boldsymbol{y})}{dp_{\text{data}}} = d\mathbb{P}\frac{dp(\boldsymbol{y}|\boldsymbol{x})}{dp(\boldsymbol{y})} \tag{42}$$

and thus:

$$\mathbb{Q}^* = \underset{\mathbb{Q}}{\arg\min}\, D_{\text{KL}}(\mathbb{Q}||\mathbb{P}) - \mathbb{E}_{\boldsymbol{H}_0 \sim \mathbb{Q}_0}[\ln p(\boldsymbol{y}|\boldsymbol{H}_0)] \tag{43}$$

with $\mathbb{Q}_T = \text{Law}\,(\boldsymbol{X}_T) \approx \mathcal{N}(0, \mathbf{I})$ when $\boldsymbol{X}_0 \sim p(\boldsymbol{x}|\boldsymbol{y})$. Furthermore, we can parametrise $\mathbb{P}$ as :

$$d\boldsymbol{X}_t = \left(f_t(\boldsymbol{X}_t) - \sigma_t^2 \nabla_{\boldsymbol{X}_t} \ln p_t(\boldsymbol{X}_t)\right) dt + \sigma_t \overline{d\mathbf{W}_t}, \quad \boldsymbol{X}_0 \sim Q_T^{f_t}[p_{\text{data}}(\boldsymbol{x}_0)] \tag{44}$$

and thus $\mathbb{Q}$ as

$$\boldsymbol{H}_T \sim Q_T^{f_t}[p(\boldsymbol{x}_0|\boldsymbol{y})]$$
$$d\boldsymbol{H}_t = \left(f_t(\boldsymbol{H}_t) - \sigma_t^2(\nabla_{\boldsymbol{H}_t} \ln p_t(\boldsymbol{H}_t) + h_t(\boldsymbol{H}_t))\right) dt + \sigma_t \overline{d\mathbf{W}_t}, \tag{45}$$

then via Girsanov Theorem we can re-express the KL term in Eqn. (43) as

$$h^* = \underset{h}{\arg\min}\, \mathbb{E}_{\mathbb{Q}}\left[\frac{1}{2}\int_0^T \sigma_t^2 ||h(\boldsymbol{H}_t)||^2 dt\right] - \mathbb{E}_{\boldsymbol{H}_0 \sim \mathbb{Q}_0}[\ln p(\boldsymbol{y}|\boldsymbol{H}_0)]. \tag{46}$$

Now noticing that Eqn. (46) is a standard stochastic control problem [48, 31] we can characterise its minimiser as (using Theorem 2.2 in [48] and the Hopf-Cole transform [23])

$$h_t^*(\boldsymbol{x}) = \nabla_{\boldsymbol{x}} \ln \mathbb{E}_{\boldsymbol{X}_0 \sim p_{0|t}(\cdot|\boldsymbol{x})}[p(\boldsymbol{y}|\boldsymbol{X}_0)], \tag{47}$$

and thus the SDE in Eqn. 45 with $h_t^*$ hits the target posterior $p(\boldsymbol{y}|\boldsymbol{x}_0)$ at time 0 as it is the minimiser of half-bridge posed in Eqn. (41). $\qquad\square$

Notice that in the case of a VP-SDE, the stochastic control objective reduces to:

$$\underset{h}{\arg\min}\, \mathbb{E}_{\mathbb{Q}}\left[\int_0^T \frac{\beta_t}{2}||h(\boldsymbol{H}_t)||^2 dt\right] - \mathbb{E}_{\boldsymbol{H}_0 \sim \mathbb{Q}_0}[\ln p(\boldsymbol{y}|\boldsymbol{H}_0)] \tag{48}$$

**Discretisation of inverse problem objective** Following [74] we will discretise the objective presented in Eqn. (48). Let us consider the pre-trained score SDE with an added tuning network:

$$\boldsymbol{H}_T \sim \mathcal{N}(0, I)$$
$$d\boldsymbol{H}_t = -\beta_t(\boldsymbol{H}_t + 2s_\theta(\boldsymbol{H}_t) + 2h_\phi(\boldsymbol{H}_t)) dt + \sqrt{2\beta_t}\, \overline{d\mathbf{W}_t}, \tag{49}$$

now using an exponential-like discretisation [10] (Ideally we want to discretise in the same way we trained the model):

$$\boldsymbol{H}_{t_K} \sim \mathcal{N}(0, I)$$
$$\boldsymbol{H}_{t_{k-1}} = \left(\sqrt{1-\alpha_k}\boldsymbol{H}_{t_k} + 2(1-\sqrt{1-\alpha_k})\left(s_{\theta^*}(\boldsymbol{H}_{t_k}) + h_\phi(\boldsymbol{H}_{t_k})\right)\right) + \sqrt{\alpha_k}\varepsilon_k, \tag{50}$$

where $\alpha_k = 1 - \exp\left(\int_{t_{k-1}}^{t_k} \beta_s ds\right)$, note we will denote the distribution of the above discrete time chain as $q_\phi$. Now if we follow the sketch in Proposition 3 of [74] the discretised objective then becomes:

$$\underset{\phi}{\arg\min}\, \mathbb{E}_{\boldsymbol{H}\sim q_\phi}\left[2\sum_{k=1}^K \frac{\lambda_k^2}{\alpha_k}||h_\phi(k, \boldsymbol{H}_{t_k})||^2 - \ln p(\boldsymbol{y}|\boldsymbol{H}_0)\right] \tag{51}$$

where $\lambda_k = 1 - \sqrt{1-\alpha_k}$. For a more stable/simple objective following [74] we can make the approximation $\lambda_k = 1 - \sqrt{1-\alpha_k} \approx \alpha_k/2$ for small time steps. This leads to the following iteration (which is possibly more akin to the training update being used):

$$\boldsymbol{H}_{t_K} \sim \mathcal{N}(0, I)$$
$$\boldsymbol{H}_{t_{k-1}} = \left(\sqrt{1-\alpha_k}\boldsymbol{H}_{t_k} + \alpha_k\left(s_{\theta^*}(\boldsymbol{H}_{t_k}) + h_\phi(\boldsymbol{H}_{t_k})\right)\right) + \sqrt{\alpha_k}\varepsilon_k, \tag{52}$$

and objective:

$$\underset{\phi}{\arg\min}\, \mathbb{E}_{\boldsymbol{H}\sim q_\phi}\left[\sum_{k=1}^K \frac{\alpha_k}{2}||h_\phi(k, \boldsymbol{H}_{t_k})||^2 - \ln p(\boldsymbol{y}|\boldsymbol{H}_0)\right]. \tag{53}$$

**Discrete Time Intuition** For further intuition, we will provide a discrete-time derivation as to how this objective arises. Consider the following discrete-time VP-SDE (i.e let. $p_{k+1|k}(\boldsymbol{h}_{t_{k+1}}|\boldsymbol{h}_{t_k}) = \mathcal{N}(\boldsymbol{h}_{t_{k+1}}|\sqrt{1-\alpha_k}\boldsymbol{h}_{t_k}, \alpha_k))$ starting from the posterior:

$$p(\boldsymbol{h}_{t_1:t_k}) = \frac{p(\boldsymbol{y}|\boldsymbol{h}_0)p_{\text{data}}(\boldsymbol{h}_0)}{p(\boldsymbol{y})} \prod_{k=1}^{K} p_{k+1|k}(\boldsymbol{h}_{t_{k+1}}|\boldsymbol{h}_{t_k}) \tag{54}$$

Now applying Bayes rule and [2, Section 5] $p_{k+1|k}(\boldsymbol{h}_{t_{k+1}}|\boldsymbol{h}_{t_k}) = \frac{p^{\text{uncnd}}_{k|k+1}(\boldsymbol{h}_{t_k}|\boldsymbol{h}_{t_{k+1}})p^{\text{uncnd}}_{k+1}(\boldsymbol{h}_{t_{k+1}})}{p^{\text{uncnd}}_k(\boldsymbol{h}_{t_k})}$ and telescoping to cancel the marginals we have:

$$p(\boldsymbol{h}_{t_1:t_k}) = \frac{p^{\text{uncnd}}_K(\boldsymbol{h}_T)p(\boldsymbol{y}|\boldsymbol{h}_0)\,\cancel{p}_{\text{data}}(\boldsymbol{h}_0)}{p(\boldsymbol{y})\,\cancel{p}_{\text{data}}(\boldsymbol{h}_0)} \prod_{k=1}^{K} p^{\text{uncnd}}_{k|k+1}(\boldsymbol{h}_{t_k}|\boldsymbol{h}_{t_{k+1}}) \tag{55}$$

$$= \frac{p^{\text{uncnd}}_K(\boldsymbol{h}_T)p(\boldsymbol{y}|\boldsymbol{h}_0)}{p(\boldsymbol{y})} \prod_{k=1}^{K} p^{\text{uncnd}}_{k|k+1}(\boldsymbol{h}_{t_k}|\boldsymbol{h}_{t_{k+1}}), \tag{56}$$

where $p^{\text{uncnd}}_{k|k+1}(\boldsymbol{h}_{t_k}|\boldsymbol{h}_{t_{k+1}})$ is the transition density of the unconditional score SDE and $p^{\text{uncnd}}_k(\boldsymbol{h}_{t_k})$ correspond to its marginals. Now we would like to learn a backwards process that matches the above process (reverses the VP-SDE starting from the posterior). We can do so by minimising the KL:

$$D_{\text{KL}}(q^\phi||p) \propto \mathbb{E}_q \left[ \ln \frac{\prod_k q^\phi_{k|k+1}(\boldsymbol{h}_{t_k}|\boldsymbol{h}_{t_{k+1}})}{\prod_k p^{\text{uncnd}}_{k|k+1}(\boldsymbol{h}_{t_k}|\boldsymbol{h}_{t_{k+1}})} - \ln p(\boldsymbol{y}|\boldsymbol{h}_0) \right] \tag{57}$$

where we can approximate the score transition via:

$$p^{\text{uncnd}}_{k-1|k}(\boldsymbol{h}_{t_k}|\boldsymbol{h}_{t_{k+1}}) \approx \mathcal{N}(\boldsymbol{h}_{t_{k-1}}|\sqrt{1-\alpha_k}\boldsymbol{h}_{t_k} + 2(1-\sqrt{1-\alpha_k})s_{\theta^*}(\boldsymbol{h}_{t_k}), \alpha_k)$$

and parametrise the new conditional denoiser as

$$q^\phi_{k-1|k}(\boldsymbol{h}_{t_k}|\boldsymbol{h}_{t_{k+1}}) = \mathcal{N}(\boldsymbol{h}_{t_{k-1}}|\sqrt{1-\alpha_k}\boldsymbol{h}_{t_k} + 2(1-\sqrt{1-\alpha_k})\left(s_{\theta^*}(\boldsymbol{h}_{t_k}) + h_\phi(\boldsymbol{h}_{t_k})\right), \alpha_k)$$

making these two substitutions will lead to the objective in Eqn. (53).

## G.1 Related Work

Fine-tuning diffusion models via a optimal control perspective, e.g., devolved in [6], has received a lot of attention in recent years. In particular, in the context of fine-tuning with respect to a differentiable reward function, i.e., considering a tilted posterior [18],

$$\pi(\boldsymbol{x}_0) = \frac{e^{r(\boldsymbol{x}_0)}p_{\text{data}}(\boldsymbol{x}_0)}{\mathcal{Z}}, \tag{58}$$

where $e^{r(\boldsymbol{x}_0)}$ serves an equivalent role to the likelihood $p(\boldsymbol{y}|\boldsymbol{x}_0)$ in our setting.

The DRaFT framework [13] proposes a heuristic method to estimate the $h$-transform by only optimising the reward function. We want to highly that concurrently [72] develop the same stochastic control formulation as we do, and arrive at the same insight that the optimal starting distribution is given by $p_T = Q_T^{f_t}[\pi]$, however, they chose to learn this distribution which we argue is not necessary for diffusion models due to the mixing property of the OU process leading to a negligible error by approximating $Q_T^{f_t}[\pi] \approx \mathcal{N}(0, I)$ with a Gaussian.

## G.2 Connection to [18] - Value Function Bias

In [18], it is argued that minimising the stochastic control objective does not lead to hitting the posterior $p(\boldsymbol{x}_0|\boldsymbol{y})$ at time $t = 0$ due to bias introduced by the value function. We can apply Proposition G.1 to the tilted posterior $\pi(\boldsymbol{x}_0)$ from (58) which yields the following objective:

$$h^* = \arg\min_h \mathbb{E}_{\mathbb{Q}} \left[ \frac{1}{2} \int_0^T \sigma_t^2 ||h(\boldsymbol{H}_t)||^2 \text{d}t \right] - \mathbb{E}_{\boldsymbol{H}_0 \sim \mathbb{Q}_0}[r(\boldsymbol{H}_0)] \tag{59}$$

with

$$\boldsymbol{H}_T \sim Q_T^{f_t}[p(\boldsymbol{x}_0|\boldsymbol{y})]$$
$$\mathrm{d}\boldsymbol{H}_t = \left(f_t(\boldsymbol{H}_t) - \sigma_t^2(\nabla_{\boldsymbol{H}_t} \ln p_t(\boldsymbol{H}_t) + h_t(\boldsymbol{H}_t))\right) \mathrm{d}t + \sigma_t \overleftarrow{\mathrm{d}\mathbf{W}}_t, \tag{60}$$

then by Theorem 2.1 in [71] the optimal transition density of the controlled process is given by ($s \leq t$ and let $\tilde{h}(\boldsymbol{x}_t) = \ln p_{y|t}(\boldsymbol{y}|\boldsymbol{x}_t)$):

$$p_{s|t}^*(\boldsymbol{x}|\boldsymbol{y}) = e^{\tilde{h}(\boldsymbol{x},s) - \tilde{h}(\boldsymbol{y},t)} p_{s|t}^{\mathrm{ref}}(\boldsymbol{x}|\boldsymbol{y}) \tag{61}$$

where the log of $h$-transform $\tilde{h}$ (note we have used $\tilde{h}$ to denote the log of the $h$-transform) coincides with the negative of the value function in [18, 71]. Then for $s = 0$ and $t = T$ this induces the following joint distribution:

$$p_{0,T}^*(\boldsymbol{x}|\boldsymbol{y}) = e^{r(\boldsymbol{x}) - \ln \mathcal{Z} - \tilde{h}(\boldsymbol{y},T)} p_{s|t}^{\mathrm{ref}}(\boldsymbol{x}|\boldsymbol{y}) p_T(\boldsymbol{y}) \tag{62}$$

where [18] argue that in general the term $e^{-h(\boldsymbol{y},T)}$ induces a bias such that when we marginalise out $\boldsymbol{y}$, $p_0^*(\boldsymbol{x})$ is not the tilted distribution, more precisely:

$$p_0^*(\boldsymbol{x}) = e^{r(\boldsymbol{x}) - \ln \mathcal{Z}} \int e^{-\tilde{h}(\boldsymbol{y},T)} p_{0|T}^{\mathrm{ref}}(\boldsymbol{x}|\boldsymbol{y}) p_T(\boldsymbol{y}) \mathrm{d}\boldsymbol{y} \neq \frac{e^{r(\boldsymbol{x})} p_{\mathrm{data}}(\boldsymbol{x})}{\mathcal{Z}} \tag{63}$$

However, let's look more closely as to why this is the case; first let us re-express the h-transform at time $T$ as a ratio of densities:

$$e^{\tilde{h}(\boldsymbol{y},T)} = \int \frac{\mathrm{d}\pi}{\mathrm{d}p_{\mathrm{data}}}(\boldsymbol{y}_0) p_{0|T}^{\mathrm{ref}}(\boldsymbol{y}_0|\boldsymbol{y}_T) \mathrm{d}\boldsymbol{y}_0 \tag{64}$$

$$= \int \frac{e^{r(\boldsymbol{y}_0)} p_{\mathrm{data}}(\boldsymbol{y}_0)}{\mathcal{Z} p_{\mathrm{data}}(\boldsymbol{y}_0)} p_{0|T}^{\mathrm{ref}}(\boldsymbol{y}_0|\boldsymbol{y}_T) \mathrm{d}\boldsymbol{y}_0 \tag{65}$$

$$= \frac{1}{p_T^{\mathrm{ref}}(\boldsymbol{y}_T)} \int \pi(\boldsymbol{y}_0) p_{T|0}^{\mathrm{ref}}(\boldsymbol{y}_T|\boldsymbol{y}_0) \mathrm{d}\boldsymbol{y}_0 \tag{66}$$

$$= \frac{Q_T^{f_t}[\pi](\boldsymbol{y}_T)}{p_T^{\mathrm{ref}}(\boldsymbol{y}_T)} \tag{67}$$

substituting back into (62) and marginalizing we have:

$$p_0^*(\boldsymbol{x}) = e^{r(\boldsymbol{x}) - \ln \mathcal{Z}} \int p_{0|T}^{\mathrm{ref}}(\boldsymbol{x}|\boldsymbol{y}) p_T(\boldsymbol{y}) \frac{p_T^{\mathrm{ref}}(\boldsymbol{y})}{Q_T^{f_t}[\pi](\boldsymbol{y})} \mathrm{d}\boldsymbol{y} \tag{68}$$

now notice $p_T(\boldsymbol{y})$ is the distribution we simulate our stochastic control from which in our case we have chosen to be $p_T = Q_T^{f_t}[\pi]$ making the cancellation

$$p_0^*(\boldsymbol{x}) = e^{r(\boldsymbol{x}) - \ln \mathcal{Z}} \int p_{0|T}^{\mathrm{ref}}(\boldsymbol{x}|\boldsymbol{y}) p_T^{\mathrm{ref}}(\boldsymbol{y}) \mathrm{d}\boldsymbol{y} = \frac{e^{r(\boldsymbol{x})} p_{\mathrm{data}}(\boldsymbol{x})}{\mathcal{Z}} \tag{69}$$

Leading us to the following remark

**Remark 1.** *The choice of setting $p_T = Q_T^{f_t}[\pi]$ leads to removing the value function bias [18] in Equation 63 . Note the authors of [18] pursue a different avenue for removing this bias by altering the noise of the controlled SDE.*

**Proposition G.2.** *(Value function bias when approximating $Q_T^{f_t}[\pi] \approx \mathcal{N}(0, I)$) In practice, we often do not have access to $Q_T^{f_t}[\pi]$ and thus we may make an approximation with some tractable distribution $p_T$, then we can bound the value function bias as follows*

$$\left\| p_0^*(\boldsymbol{x}) - \frac{e^{r(\boldsymbol{x})} p_{\mathrm{data}}(\boldsymbol{x})}{\mathcal{Z}} \right\|_{\mathrm{TV}} \leq \left\| p_T - Q_T^f[\pi] \right\|_{\mathrm{TV}}, \tag{70}$$

*then in the VP-SDE with a time homogenous $\beta_t = \beta$ based diffusion model (for simplicity), where chose $p_T = \mathcal{N}(0, I)$ we can obtain a tight bound,*

$$\left\| p_0^*(\boldsymbol{x}) - \frac{e^{r(\boldsymbol{x})} p_{\mathrm{data}}(\boldsymbol{x})}{\mathcal{Z}} \right\|_{\mathrm{TV}} \leq C e^{-\beta T} \tag{71}$$

*for some constant $C > 0$, thus the value function bias [18] is exponentially small for score based diffusion models.*

*Proof.* Applying Theorem 17 from [15] we have:

$$\left\| p_0^*(\boldsymbol{x}) - \frac{e^{r(\boldsymbol{x})} p_{\text{data}}(\boldsymbol{x})}{\mathcal{Z}} \right\|_{\text{TV}} = \left\| P_0^{f_t + h_t^*}[\mathcal{N}(0, I)] - P_0^{f_t + h_t^*}[Q^f[\pi]] \right\|_{\text{TV}} \tag{72}$$

$$\leq \left\| \mathcal{N}(0, I) - Q_T^f[\pi] \right\|_{\text{TV}} = C e^{-\beta T}, \tag{73}$$

the final equality follows from the mixing properties of the OU process [4], where

$$P_0^{f_t + h_t^*}[\mu](\boldsymbol{x}) = \int p_{0|T}^*(\boldsymbol{x}|\boldsymbol{y}) \mu(\boldsymbol{x}) \mathrm{d}\boldsymbol{y} \tag{74}$$

$\square$

Finally we want to highlight that the benefits of Proposition G.2 can only be leveraged in score matching settings where we have a clear characterisation of the forward process and we are able to tractably characterise $p_0^*$ however in settings such as flow matching [38, 41] and stochastic interpolants [1] where the forward process is not explicitly characterised we wither have to learn $p_0^*$ like in [72] or use a memoryless noise schedule as proposed in [18].

### G.3 Scaling up the Control Objective

Naively trying to minimise Eqn. (37) is demanding, as the full chain has to be kept in memory, which is infeasible for high dimensional problems. To alleviate this problem, one could make use of the stochastic adjoint sensitivity method [36], in which an adjoint SDE is solved to estimate the gradients of the stochastic control loss in Theorem 3.1 3). This method has the advantage of a constant memory cost. However, the computational cost increases as both the reverse SDE and the adjoint SDE must be simulated. Instead, we discuss two alternative approaches to reduce the memory requirements.

**VarGrad** We can make use of a VarGrad [53, 54] type loss to reduce the memory requirements. In contrast to the KL loss of Eqn. (37), then the VarGrad loss is given by:

$$D_{\text{logvar}}(\mathbb{Q}, \mathbb{P}; \mathbb{W}) = \mathbb{E}_{\boldsymbol{H}_{0:T}^{g_t} \sim \mathbb{W}} \left[ \left( \ln \frac{\mathrm{d}\mathbb{Q}}{\mathrm{d}\mathbb{P}}(\boldsymbol{H}_{0:T}^{g_t}) - \mathbb{E}\left[ \ln \frac{\mathrm{d}\mathbb{Q}}{\mathrm{d}\mathbb{P}}(\boldsymbol{H}_{0:T}^{g_t}) \right] \right)^2 \right], \tag{75}$$

where $\mathbb{Q}$ and $\mathbb{P}$ are defined as in the proof of Proposition G.1, i.e., $\mathbb{Q}$ is given by the conditional SDE and $\mathbb{P}$ is given by the unconditional SDE. The RND in Eqn. (77) is evaluated at the trajectory of a reference process $\mathbb{W} = \text{Law}(\boldsymbol{H}_{0:T}^{g_t})$, given by

$$\boldsymbol{H}_T \sim Q_T^{f_t}[p(\boldsymbol{x}_0|\boldsymbol{y})]$$
$$\mathrm{d}\boldsymbol{H}_t = \left( f_t(\boldsymbol{H}_t) - \sigma_t^2(\nabla_{\boldsymbol{H}_t} \ln p_t(\boldsymbol{H}_t) + g_t(\boldsymbol{H}_t)) \right) \mathrm{d}t + \sigma_t \overline{\mathrm{d}\mathbf{W}}_t. \tag{76}$$

The Radon–Nikodym derivative (RND) in Eqn (75) can be evaluated as (Using the RND for time reverse SDEs see Equation 64 in [76]):

$$\ln \frac{\mathrm{d}\mathbb{Q}}{\mathrm{d}\mathbb{P}}(\boldsymbol{H}_{0:T}^{g_t}) = -\frac{1}{2} \int_0^T \sigma_t^2 \|h_t(\boldsymbol{H}_t^{g_t})\|^2 \mathrm{d}t + \int_0^T \sigma_t^2 (g_t^\top h_t)(\boldsymbol{H}_t^{g_t}) \mathrm{d}t - \ln p(\boldsymbol{y}|\boldsymbol{x}_0^{g_t})$$
$$+ \int_0^T \sigma_t h_t^\top (\boldsymbol{H}_t^{g_t}) \overline{\mathrm{d}\mathbf{W}}_t, \tag{77}$$

for the reference process $\mathbb{W}$. The core advantage of VarGrad is that we can choose this reference process. In particular, the choice $g_t = \text{stop\_grad}(h_t)$ gives us a way to detach the trajectories, saving us from having to score all gradients in memory.

**Trajectory Balance** An alternative to the VarGrad loss in Eqn. (75) is the following trajectory balance [42, 43] loss

$$\mathcal{L}_{\text{TB}}^{\mathbb{W}}(\mathbb{Q}, \mathbb{P}; k) = \mathbb{E}_{\boldsymbol{H}_{0:T}^{g_t} \sim \mathbb{W}} \left[ \left( \ln \frac{\mathrm{d}\mathbb{Q}}{\mathrm{d}\mathbb{P}}(\boldsymbol{H}_{0:T}^{g_t}) - k \right)^2 \right], \tag{78}$$

where $k \in \mathbb{R}$ is a learnable parameter and $\mathbb{W}$ is the same reference process as above. We again choose $g_t = \text{stop\_grad}(h_t)$ This loss is also motivated by a valid divergence, see [48]. In difference to the VarGrad loss (75), the inner expectation is exchanged with $k$, which approximates a running mean. In practice, we optimise $k$ and and $h_t$ at the same time.

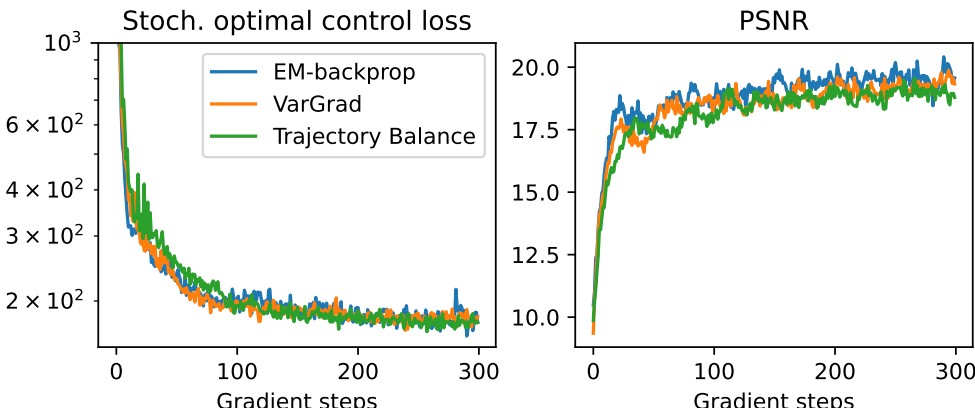

Figure 11: Left: Tracking the stochastic optimal control loss (37) for the three methods. Right: Mean PSNR of samples.

**Trajectory Subsampling** We observed that backpropagating gradients for only a random subset of discrete timesteps in the RND can be an effective strategy. This subsampling reduces memory costs at the expense of increased variance and the introduction of a small bias. Nevertheless, the reduced memory usage per example enables larger batch sizes, which can mitigate these effects. Notably, we found that backpropagating for only $20\%$ of the timesteps still achieves comparable performance.

### G.4 Stochastic Optimal Control - Experiments

We provide some initial proof of concept experiments on the MNIST dataset [34] of handwritten digits. In particular, we make use of a parallel beam Radon transform with 5 angles as the forward operator and perturb the observations with $10\%$ additive Gaussian noise. All SDEs are discretised using an Euler-Maruyma scheme and the integrals are estimated using simple quadrature rules. We use a non-equidistant time grid according to a square root function, i.e., let $0 = t_0 \leq \cdots \leq t_{K-1} = 1$ be an equidistant grid of $[0, 1]$ for $K$ time points. We then use $t_0^2, t_1^2, \ldots, t_{K-1}^2$ as the time points for evaluating the SDEs. This gives us a finer discretisation closer to $t = 0$. The unconditional MNIST model is based on the attention U-Net architecture [16] with about 3M parameters. The $h$-transform is implemented using the same DEFT parametrisation as in Section 3.3 with about $70\,000$ parameters. We compare `EM-backprop`, i.e., directly backpropagting through the discrete SDE solver (see e.g. [36]), with `VarGrad` and `TrajectoryBalance` from Section G.3. For `EM-backprop` we use a batch size of 16 and use 60-80 time steps. Instead, for `VarGrad` and `TrajectoryBalance` we were able to use a batch size of 26 and 80-140 time steps. For training we used a single GeForce RTX 3090 and the training time took about 1h. The results are presented in Figure 11, where the loss trajectory of Eqn. (37) and the mean PSNR of samples is shown. Both `VarGrad` and `TrajectoryBalance` are able to minimise the stochastic optimal control objective to the same extend as `EM-backprop`. Further, in Figure 12 we provide an overview of the `TrajectoryBalance` training. Here, we observe that $k$ is working as a estimator of the mean RND with a lower variance.

## H  Amortised Conditional Training

In this section, we discuss an objective for learning the full conditional score at training time in an amortised fashion instead of enforcing the constraint during inference time as before in reconstruction guidance approaches. This objective is akin to CDE [5] with the difference that we propose amortising over the the forward operator, for example in image inpainting or motif-scaffolding.

Note that since $\overline{P}_{0|t}(\boldsymbol{Y} = \boldsymbol{y}|\boldsymbol{X}_t = \boldsymbol{x}) = \overrightarrow{P}_{t|0}(\boldsymbol{x}|\boldsymbol{Y} = \boldsymbol{y})p_0(\boldsymbol{Y} = \boldsymbol{y})/p_t(\boldsymbol{X}_t = \boldsymbol{x})$, we can re-express the Doob's transformed SDE of a reversed OU process as:

$$\mathrm{d}\boldsymbol{H}_t = -\beta_t \left( \boldsymbol{H}_t + 2\nabla_{\boldsymbol{H}_t} \ln \overline{P}_{t|0}(\boldsymbol{H}_t|\boldsymbol{Y} = \boldsymbol{y}) \right) \mathrm{d}t + \sqrt{2\beta_t}\, \overline{\mathrm{d}\mathbf{W}}_t, \quad \boldsymbol{H}_T \sim \mathrm{Law}\left(\boldsymbol{X}_T\right).$$

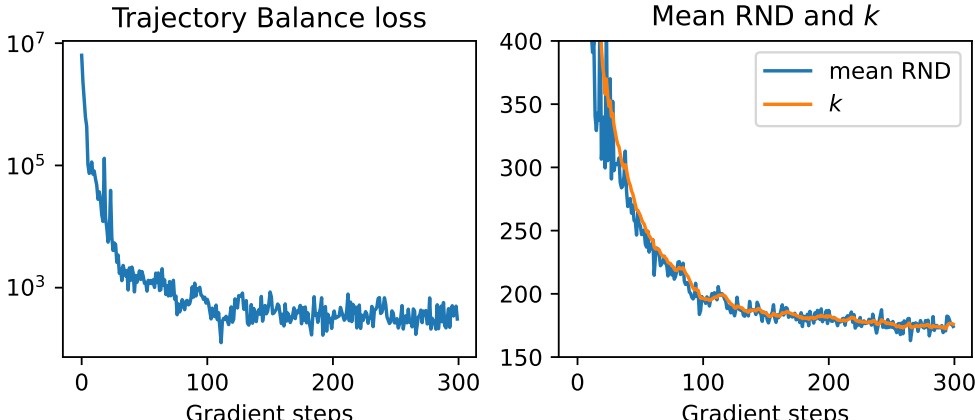

Figure 12: Training using `TrajectoryBalance`. Left: The trajectory balance objective loss (78) over training steps. Right: The mean RND and the trained $k$. We see that $k$ follows the mean RND. However, it has a smaller variance.

**Proposition H.1.** *The minimiser of*

$$f^* = \underset{h}{\arg\min} \ \underset{\substack{(\boldsymbol{X}_0, \boldsymbol{Y}) \sim p(\boldsymbol{x}_0, \boldsymbol{Y}) \\ t \sim \mathrm{U}(0,T), \boldsymbol{H}_t \sim p_{t|0}(\boldsymbol{x}_t|\boldsymbol{x}_0)}}{\mathbb{E}} \left[ ||h(t, \boldsymbol{H}_t, \boldsymbol{y}) - \nabla_{\boldsymbol{H}_t} \ln \vec{p}_{t|0}(\boldsymbol{H}_t|\boldsymbol{X}_0)||^2 \right], \tag{79}$$

*is given by the conditional score* $f_t^*(\boldsymbol{x}, \boldsymbol{y}) = \nabla_{\boldsymbol{x}} \ln \vec{p}_{t|0}(\boldsymbol{x}|\boldsymbol{Y} = \boldsymbol{y})$.

*Proof.* Via the mean squared error property of the conditional expectation the minimiser is given by:

$$h_t^*(\boldsymbol{x}, \boldsymbol{y}) = \mathbb{E}\left[ \nabla_{\boldsymbol{H}_t} \ln \vec{p}_{t|0}(\boldsymbol{X}_t|\boldsymbol{X}_0)|\boldsymbol{Y} = \boldsymbol{y}, \boldsymbol{H}_t = \boldsymbol{x} \right] \tag{80}$$

Then:

$$
\begin{aligned}
h_t^*(\boldsymbol{x}, \boldsymbol{y}) &= \int \nabla_{\boldsymbol{x}} \ln \vec{p}_{t|0}(\boldsymbol{x}|\boldsymbol{x}_0) \vec{p}_{0|t}(\boldsymbol{x}_0|\boldsymbol{H}_t = \boldsymbol{x}, \boldsymbol{Y} = \boldsymbol{y}) \mathrm{d}\boldsymbol{x}_0 \\
&= \int \frac{\nabla_{\boldsymbol{x}} \vec{p}_{t|0}(\boldsymbol{x}|\boldsymbol{x}_0)}{\vec{p}_{t|0}(\boldsymbol{x}|\boldsymbol{x}_0)} \frac{\vec{p}_{t|0}(\boldsymbol{H}_t = \boldsymbol{x}|\boldsymbol{x}_0, \boldsymbol{Y} = \boldsymbol{y}) p(\boldsymbol{x}_0|\boldsymbol{Y} = \boldsymbol{y},)}{p(\boldsymbol{H}_t = \boldsymbol{x}|\boldsymbol{Y} = \boldsymbol{y})} \mathrm{d}\boldsymbol{x}_0 \\
&= \frac{1}{p(\boldsymbol{H}_t = \boldsymbol{x}|\boldsymbol{Y} = \boldsymbol{y})} \int \frac{\nabla_{\boldsymbol{x}} \vec{p}_{t|0}(\boldsymbol{x}|\boldsymbol{x}_0)}{\vec{p}_{t|0}(\boldsymbol{x}|\boldsymbol{x}_0)} \vec{p}_{t|0}(\boldsymbol{H}_t = \boldsymbol{x}|\boldsymbol{x}_0) p(\boldsymbol{x}_0|\boldsymbol{Y} = \boldsymbol{y}) \mathrm{d}\boldsymbol{x}_0 \\
&= \frac{1}{p(\boldsymbol{H}_t = \boldsymbol{x}|\boldsymbol{Y} = \boldsymbol{y})} \nabla_{\boldsymbol{x}} \int \vec{p}_{t|0}(\boldsymbol{x}|\boldsymbol{x}_0) p(\boldsymbol{x}_0|\boldsymbol{Y} = \boldsymbol{y}) \mathrm{d}\boldsymbol{X}_0 \\
&= \frac{1}{\vec{p}(\boldsymbol{X}_t = \boldsymbol{x}|\boldsymbol{Y} = \boldsymbol{y})} \nabla_{\boldsymbol{x}} \vec{p}(\boldsymbol{X}_t = \boldsymbol{x}|\boldsymbol{Y} = \boldsymbol{y}) \tag{81} \\
&= \nabla_{\boldsymbol{x}} \ln \vec{p}(\boldsymbol{X}_t = \boldsymbol{x}|\boldsymbol{Y} = \boldsymbol{y}),
\end{aligned}
$$

where we use that $\vec{p}_{t|0}(\boldsymbol{H}_t = \boldsymbol{x}|\boldsymbol{x}_0, \boldsymbol{Y} = \boldsymbol{y}) = \vec{p}_{t|0}(\boldsymbol{H}_t = \boldsymbol{x}|\boldsymbol{x}_0)$ as $\boldsymbol{H}_t$ is independent from $\boldsymbol{Y}$ given $\boldsymbol{X}_0$. □

As with DEFT, for settings where $\mathcal{A}$ varies like in image completion we sample $\mathcal{A}$ randomly and amortise it over our learned $h$-transform, i.e. estimating $h_t^*(\boldsymbol{x}, \boldsymbol{y}, \boldsymbol{A})$.

We refer to this approach as *amortised* learning for conditional sampling, since practically the neural network approximating the (conditional) score is amortised over $\mathcal{A}$ and $\boldsymbol{y}$, instead of learning a separate network for each condition. This approach is also reminiscent of 'classifier free guidance' [27] where the score network is amortised over some auxiliary variable (e.g. as in text-to-image models [51]), or of RFDiffusion [77] where proteins are designed given a specific subset motif, or similar to [5]. See also Appendix B for a discussion of related conditional training methodologies.

Note that conditional amortised learning is different to 'classifier free guidance' as $\mathcal{A}$ is assumed to be known (e.g. an inpainting mask). Also note that due to its formulation, classifier guidance would be unable to noise a subset of $X$ (the motif) as we do and would instead be more akin to RFDiffusion.

## H.1 Relationship to Conditional denoising estimator (CDE)

Conditional denoising estimator (CDE) [5] is the adaptation of [27, 60] to inverse problem-like settings, deriving a variation of classifier-free guidance to a measurement model styled scenario. Whilst they do not focus on the measurement model, they estimate a very similar quantity as our Proposition 2.5

$$f^{\mathrm{CDE}}(\boldsymbol{x}, \boldsymbol{y}) = \nabla_{\boldsymbol{x}} \ln \vec{p}_{t|0}(\boldsymbol{x}|\boldsymbol{Y} = \boldsymbol{y}) \tag{82}$$

In contrast to to the amortised conditional training:

$$f^{\mathrm{amortised}}(\boldsymbol{x}, \boldsymbol{y}, \boldsymbol{A}) = \nabla_{\boldsymbol{x}} \ln \vec{p}_{t|0}(\boldsymbol{x}|\boldsymbol{Y} = \boldsymbol{y}, \mathcal{A} = \boldsymbol{A}) \tag{83}$$

when explicitly considering the distribution over the measurement model, one can see that the quantities are related to one another via marginalizing the measurement model $p_{\mathcal{A}}$. This introduces several practical and conceptual differences:

- If we consider in/out painting as an example, the score network estimating $f^{\mathrm{CDE}}$ is not explicitly aware of where in the image the missing pixels are. As a result, it must perform inference over $\mathcal{A}$ (effectively marginalizing it) in order to know where to complete the image. This is clearly a much harder task for a single network to learn than conditioning on $\mathcal{A}$ where we provide this information.
- Viewed under the lens of the h-transform, $f^{\mathrm{CDE}}$ can be viewed as amortising the event $\mathcal{A}(\boldsymbol{X}_0) = \boldsymbol{y}$ for random $\mathcal{A}$. It therefore falls under the soft constraint settings since $\mathcal{A}(\boldsymbol{X}_0)|\boldsymbol{X}_0$ is not a delta. Our quantity $f^{\mathrm{amortised}}$ is amortising over $\boldsymbol{A}(\boldsymbol{X}_0) = \boldsymbol{y}$ for deterministic $\boldsymbol{A}$ and is therefore part of the more classical hard constraint domain of Doobs transform. We believe amortising over these simpler deterministic events can offer an advantage in making the problem easier to learn.

## H.2 Comparison to $h$-transform fine-tuning

We compare the amortised training framework against our $h$-transform fine-tuning on the FLOWERS dataset. The preprocessing procedure consisted of centrally cropping the image to size $64 \times 64$, and rescaling to pixel values $[-1, 1]$. The dataset is split into three parts containing 6149, 1020 and 1020 images each. We use the first part to train the unconditional and amortised model. The second part is used for the $h$-transform fine-tuning. The third part is used for evaluation.

For this experiment, we choose both an inpainting and an outpainting task. For the inpainting task a random $18 \times 18$px patch from the image is removed. In difference, for outpainting only a random $18 \times 18$px patch remains and the rest of the image is removed. Thus, the outpainting tasks tests better the generational capabilities of our framework. For the unconditional and amortised model, we use a standard attention U-Net [16] in the discrete DDPM framework. Both the unconditional and the amortised model have about 24M parameters. There is a minor difference due the fact that the unconditional network has 3 input channels and the amortised model has 7 input channels, i.e., the noisy image, the observations and the mask. The $h$-transform is implemented according to the parametrisation in Section 3.3 with an attention U-Net [16] for $\mathrm{NN}_1^{\phi}$. In total, the $h$-transform has about 4M parameters, i.e., about $18\%$ of the size of the amortised model. We evaluate three different settings for the amortised model:

- AMORTISED (20X, FULL DATA): trained on the full training dataset for 1200 epochs,
- AMORTISED (2X): trained on the fine-tuning dataset for 700 epochs,
- AMORTISED (1X): trained on the fine-tuning dataset with the same computational budget as DEFT (300 epochs).

We trained all models on a single GeForce RTX 3090. Training time for AMORTISED (20X, FULL DATA) and the unconditional model was about 50h. The training time for AMORTISED (2X) was 4.5h, while the fine-tuning and AMORTISED (1X) took about 2.5h.

Table 8: Comparing the full amortised training with our conditional fine-tuning objective on the Flowers dataset for inpainting, outpainting and blur.

| | INPAINTING | | | OUTPAINTING | | | BLUR | | |
| | PSNR | SSIM | KID | PSNR | SSIM | KID | PSNR | SSIM | KID |
|---|---|---|---|---|---|---|---|---|---|
| AMORTISED (20X, FULL DATA) | 27.81 | 0.936 | 0.000057 | 12.14 | 0.221 | 0.028 | 23.02 | 0.734 | 0.0196 |
| AMORTISED (2X) | 25.22 | 0.912 | 0.0016 | 10.84 | 0.192 | 0.0821 | 22.66 | 0.716 | 0.0289 |
| AMORTISED (1X) | 16.28 | 0.806 | 0.006 | 9.985 | 0.173 | 0.1 | 21.75 | 0.705 | 0.0374 |
| DPS [12] | 26.29 | 0.897 | 0.0036 | 11.88 | 0.215 | 0.0389 | 22.51 | 0.683 | 0.0624 |
| DEFT | 26.18 | 0.916 | 0.0019 | 11.18 | 0.160 | 0.11 | 23.16 | 0.709 | 0.0529 |

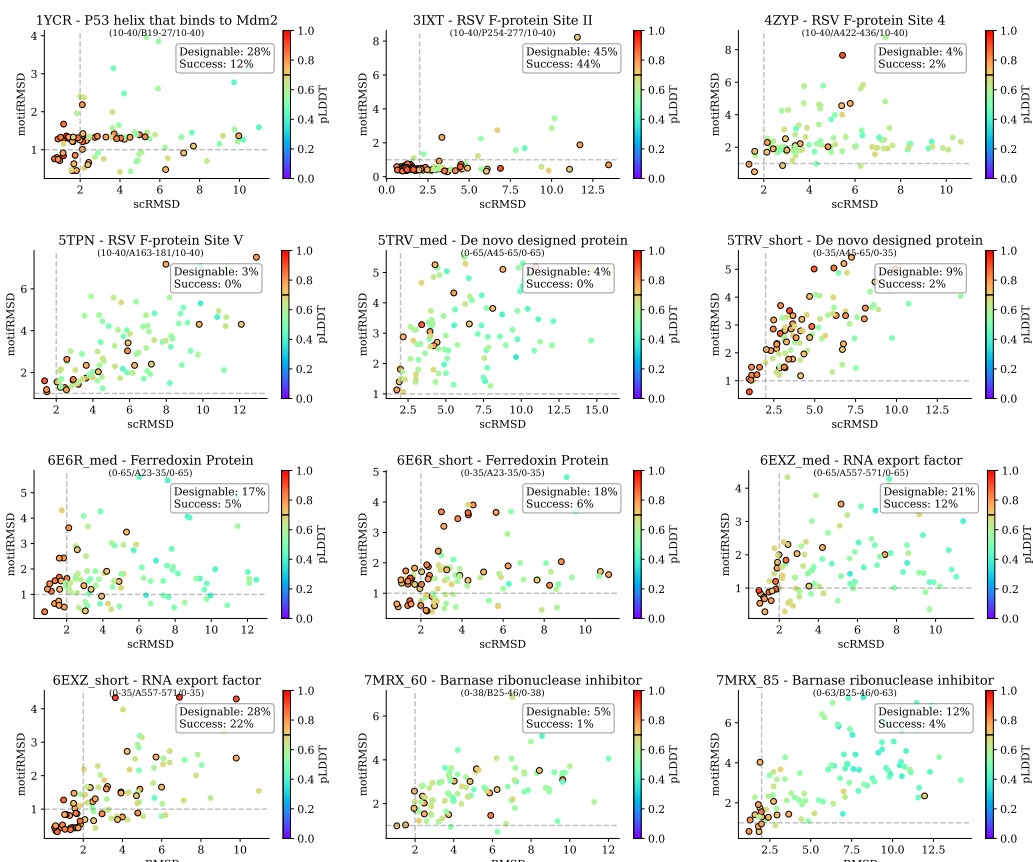

Figure 13: Full results for DEFT (9% model). For each task, we show the full scatter plot of scRMSD and motifRMSD for all 100 samples. The colour indicates the pLDDT confidence score of the re-folded structure with ESMFold. Samples with pLDDT ≥ 0.7 are outlined.

Results are presented in Table 8. With a same computational budget, DEFT outperforms the amortised model on all tasks. Training the amortised model with a larger computational budget, recovers a similar performance to DEFT. Finally, in the scenario of full access to the complete dataset and large computational budget the amortised model is able to outperform both DEFT and DPS.

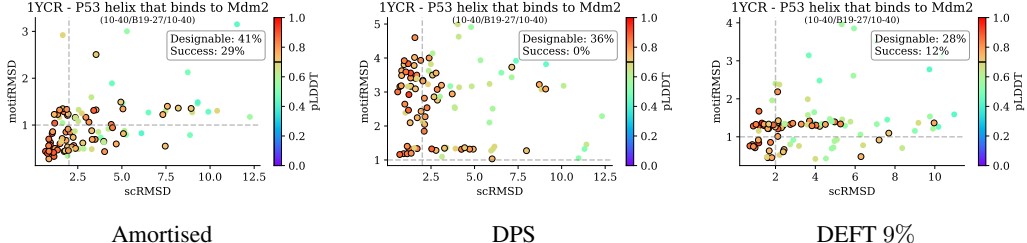

Figure 14: Comparison of the amortised model, DPS and DEFT (9%) on the task 1YCR. We see the general trend for DPS that for low guidance scales the samples have high designability but do not adhere to the motif constraint, while for higher guidance scales they adhere to the motif constraint but have low designability.

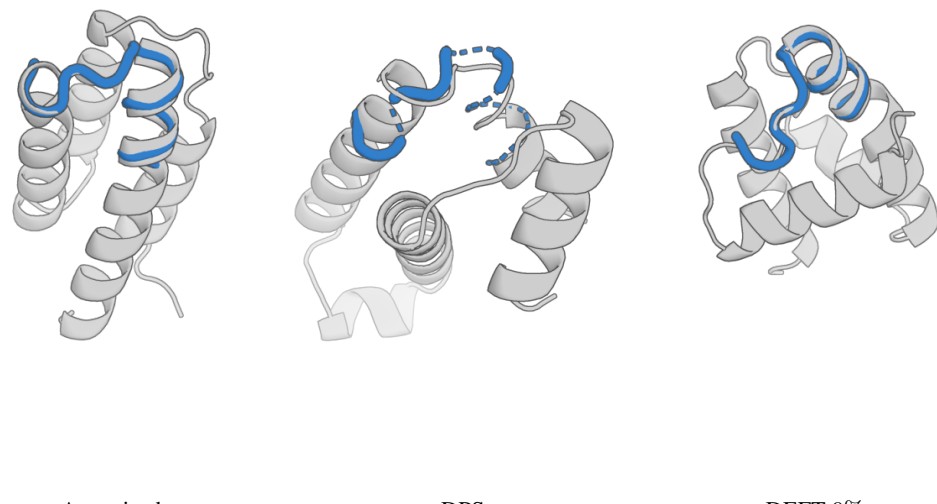

Figure 15: Comparison of samples from the amortised model, DPS and DEFT (9%) on the tasks 6EXZ med. One can see that while the amortised and DEFT samples incorporate the motif into a realistic backbone, this is not the case for DPS. We generally observed that at small guidance scales DPS produced realistic backbones without the desired motif and at high guidance scales it placed the motif into an unrealistic backbone.

# I Algorithms

In this section, we reformulate multiple algorithms from the literature under our common framework as a reference for practitioners. In these algorithms, we use the following conventions: our dataset is drawn from the law $\mathcal{P}_{\text{data}}$, but we can only sample from the simpler law $\mathcal{P}_{\text{sampling}}$ at inference time, which is often chosen as multivariate standard normal $\mathcal{P}_{\text{sampling}} = \mathcal{N}(0, \mathbf{I})$. Therefore, we construct a forward noising process $\mathcal{P}_{\text{data}} \to \mathcal{P}_{\text{sampling}}$ that is parametrised via the noise schedule $\beta_t = \beta(t), \bar{\alpha}_t = \bar{\alpha}(t)$ and try to learn the reverse denoising process $\mathcal{P}_{\text{sampling}} \to \mathcal{P}_{\text{data}}$. Due to this notion of "forward", and to keep consistency with the literature on denoising diffusion models, we explicate the nomenclature $\mathcal{P}_{\text{data}} = \mathcal{P}_0$ and $\mathcal{P}_{\text{sampling}} = \mathcal{P}_T$.

There is an additional law $\mathcal{P}_{\text{noise}}$ that is sometimes confused with $\mathcal{P}_{\text{sampling}}$ since in practice both are often chosen as $\mathcal{N}(0, \mathbf{I})$, but they are two distinct laws that could in principle be different. $\mathcal{P}_{\text{noise}}$ is the law from which the noise added during the forward noising process as well as the during the reverse diffusion process is drawn from.

---

**Algorithm 1** | Unconditional training of denoising diffusion models [28]

---

**Require:** Dataset drawn from law $\mathcal{P}_{\text{data}} = \mathcal{P}_0$            ▷ Dataset law $\mathcal{P}_{\text{data}}$
**Require:** Noise schedule $\beta_t = \beta(t), \bar{\alpha}_t = \bar{\alpha}(t)$, parametrising process $\mathcal{P}_{\text{data}} \to \mathcal{P}_{\text{sampling}}$
**Require:** Untrained noise predictor function $\epsilon_t^{\theta}(\boldsymbol{x})$ with parameters $\theta$
1: **repeat**
2:     $\boldsymbol{x}_0 \sim \mathcal{P}_0 = \mathcal{P}_{\text{data}}$
3:     $t \sim \text{Uniform}(\{1, ..., T\})$

4:     ▷ Forward noise sample, $\boldsymbol{x}_t \sim \bar{p}_{t|0}(\boldsymbol{x}_0)$                               ◁
5:     $\boldsymbol{\varepsilon}_t \sim \mathcal{P}_{\text{noise}}$             ▷ Often Brownian motion, $\mathcal{P}_{\text{noise}} = \mathcal{N}(0, \mathbf{I})$
6:     $\boldsymbol{x}_t \leftarrow \sqrt{\bar{\alpha}_t}\boldsymbol{x}_0 + \sqrt{1 - \bar{\alpha}_t}\boldsymbol{\varepsilon}_t$

7:     ▷ Estimate noise of noised sample                                     ◁
8:     $\hat{\boldsymbol{\varepsilon}}_{\theta} \leftarrow \epsilon_t^{\theta}(\boldsymbol{x}_t)$

9:     Take gradient descent step on
         $\nabla_{\theta} L(\boldsymbol{\varepsilon}_t, \hat{\boldsymbol{\varepsilon}}_{\theta})$         ▷ Typically, loss $L(\boldsymbol{x}_{\text{true}}, \boldsymbol{x}_{\text{pred}}) = ||\boldsymbol{x}_{\text{true}} - \boldsymbol{x}_{\text{pred}}||^2$
10: **until** converged or max epoch reached

---

**Algorithm 2** | Unconditional sampling with denoising diffusion models [28]

---

**Require:** Unconditionally trained noise predictor $\epsilon_t^{\theta}(\boldsymbol{x}_t)$
**Require:** Noise schedule $\beta_t = \beta(t), \bar{\alpha}_t = \bar{\alpha}(t)$, parametrising process $\mathcal{P}_{\text{data}} \to \mathcal{P}_{\text{sampling}}$
1: ▷ Sample a starting point $\boldsymbol{x}_T$                                   ◁
2: $\boldsymbol{x}_T \sim \mathcal{P}_T = \mathcal{P}_{\text{sampling}}$             ▷ Often $\mathcal{P}_T = \mathcal{N}(0, \mathbf{I})$

3: ▷ Iteratively denoise for $T$ steps                               ◁
4: **for** $t$ in $(T, T-1, \ldots, 1)$ **do**
5:     ▷ Predict noise with learned network                       ◁
6:     $\hat{\boldsymbol{\varepsilon}}_{\theta} \leftarrow \epsilon_t^{\theta}(\boldsymbol{x}_t)$

7:     ▷ Denoise sample with learned reverse process $\boldsymbol{x}_{t-1} \sim \bar{p}_{t-1|t}(\boldsymbol{x}_t)$     ◁
8:     ▷ Perform reverse drift                                      ◁
9:     $\boldsymbol{x}_{t-1} \leftarrow \dfrac{1}{\sqrt{1 - \beta_t}}\left(\boldsymbol{x}_t - \dfrac{\beta_t}{\sqrt{1 - \bar{\alpha}_t}}\hat{\boldsymbol{\varepsilon}}_{\theta}\right)$

10:     ▷ Perform reverse diffusion, which is often Brownian motion in $\mathbb{R}^n$, i.e. $\mathcal{P}_{\text{noise}} = \mathcal{N}(0, \mathbf{I})$ ◁
11:     $\boldsymbol{\varepsilon}_t \sim \mathcal{P}_{\text{noise}}$ if $t > 1$ else $\boldsymbol{\varepsilon}_t \leftarrow 0$
12:     $\boldsymbol{x}_{t-1} \leftarrow \boldsymbol{x}_{t-1} + \sigma_t \boldsymbol{\varepsilon}_t$            ▷ A common choice is $\sigma_t = \beta(t)$
13: **return** $\boldsymbol{x}_0$

---

**Algorithm 3** | RFDiffusion conditional training [77]

---

**Require:** Dataset drawn from $\mathcal{P}_{\text{data}}$          ▷ Dataset law $\mathcal{P}_{\text{data}}$
**Require:** Noise schedule $\beta_t = \beta(t), \bar{\alpha}_t = \bar{\alpha}(t)$, parametrising process $\mathcal{P}_{\text{data}} \to \mathcal{P}_{\text{sampling}}$
**Require:** Untrained conditional noise predictor function $\mathbf{f}_\theta(\boldsymbol{x}, t, M)$ with parameters $\theta$

1: **repeat**
2:     $\boldsymbol{x}_0 \sim \mathcal{P}_0 = \mathcal{P}_{\text{data}}$
3:     $t \sim \text{Uniform}(\{1, ..., T\})$
4:     $\boldsymbol{x}_0^{[M]} \cup \boldsymbol{x}_0^{[\backslash M]} \leftarrow \boldsymbol{x}_0$         ▷ Randomly partition data point into motif and rest

5:     ▷ Forward noise the non-motif rest via sampling from $\vec{p}_{0|t}(\boldsymbol{x}_0)$         ◁
6:     $\boldsymbol{\varepsilon}_t \sim \mathcal{P}_{\text{noise}}$
7:     $\boldsymbol{x}_t^{[\backslash M]} \leftarrow \sqrt{\bar{\alpha}_t}\boldsymbol{x}_0^{[\backslash M]} + \sqrt{1 - \bar{\alpha}_t}\boldsymbol{\varepsilon}_t^{[\backslash M]}$

8:     ▷ Combine unnoised motif with noised rest and set timestep of motif part to 0     ◁
9:     $\boldsymbol{x}_t \leftarrow \boldsymbol{x}_0^{[M]} \cup \boldsymbol{x}_t^{[\backslash M]}$
10:    $t^{[M]} \leftarrow 0$
11:    $\hat{\boldsymbol{\varepsilon}}_\theta \leftarrow \mathbf{f}_\theta(\boldsymbol{x}_t, t, M)$         ▷ Estimate noise of sample with noised rest
12:    Take gradient descent step on
        $\nabla_\theta L(\boldsymbol{\varepsilon}, \hat{\boldsymbol{\varepsilon}}_\theta)$         ▷ Typically, $L(x_{\text{true}}, x_{\text{pred}}) = ||x_{\text{true}} - x_{\text{pred}}||^2$
13: **until** converged or max epoch reached

---

**Algorithm 4** | Amortised training – i.e. Doob's $h$-transform conditional training for motif-scaffolding

---

**Require:** Dataset drawn from $\mathcal{P}_{\text{data}}$          ▷ Dataset law $\mathcal{P}_{\text{data}}$
**Require:** Noise schedule $\beta_t = \beta(t), \bar{\alpha}_t = \bar{\alpha}(t)$, parametrising process $\mathcal{P}_{\text{data}} \to \mathcal{P}_{\text{sampling}}$
**Require:** Untrained amortised noise predictor function $\mathbf{f}_\theta(\mathbf{x}, t, \mathbf{x}^{[M]}, M)$ with parameters $\theta$

1: **repeat**
2:     $\mathbf{x}_0 \sim \mathcal{P}_0 = \mathcal{P}_{\text{data}}$
3:     $t \sim \text{Uniform}(\{1, ..., T\})$
4:     $\mathbf{x}_0^{[M]} \cup \mathbf{x}_0^{[\backslash M]} \leftarrow \mathbf{x}_0$         ▷ Randomly partition data point into motif and rest

5:     ▷ Forward noise full sample via sampling from $\vec{p}_{0|t}(\mathbf{x}_0)$         ◁
6:     $\boldsymbol{\varepsilon}_t \sim \mathcal{P}_{\text{noise}}$
7:     $\mathbf{x}_t \leftarrow \sqrt{\bar{\alpha}_t}\mathbf{x}_0 + \sqrt{1 - \bar{\alpha}_t}\boldsymbol{\varepsilon}_t$

8:     ▷ Estimate noise of sample with original motif as additional input         ◁
9:     $\hat{\boldsymbol{\varepsilon}}_\theta \leftarrow \mathbf{f}_\theta(\mathbf{x}_t, t, \mathbf{x}_0^{[M]}, M)$
10:    Take gradient descent step on
        $\nabla_\theta L(\boldsymbol{\varepsilon}, \hat{\boldsymbol{\varepsilon}}_\theta)$         ▷ Typically, $L(x_{\text{true}}, x_{\text{pred}}) = ||x_{\text{true}} - x_{\text{pred}}||^2$
11: **until** converged or max epoch reached

---

---

**Algorithm 5** | $h$-transform fine-tuning (new)

---

**Require:** Dataset drawn from $\mathcal{P}_{\text{data}}$                    ▷ Dataset law $\mathcal{P}_{\text{data}}$
**Require:** Noise schedule $\beta_t = \beta(t), \bar{\alpha}_t = \bar{\alpha}(t)$, parametrising process $\mathcal{P}_{\text{data}} \to \mathcal{P}_{\text{sampling}}$
**Require:** Trained noise predictor function $\epsilon_t^\theta(\boldsymbol{x})$ with parameters $\theta$
**Require:** Untrained $h$-transform $h_t^\phi(\boldsymbol{x}, \hat{\boldsymbol{x}}_0, \boldsymbol{y})$ with parameters $\phi$

 1: **repeat**
 2:      $\boldsymbol{x}_0 \sim \mathcal{P}_0 = \mathcal{P}_{\text{data}}$
 3:      $t \sim \text{Uniform}(\{1, ..., T\})$
 4:      $\boldsymbol{y} \sim p(\boldsymbol{y}|\boldsymbol{x}_0)$                           ▷ Simulate observations
 5:      ▷ Forward noise full sample via sampling from $\vec{p}_{0|t}(\boldsymbol{x}_0)$          ◁
 6:      $\boldsymbol{\varepsilon}_t \sim \mathcal{P}_{\text{noise}}$
 7:      $\boldsymbol{x}_t \leftarrow \sqrt{\bar{\alpha}_t}\boldsymbol{x}_0 + \sqrt{1 - \bar{\alpha}_t}\boldsymbol{\varepsilon}_t$
 8:      $\hat{\boldsymbol{\varepsilon}}_\theta \leftarrow \epsilon_t^\theta(\boldsymbol{x}_t)$            ▷ Estimate noise of sample with pretrained model
 9:      $\hat{\boldsymbol{x}}_0 \leftarrow (\boldsymbol{x}_t - \sqrt{1 - \bar{\alpha}_t}\hat{\boldsymbol{\varepsilon}}_\theta)/\sqrt{\bar{\alpha}_t}$
10:      $\hat{\boldsymbol{\epsilon}}_\phi \leftarrow h_t^\phi(\boldsymbol{x}_t, \hat{\boldsymbol{x}}_0, \boldsymbol{y})$               ▷ Estimate noise of sample with $h$-transform
11:      Take gradient descent step w.r.t. $\phi$ on
          $\nabla_\theta L(\boldsymbol{\varepsilon}, \hat{\boldsymbol{\varepsilon}}_\theta + \hat{\boldsymbol{\varepsilon}}_\phi)$           ▷ Typically, $L(\boldsymbol{x}_{\text{true}}, \boldsymbol{x}_{\text{pred}}) = ||\boldsymbol{x}_{\text{true}} - \boldsymbol{x}_{\text{pred}}||^2$
12: **until** converged or max epoch reached

---

**Algorithm 6** | $h$-transform DDIM sampling (new)

---

**Require:** Trained $h$-transform $h_t^\phi(\boldsymbol{x}, \hat{\boldsymbol{x}}_0, \boldsymbol{y})$ with parameters $\phi$
**Require:** Unconditionally trained noise predictor $\epsilon_t^\theta(\boldsymbol{x}_t)$
**Require:** Noise schedule $\beta_t = \beta(t), \bar{\alpha}_t = \bar{\alpha}(t)$, parametrising process $\mathcal{P}_{\text{data}} \to \mathcal{P}_{\text{sampling}}$
**Require:** Schedule $\sigma_t = \sigma(t)$
**Require:** Observation $\boldsymbol{y}$

 1: ▷ Sample a starting point $\boldsymbol{x}_T$                                     ◁
 2: $\boldsymbol{x}_T \sim \mathcal{P}_T = \mathcal{P}_{\text{sampling}}$                        ▷ Often $\mathcal{P}_T = \mathcal{N}(0, \mathbf{I})$

 3: ▷ Iteratively denoise for $T$ steps                                ◁
 4: **for** $t$ in $(T, T-1, \ldots, 1)$ **do**
 5:      ▷ Predict unconditional noise with learned network            ◁
 6:      $\hat{\boldsymbol{\varepsilon}}_\theta \leftarrow \epsilon_t^\theta(\boldsymbol{x}_t)$
 7:      $\hat{\boldsymbol{x}}_0 \leftarrow \dfrac{\boldsymbol{x}_t - \sqrt{1 - \bar{\alpha}_t}\hat{\boldsymbol{\varepsilon}}_\theta}{\sqrt{\bar{\alpha}_t}}$
 8:      $\hat{\boldsymbol{\epsilon}}_\phi \leftarrow h_t^\phi(\boldsymbol{x}_t, \hat{\boldsymbol{x}}_0, \boldsymbol{y})$
 9:      ▷ Estimate posterior noise                                  ◁
10:      $\hat{\epsilon} \leftarrow \hat{\boldsymbol{\varepsilon}}_\theta + \hat{\epsilon}_\phi$
11:      $\boldsymbol{\varepsilon}_t \sim \mathcal{P}_{\text{noise}}$ if $t > 1$ else $\boldsymbol{\varepsilon}_t \leftarrow 0$
12:      $\boldsymbol{x}_{t-1} \leftarrow \sqrt{\bar{\alpha}_{t-1}}\left(\dfrac{\boldsymbol{x}_t - \sqrt{1 - \bar{\alpha}_t}\hat{\epsilon}}{\sqrt{\bar{\alpha}_t}}\right) + \sqrt{1 - \bar{\alpha}_{t-1} - \sigma_t^2}\hat{\epsilon} + \sigma_t \boldsymbol{\varepsilon}_t$
13: **return** $\boldsymbol{x}_0$

---

**Algorithm 7** | RFDiffusion conditional sampling [77]

---

**Require:** Conditionally trained noise predictor $\mathbf{f}_\theta(\mathbf{x}, t, M)$
**Require:** Target motif/context $\mathbf{x}_0^{[M]}$
**Require:** Noise schedule $\beta_t = \beta(t), \bar{\alpha}_t = \bar{\alpha}(t)$, parametrising process $\mathcal{P}_{\text{data}} \to \mathcal{P}_{\text{sampling}}$

  1:  ▷ Sample a starting point $\mathbf{x}_T$       ◁
  2:  $\mathbf{x}_T \sim \mathcal{P}_T = \mathcal{P}_{\text{sampling}}$

  3:  ▷ Iteratively denoise for $T$ steps       ▷ Often $\mathcal{P}_T = \mathcal{N}(0, \mathbf{I})$ ◁
  4:  **for** $t$ in $(T, T-1, \ldots, 1)$ **do**
  5:     ▷ Overwrite motif variables with target motif and reset their time parameter     ◁
  6:     ▷ Note: Original RFDiffusion zero-centers $\mathbf{x}_t$ and $\mathbf{x}_0^{[M]}$ individually for equivariance.     ◁
  7:     $\mathbf{x}_t^{[M]} \leftarrow \mathbf{x}_0^{[M]}$       ▷ Set noisy motif to unnoised motif
  8:     $t^{[M]} \leftarrow 0$       ▷ Set timesteps for motif to 0
  9:     $\hat{\varepsilon}_\theta = \mathbf{f}_\theta(\mathbf{x}_t, t, M)$       ▷ Predict noise with learned network

10:     ▷ Denoise sample with learned reverse process $\mathbf{x}_{t-1} \sim \bar{p}_{t-1|t}(\mathbf{x}_t)$     ◁
11:     ▷ Perform reverse drift     ◁
12:     $\mathbf{x}_{t-1} \leftarrow \dfrac{1}{\sqrt{1-\beta_t}}\left(\mathbf{x}_t - \dfrac{\beta_t}{\sqrt{1-\bar{\alpha}_t}}\hat{\varepsilon}_\theta\right)$

13:     ▷ Perform reverse diffusion, which is often Brownian motion in $\mathbb{R}^n$, i.e. $\mathcal{P}_{\text{noise}} = \mathcal{N}(0, \mathbf{I})$ ◁
14:     $\varepsilon_t \sim \mathcal{P}_{\text{noise}}$ if $t > 1$ else $\varepsilon_t \leftarrow 0$
15:     $\mathbf{x}_{t-1} \leftarrow \mathbf{x}_{t-1} + \sigma_t \varepsilon_t$       ▷ A common choice is $\sigma_t = \beta(t)$
16: **return** $\mathbf{x}_0$

---

**Algorithm 8** | Replacement conditional sampling for motif-scaffolding

---

**Require:** Unconditionally trained noise predictor $\epsilon_t^\theta(\mathbf{x}_t)$
**Require:** Noise schedule $\beta_t = \beta(t), \bar{\alpha}_t = \bar{\alpha}(t)$, parametrising process $\mathcal{P}_{\text{data}} \to \mathcal{P}_{\text{sampling}}$
**Require:** Target motif $\mathbf{x}_0^{[M]}$

  1:  ▷ Sample a starting point $\mathbf{x}_T$     ◁
  2:  $\mathbf{x}_T \sim \mathcal{P}_T = \mathcal{P}_{\text{sampling}}$

  3:  ▷ Iteratively denoise for $T$ steps       ▷ Often $\mathcal{P}_T = \mathcal{N}(0, \mathbf{I})$ ◁
  4:  **for** $t$ in $(T, T-1, \ldots, 1)$ **do**
  5:     ▷ Predict noise with learned network     ◁
  6:     $\hat{\varepsilon}_\theta \leftarrow \epsilon_t^\theta(\mathbf{x}_t)$

  7:     ▷ Denoise sample with learned reverse process $\mathbf{x}_{t-1} \sim \bar{p}_{t-1|t}(\mathbf{x}_t)$     ◁
  8:     ▷ Perform reverse drift     ◁
  9:     $\mathbf{x}_{t-1} \leftarrow \dfrac{1}{\sqrt{1-\beta_t}}\left(\mathbf{x}_t - \dfrac{\beta_t}{\sqrt{1-\bar{\alpha}_t}}\hat{\varepsilon}_\theta\right)$

10:     ▷ Perform reverse diffusion, which is often Brownian motion in $\mathbb{R}^n$, i.e. $\mathcal{P}_{\text{noise}} = \mathcal{N}(0, \mathbf{I})$ ◁
11:     $\varepsilon_t \sim \mathcal{P}_{\text{noise}}$ if $t > 1$ else $\varepsilon_t \leftarrow 0$
12:     $\mathbf{x}_{t-1} \leftarrow \mathbf{x}_{t-1} + \sigma_t \varepsilon_t$       ▷ A common choice is $\sigma_t = \beta(t)$

13:     ▷ Forward noise the target motif $\mathbf{x}_{t-1}^{[M]} \sim \vec{p}_{0|t-1}(\mathbf{x}_0^{[M]})$     ◁
14:     $\boldsymbol{\eta}_{t-1} \sim \mathcal{P}_{\text{noise}}$ if $t > 1$ else $\boldsymbol{\eta}_{t-1} \leftarrow 0$
15:     $\mathbf{x}_{t-1}^{[M]} \leftarrow \sqrt{\bar{\alpha}_{t-1}}\mathbf{x}_0^{[M]} + \sqrt{1-\bar{\alpha}_{t-1}}\boldsymbol{\eta}_{t-1}$
16:     $\mathbf{x}_{t-1} \leftarrow \mathbf{x}_{t-1}^{[\backslash M]} \cup \mathbf{x}_{t-1}^{[M]}$       ▷ Insert noised motif into current sample
17: **return** $\mathbf{x}_0$

---

---

**Algorithm 9** | Reconstruction Guidance (i.e. Moment Matching (MM) Approximation to $h$-transform, DPS [12]) for general inverse problems $\boldsymbol{y} \sim \text{noise}(\mathcal{A}(\boldsymbol{x}))$

---

**Require:** Unconditionally trained noise predictor $\epsilon_t^\theta(\boldsymbol{x}_t)$, observation $\boldsymbol{y}$.
**Require:** Noise schedule $\beta_t = \beta(t), \bar{\alpha}_t = \bar{\alpha}(t)$, parameterising process $\mathcal{P}_{\text{data}} \to \mathcal{P}_{\text{sampling}}$
**Require:** Guidance scale (schedule) $\gamma_t = \gamma(t)$
**Require:** Conditioning loss $l(\boldsymbol{y}_{\text{pred}}, \boldsymbol{y})$. e.g, Gaussian MM $l(\boldsymbol{y}_{\text{pred}}, \boldsymbol{y}) = ||\boldsymbol{y}_{\text{pred}} - \boldsymbol{y}||^2$

1: ▷ Sample a starting point $\boldsymbol{x}_T$ ◁
2: $\boldsymbol{x}_T \sim \mathcal{P}_T = \mathcal{P}_{\text{sampling}}$ ▷ Often $\mathcal{P}_T = \mathcal{N}(0, \mathbf{I})$

3: ▷ Iteratively denoise and condition for $T$ steps ◁
4: **for** $t$ in $(T, T-1, \dots, 1)$ **do**
5: $\quad$ $\hat{\boldsymbol{\varepsilon}}_\theta \leftarrow \epsilon_t^\theta(\boldsymbol{x}_t)$ ▷ Predict noise with learned network

6: $\quad$ ▷ Estimate current denoised estimate via Tweedie's formula ◁
7: $\quad$ $\hat{\boldsymbol{x}}_0(\boldsymbol{x}_t, \hat{\boldsymbol{\varepsilon}}_\theta) \leftarrow \frac{1}{\sqrt{\bar{\alpha}_t}}(\boldsymbol{x}_t - \sqrt{1 - \bar{\alpha}_t}\hat{\boldsymbol{\varepsilon}}_\theta)$ ▷ c.f. also eq. 15 in [28]

8: $\quad$ ▷ Perform gradient descent step towards data consistency ◁
9: $\quad$ $\boldsymbol{x}_t \leftarrow \boldsymbol{x}_t - \gamma_t \nabla_x l(\mathcal{A}(\hat{\boldsymbol{x}}_0), \boldsymbol{y})$ ▷ Requires backprop through $\epsilon_t^\theta$ via e.g. $L_2$ loss

10: $\quad$ ▷ Denoise sample with learned reverse process $\boldsymbol{x}_{t-1} \sim \bar{p}_{t-1|t}(\boldsymbol{x}_t)$ ◁
11: $\quad$ $\boldsymbol{x}_{t-1} \leftarrow (1-\beta_t)^{-1/2}\left(\boldsymbol{x}_t - \beta_t(1-\bar{\alpha}_t)^{-1/2}\hat{\boldsymbol{\varepsilon}}_\theta\right)$ ▷ Perform reverse drift
12: $\quad$ ▷ Perform reverse diffusion, which is often Brownian motion in $\mathbb{R}^n$, i.e. $\mathcal{P}_{\text{noise}} = \mathcal{N}(0, \mathbf{I})$ ◁
13: $\quad$ $\boldsymbol{\varepsilon}_t \sim \mathcal{P}_{\text{noise}}$ if $t > 1$ else $\boldsymbol{\varepsilon}_t \leftarrow 0$
14: $\quad$ $\boldsymbol{x}_{t-1} \leftarrow \boldsymbol{x}_{t-1} + \sigma_t \boldsymbol{\varepsilon}_t$ ▷ A common choice is $\sigma_t = \beta(t)$
15: **return** $\boldsymbol{x}_0$

---

**Algorithm 10** | Reconstruction Guidance (i.e. Moment Matching (MM) Approximation to $h$-transform, DPS [12]) for motif scaffolding

---

**Require:** Unconditionally trained noise predictor $\epsilon_t^\theta(\boldsymbol{x}_t)$, target motif/context $\boldsymbol{x}_0^{[M]}$.
**Require:** Noise schedule $\beta_t = \beta(t), \bar{\alpha}_t = \bar{\alpha}(t)$, parameterising process $\mathcal{P}_{\text{data}} \to \mathcal{P}_{\text{sampling}}$
**Require:** Guidance scale (schedule) $\gamma_t = \gamma(t)$
**Require:** Conditioning loss $l(\boldsymbol{x}_{\text{true}}, \boldsymbol{x}_{\text{pred}})$. e.g, Gaussian MM $l(\boldsymbol{x}_{\text{true}}, \boldsymbol{x}_{\text{pred}}) = ||\boldsymbol{x}_{\text{true}} - \boldsymbol{x}_{\text{pred}}||^2$

1: ▷ Sample a starting point $\boldsymbol{x}_T$ ◁
2: $\boldsymbol{x}_T \sim \mathcal{P}_T = \mathcal{P}_{\text{sampling}}$ ▷ Often $\mathcal{P}_T = \mathcal{N}(0, \mathbf{I})$

3: ▷ Iteratively denoise and condition for $T$ steps ◁
4: **for** $t$ in $(T, T-1, \dots, 1)$ **do**
5: $\quad$ $\hat{\boldsymbol{\varepsilon}}_\theta \leftarrow \epsilon_t^\theta(\boldsymbol{x}_t)$ ▷ Predict noise with learned network

6: $\quad$ ▷ Estimate current denoised estimate via Tweedie's formula ◁
7: $\quad$ $\hat{\boldsymbol{x}}_0(\boldsymbol{x}_t, \hat{\boldsymbol{\varepsilon}}_\theta) \leftarrow \frac{1}{\sqrt{\bar{\alpha}_t}}(\boldsymbol{x}_t - \sqrt{1 - \bar{\alpha}_t}\hat{\boldsymbol{\varepsilon}}_\theta)$ ▷ c.f. also eq. 15 in [28]

8: $\quad$ ▷ Perform gradient descent step towards condition on motif dimensions $M$ ◁
9: $\quad$ $\boldsymbol{x}_t \leftarrow \boldsymbol{x}_t - \gamma_t \nabla_x l(\boldsymbol{x}_0^{[M]}, \hat{\boldsymbol{x}}_0^{[M]}(\boldsymbol{x}_t, \hat{\boldsymbol{\varepsilon}}_\theta))$ ▷ Requires backprop through $\epsilon_t^\theta$ via e.g. $L_2$ loss

10: $\quad$ ▷ Denoise sample with learned reverse process $\boldsymbol{x}_{t-1} \sim \bar{p}_{t-1|t}(\boldsymbol{x}_t)$ ◁
11: $\quad$ $\boldsymbol{x}_{t-1} \leftarrow (1-\beta_t)^{-1/2}\left(\boldsymbol{x}_t - \beta_t(1-\bar{\alpha}_t)^{-1/2}\hat{\boldsymbol{\varepsilon}}_\theta\right)$ ▷ Perform reverse drift
12: $\quad$ ▷ Perform reverse diffusion, which is often Brownian motion in $\mathbb{R}^n$, i.e. $\mathcal{P}_{\text{noise}} = \mathcal{N}(0, \mathbf{I})$ ◁
13: $\quad$ $\boldsymbol{\varepsilon}_t \sim \mathcal{P}_{\text{noise}}$ if $t > 1$ else $\boldsymbol{\varepsilon}_t \leftarrow 0$
14: $\quad$ $\boldsymbol{x}_{t-1} \leftarrow \boldsymbol{x}_{t-1} + \sigma_t \boldsymbol{\varepsilon}_t$ ▷ A common choice is $\sigma_t = \beta(t)$
15: **return** $\boldsymbol{x}_0$

