# OpenReview forum: "DEFT: Efficient Fine-tuning of Diffusion Models by Learning the Generalised $h$-transform"
_NeurIPS.cc/2024/Conference — NeurIPS 2024 poster_

### Official Review · Reviewer_ZJuc · 2024-07-12

**Soundness:** 3
**Presentation:** 3
**Contribution:** 2
**Rating:** 5
**Confidence:** 3

**Summary:**

The authors propose a method for conditional generation using diffusion models, named DEFT. The idea is to combine a fixed, pre-trained unconditional model with an additionally learned conditional correction term to generate conditionally. The authors provide extensive experiments to demonstrate the effectiveness of DEFT.

**Strengths:**

- The paper is well-written and easy to follow.
- The proposed method is simple, easy to understand, and should be easy to implement, making it potentially very useful for practical applications. Additionally, DEFT only requires the inference of a pretrained model, eliminating the need for fine-tuning or differentiation.
- The authors provide theoretical motivation for their method through the Doob h-transform.
- The authors conduct extensive experiments across different domains and provide an ablation study.
- The code is provided.

**Weaknesses:**

- Compared to some other approaches like DPS, DEFT requires additional training.
- Some image-to-image generative models (such as I2SB and DDBM, which the authors discuss in the appendix) demonstrate great performance in conditional generation. Unlike DEFT, they require fully training a generative model. Despite this difference, for completeness of comparison, it would be good to include both: resulting performance and training budget for these models and DEFT.

**Questions:**

Honestly speaking, I have only one question that may affect my decision. Do you claim to be the first to come up with the idea of learning a small conditional corrector to make an unconditional model conditional?

I’m not familiar with all relevant research on conditional generation. However, if so, I believe your method might be very useful for others, and the paper must be accepted. If not, I would like to see a detailed discussion of DEFT’s contributions compared to prior works.

**Limitations:**

The authors have adequately addressed the limitations.

---

> ### Author Rebuttal · Authors · 2024-08-06
>
> We thank the reviewer for the valuable feedback, we will incorporate the following discussion in a revised version of the manuscript.
>
> ### Q1: Do you claim to be the first to come up with the idea of learning a small conditional corrector to make an unconditional model conditional?
> **A1**: Thank you for your question. We appreciate the opportunity to clarify the novelty of our work.
> The idea of learning a conditional corrector to make an unconditional model conditional has indeed been explored in earlier works. In [1] Dhariwal and Nichol, propose classifier guidance and utilize a pre-trained classifier to enable conditional sampling of an unconditional diffusion model. In a different work, ControlNet [3] proposes learning a conditional corrector based on an additional dataset for text-to-image diffusion models. However, our approach introduces several novel strategies and a mathematical foundation for these fine-tuning approaches.
> As far as we know, DEFT is the first fine-tuning approach that learns a purely **additive** corrector ($\epsilon_\text{new} = \epsilon_\text{old} + \epsilon_\text{corrector}$), allowing for a small  (in model size) corrector term applicable to a wide range of inverse problems. This includes scenarios where measurements $y$ are either real-valued or discrete (whilst [1] is strictly limited to the discrete setting).
> Further, DEFT is **model agnostic** and makes no assumptions on the specific implementation of the pre-trained unconditional diffusion model. This is in contrast to ControlNet, which employs a non-linear composition of the pre-trained and fine-tuned network, which requires knowledge (and access) of the underlying architecture. In DEFT, we only need the availability of evaluating the pre-trained unconditional diffusion models, making our approach applicable even in cases where the pre-trained model is hidden behind an API. Further, DEFT **only adds 3-10% of additional parameters**, making it more parameter-efficient and making it possible to store multiple fine-tuning networks for different tasks. In contrast ControlNet adds about 30-40% additional parameters.
> ### Q2: DEFT requires additional training
> **A2**: We acknowledge that DEFT requires additional training, which in turn requires an additional dataset and an additional training budget. This is a common characteristic of many fine-tuning methods. However, as we discussed in the overall response we see that even when training with 100 or 200 images, we can get quite good results for inpainting on ImageNet. Further, after the initial training phase, sampling is more efficient compared to approaches such as DPS or RED-diff. Please also see our discussion in the overall response.
>
> ### Q3: Comparison against conditional trained image-to-image generative models
> **A3**: We were able to compare DEFT against I2SB [3]. Here, we used the pre-trained checkpoint on their github for `sr4x-pool` and `inpaint-freeform2030`, which directly correspond to our super-resolution and inpainting settings on ImageNet. The results can be found in the 1-page PDF. On inpainting, I2SB outperforms DEFT on all metrics, for example top-1 71.7% vs 74.5% or PSNR 22.18dB vs 23.26dB.
>
> In the following tables we present the results for inpainting and super-resolution on ImageNet. We see that the results are comparable even though I2SB was trained on the complete ImageNet training set. On super-resolution we are even able to outperform I2SB on PSNR and SSIM.
>
> | Inpainting | PSNR (↑)  | SSIM (↑) | LPIPS (↓) | top-1 (↑) | KID (↓)  |
> |------------|-------|------|-------|-------|-------|
> | DEFT   	| 22.18 | 0.85 | 0.09  | 71.7  | 0.29  |
> | I2SB   	| 23.26 | 0.86 | 0.068 | 74.5  | 0.238 |
>
> | Super-resolution | PSNR (↑) | SSIM (↑) | LPIPS (↓) | top-1 (↑) | KID (↓)   |
> |------------------|-------|------|-------|-------|-------|
> | DEFT         	| 24.92 | 0.71 | 0.12  | 71.9  | 1.78  |
> | I2SB         	| 23.95 | 0.64 | 0.11  | 71.6  | 0.004 |
>
>
>
> The conditional trained generative models can be seen as an upper limit on the image quality of fine-tuned models, as they are generally trained on a large dataset. For example, the I2SB models were trained on 1M gradient steps on the full ImageNet training dataset. While DEFT was only trained on a subset of 1000 images.
> In addition, we present a comparison with conditional diffusion models in Appendix H.2 (see Table 7) using the Flowers dataset. Our results show that, with identical training budgets and dataset sizes, DEFT surpasses the conditional model [4] on several image restoration tasks. However, when provided with a substantially larger dataset (6 times the size) and a significantly greater compute budget (20 times more), the conditional diffusion model outperforms DEFT. Note that in many situations (such as medical imaging) the available fine-tuning dataset may be too small to effectively train a conditional diffusion model from scratch.
>
> [1] Dhariwal and Nichol, Diffusion models beat gans on image synthesis, Neurips 2021.
>
> [2] Zhang et al., Adding conditional control to text-to-image diffusion models, IEEE CVPR 2023.
>
> [3] Liu et al., I2SB: Image-to-Image Schrödinger Bridge, ICML 2023.
>
> [4] Batzolis et. al, Conditional Image Generation with Score-Based Diffusion Models, arXiv preprint 2021.

---

> > ### Comment · Reviewer_ZJuc · 2024-08-09
> >
> > I thank the authors for their comprehensive clarifications and additional comparisons. I will maintain my score. My final recommendation will depend on the entire discussion.

---

> > > ### Author Response · Authors · 2024-08-11
> > > **Thank you for your reponse !**
> > >
> > > We thank the reviewer for their prompt follow-up and helpful discussion.
> > >
> > > To aid the discussion further we would like to highlight that have conducted a more extensive literature survey, and we can conclude that our method is uniquely novel in its scalability and design of the conditional model, which is very different from existing techniques that suffer a litany of issues.
> > >
> > > In short to clarify and answer the reviewers' question we would like to highlight that:
> > > ```
> > > We believe our approach is the first method that enables learning a small corrector network for real valued inverse problems.
> > > ```
> > >
> > > Due to our corrector’s additive nature and inductive biases, it allows for parametrising **very small and efficient correctors**, unlike approaches such as ControlNet, which do not offer a significant enough parameter reduction. For example, we are able to achieve much better performance with as little as 3-10% of the original model’s size compared to ControlNet’s 40%.
> > > We would also like to point out that prior well-known approaches such as classifier guidance _do not apply_ when considering real valued inverse problems, which is the main task we tackle in the formulation we propose.
> > >
> > > We hope this shows how our approach is valuable to the community and that it is the first highly efficient learned corrector for general inverse problems, also highlighted by reviewer **no8y** `I believe that this work could have significant impact`.
> > > As before we thank the reviewer for their input and please let us know if there are any other questions which can help aid the dicussion.
> > >
> > > P.S. We have added all the additional experiments requested by the reviewer. Let us know if there are any additional comparisons we can perform in order to strengthen the presentation of the paper for the reviewer’s consideration.

---

### Official Review · Reviewer_ESdx · 2024-07-12

**Soundness:** 3
**Presentation:** 3
**Contribution:** 3
**Rating:** 5
**Confidence:** 4

**Summary:**

The authors propose a novel conditional diffusion sampling strategy for solving inverse problems. Previous conditional diffusion-based inverse solvers are heuristically motivated, lack a unifying framework, and suffer from sensitivity to hyperparameters and heave computation of the Jacobian of the trained score network. The proposed method involves fine-tuning of a small network to quickly learn the conditional $h$-transform, enabling conditional sampling without altering the large unconditional network. The authors demonstrated the efficiency of their method by achieving SOTA performance across various benchmarks with faster inference times.

**Strengths:**

### Theoretical Foundation
This paper is built on solid mathematical theory, specifically “Doob’s $h$-transform,” which enhances our understanding to diffusion models and paves the way for future improvements in their applications.

### Comprehensive Experiments
The authors conducted extensive experiments on their method, covering both linear and nonlinear inverse problems across domains such as natural images, medical images, and conditional protein design. They provide clear evaluation metrics and inference time, supporting their claims about the method’s efficiency.

Furthermore, the network architecture is derived from theoretical principles, enabling systematic improvements. Additionally, they propose an extra loss function for fine-tuning that is applicable without paired data, potentially inspiring future research.

**Weaknesses:**

The proposed claims are poorly presented or difficult to understand. It would be beneficial to supplement the contents in the paper by addressing the following questions.
1. How does Doob’s h transform unify existing methods for diffusion-based inverse solvers?
2. What is the difference between $X_t$ and $H_t$?
3. I do not understand the rationale behind the DEFT network parametrization in Section 3.3. How is equation (13) derived from Doob’s $h$-transform?
4. In Line 281, it is stated that “DEFT assumes no knowledge of the forward operator.” How is this possible? To my understanding, the forward operator is needed to compute $\ln p(y|X_t)$.

Errata bold 5.2 for Time(hrs) of DEFT for super-resolution

**Questions:**

See the weaknesses part

**Limitations:**

Nothing to mention

---

> ### Author Rebuttal · Authors · 2024-08-06
>
> Thank you for this valuable feedback on the presentation - this is very helpful for clarifying our work. We have made updates to the manuscript to address the points you suggest for clarification. The error in Table 1 will be fixed in the camera-ready version. In detail:
> ### Q1: How does Doob’s h transform unify existing methods for diffusion-based inverse solvers?
> A1: Doob’s h-transform is the formal approach to conditioning an SDE and well established in SDE literature, see for example [1]. Yet, Doob’s transform has not been discussed in or been connected to the conditional generative modeling literature. By spelling out the mathematics explicitly, we can identify previous work as special case approximations to the Doob’s h-transform term, thereby providing an underlying framework in conditional generative modeling. Specifically, we derive previous methods as Doob’s h-transform approximations: 1. Reconstruction Guidance (DPS, FreeDoM [3], MPGD [4] etc), 2. CDE (amortized, conditional training) and 3. Classifier Guidance [2].
> ### Q2: What is the difference between $X_t$ and $H_t$?
> A2: We denote with $X_t$ the unconditional diffusion process and with $H_t$ the conditional diffusion process. We will make sure to clarify this in the manuscript when $H_t$ is first introduced.
> ### Q3: How is equation (13) derived from Doob’s ℎ-transform?
> A3: The network architecture in equation (13) defines an inductive bias for DEFT. As discussed, we can derive reconstruction guidance (i.e., DPS) as a special case of the h-transform (see Equations 30-31 Appendix C3 of the manuscript). As the conditional update in DPS has a high computational cost, we omit the Jacobian of the unconditional diffusion model. This then serves as an inductive bias for the architecture of DEFT and is an important ingredient for the approach. We will make this clearer in the camera-ready, ensuring we talk more about the intuition behind using the specific network architecture we recommend.
> Please also refer to Table 5 in the appendix. In this ablation we test a version of DEFT without this inductive bias, leading to a worse PSNR/SSIM for the low-dose computed tomography experiments, showing that this inductive bias improves performance.
> ### Q4: In Line 281, it is stated that “DEFT assumes no knowledge of the forward operator.” How is this possible?
> Thank you for pointing out this imprecision on our part. Indeed, the forward operator is necessary to compute $p(y|x_t)$ and also to use the parameterization in Eqn. (13). What we meant to emphasize was that the training objective in Eqn. (8) does not require the likelihood.  Thus, all we require for the DEFT objective is to be able to evaluate the forward operator such that we can obtain the measurement $y$ or alternatively, access to a paired data set of images and corrupted measurements. For the DEFT architecture in Eqn. (13), we use $\nabla_{x_0} || y- A(\hat{x}_0(x_t)) ||^2$ in most cases including cases where there's not necessarily an explicit likelihood available. This is again an inductive bias that helps guide our network in early iterations but does not require an explicit form for $p(y|x_0)$, although it does assume the forward operator can be differentiated. In cases where it can’t, one can resort to approaches such as ΠGDM [5]. To summarize, the DEFT objective **does not require explicit assumptions of the forward operator**, other than some very weak / regular assumptions for the existence of the score. We have adjusted the manuscript to reflect this.
> Also, here we want to point to the ablation in Table 5 in the Appendix, where we show that the architecture defined in Eqn. (13) leads to a boost in performance. However, without this inductive bias, DEFT can still work.
>
> [1] Särkkä and Solin, Applied Stochastic Differential Equation, Cambridge University Press. (2019)
>
> [2]  Dhariwal and Nichol, Diffusion models beat gans on image synthesis, Neurips 2021.
>
> [3] Yu et. al., Freedom: Training-free energy-guided conditional diffusion model. IEEE CVPR 2023
>
> [4] He et al., Manifold preserving guided diffusion. ICLR 2024
>
> [5] Song et al., Pseudoinverse-Guided Diffusion Models for Inverse Problems, ICLR 2023.

---

> > ### Comment · Reviewer_ESdx · 2024-08-13
> >
> > I find some aspects of the DEFT network parameterization, including the rationale behind it and the inductive bias, to be unclear. However, the strengthened experimental results demonstrate significant value in this work. Therefore, I remain positive about the acceptance of this paper and will keep my score unchanged.

---

> ### Author Response · Authors · 2024-08-13
> **Derivation of the DEFT architecture and inductive bias. Part I**
>
> Thank you for your response, please allow us to clarify the inductive bias and the motivation behind the DEFT network parameterization.
>
> We will provide a **short derivation** as well as **empirical evidence** motivating the architectural choice, we hope that this clarifies the reviewers concern and if not we would like to engage further and understand what is missing to make this more clear.
>
> In conditional sampling methods, the conditional score $\nabla \ln p_t(x_t | y)$ can be decomposed into the unconditional score, approximated with an unconditional score model $s_\theta(x_t, t) \approx \nabla_{x_t} \ln p_t(x_t)$, and a likelihood term, i.e.,
> 	$$\nabla_{x_t} \ln p_t(x_t | y) = s_\theta(x_t, t) +  \nabla_{x_t} \ln p_t(y | x_t) $$
>
> Here, the likelihood term $\nabla_{x_t} \ln p_t(y | x_t)$  is the h-transform that we aim to learn with a neural network in this work. We can express this term as an expectation of the inverse problem likelihood with respect to the denoiser (as done in works such as DPS [1]):
> $$ \nabla_{x_t} \ln p_t(y | x_t) = \nabla_{x_t} \ln  \mathbb{E}_{x_0 \sim p(x_t |x_0)}[p(y| x_0)] $$
>
> As a next step, we can then make a MAP styled approximation to the posterior (more precisely, approximating a posterior with a mean point mass rather than MAP is known as a “Bayes Point Machine” [2]) i.e. $p(x_t |x_0) \approx \delta_{\mathbb{E}[x_0|x_t]}(x_t)$. This results in the following:
>
> $$ \nabla_{x_t} \ln p_t(y | x_t) \approx \nabla_{x_t} \ln p(y|\mathbb{E}[x_0|x_t])  \approx \nabla_{x_t} \ln p(y| \hat{x}_0(x_t))$$
>
> Where $\mathbb{E}[x_0|x_t]$ can be estimated with Tweedies formula and the learned approximate score (i.e. $\hat{x}_0(x_t)$).
>
> If we now initialise the h-transform neural network with $\nabla_{x_t} \ln p(y| \hat{x_0})$, this is clearly a much better starting point than initialising it at $0$, as this term is an approximation of the h-transform and has been validated to perform well in these tasks. However, this approximate expression is prohibitive to train with as it requires the Jacobian $\partial_{x_t}\hat{x}_0(x_t)$ which backpropagates through the score network.
>
> To mitigate this, we follow works such as DreamFusion [3] which take the gradient with respect to $\hat{x}_0(x_t)$ rather than $x_t$. This step is completely heuristic, but it has been validated empirically. After this, we are left with:
>
> $$ \nabla_{x_t} \ln p_t(y | x_t) \approx \nabla_{\hat{x}_0} \ln p(y|\hat{x}_0) $$
>
> As we have already motivated conceptually, this expression approximates the h-transform and thus it makes sense to incorporate this in our h-transform architecture as it provides a good warm start (also note in the MCMC/sampling community these style of gradient aided NN architectures have already demonstrated a lot of succes [4]):
>
> $$\text{NN}(x_t, y, t) = \text{NN2}(x_t, y, t) + \text{NN1}(t) \nabla_{\hat{x}_0} \ln p(y | \hat{x}_0) $$
>
> Where prior to training, NN2 is initiatilised to $0$ and NN1 is initialised to 1 **such that at epoch 0  our network is initialised at this cheap approximate h-transform** $\text{NN}(x_t, y, t) =  \nabla_{\hat{x}_0} \ln p(y | \hat{x}_0) $.
>
> Empirically we found that this initialisation gave much lower starting DEFT losses than without and non surprisingly it lead to converging faster as well as better results.
>
> In a way, the architecture $ \text{NN2}(x_t, y, t) + \text{NN1}(t) \nabla_{\hat{x}_0} \ln p(y | \hat{x}_0) $ achieves the best of both worlds: cheap like conditional sampling methods, accurate like conditional training methods. It starts off with the cheap guidance term in early epochs, thereby providing a good warm start to our objective. But in later epochs, it is able to learn a more accurate approximation (without MAP-like approximations or heuristics) to the h-transform with the term $ \text{NN2}(x_t, y, t) $. Note the term $\text{NN1}(t)$ is there to serve as a “guidance scale like” term. In methods like DPS [1] these terms are typically tuned on a small dataset. In contrast, we believe that the term $\text{NN1}(t)$ is particularly helpful in early iterations as it allows the guidance term to quickly become well tuned, whilst the NN2 network uses this warm start to more slowly learn the full h-transform. For empirical evidence of this hypothesis, see Figure 6 where NN1(t) is plotted and learns very expected guidance scales that increase along the diffusion trajectory as expected.

---

> ### Author Response · Authors · 2024-08-13
> **Derivation of the DEFT architecture and inductive bias. Part II**
>
> Please see the following table (can also be found in our appendix as Table 6) where we carefully ablate our architectural choice and prove its success.
>
> | Parametrisation                                                                                                                                        	|   PSNR  |   SSIM  |
> |------------------------------------------------------------------------------------------------------------------------------------------------------------|:-------:|:-------:|
> | $\text{NN2}(x, \hat{x}_0, \nabla \ln p(y \hat{x}_0), t) + \text{NN1}(t) \nabla \ln p(y \|  \hat{x}_0)$ | $35.81$ | $0.876$ |
> | $\text{NN2}(x, \nabla \ln p(y \| \hat{x}_0), t) + \text{NN1}(t) \nabla \ln p(y \| \hat{x}_0)$          	| $35.74$ | $0.875$ |
> | $\text{NN2}(x, \hat{x}_0, A^*y, t)$                                                                                                         	| $34.04$ | $0.851$ |
> | $\text{NN2}(x, A^*y, t)$                                                                                                                      	| $26.62$ | $0.724$ |
>
> $\nabla =  \nabla_{\hat{x}_0}$ in the table above.
>
> As you can see, the added gradient-guided inductive bias boosts the performance of the DEFT objective significantly, showing that it is a good architectural choice.
>
> We have now provided both thorough empirical and conceptual motivations to the reviewer for our architecture. We are happy to continue clarifying where needed. We will add a detailed derivation in the revised version of the manuscript.
>
> [1] Chung et al. 2022. Diffusion posterior sampling for general noisy inverse problems. arXiv preprint arXiv:2209.14687.
>
> [2] Herbrich et al.. Bayes point machines. Journal of Machine Learning Research, 2001.
>
> [3] Poole et al. DreamFusion: Text-to-3D using 2D Diffusion, ICLR 2023.
>
> [4] Zhang and Chen, 2021. Path integral sampler: a stochastic control approach for sampling. arXiv preprint arXiv:2111.15141.

---

> ### Author Response · Authors · 2024-08-14
> **Small typo correction to - "Derivation of the DEFT architecture and inductive bias. Part I"**
>
> Dear Reviewer ESdx,
>
> We have just made some very small typo corrections (the cheap initialisation gradient we use in DEFT is with respect to $\hat{x}_0$ and there was a typo in the part I response ) to the above derivation  (Part I) detailing how we arrive at the additional term in our architecture from the h-transform.
>
> We hope that this makes our derivation / conceptual motivation of the architecture more clear and as before we thank you for your feedback and continued input.

---

### Official Review · Reviewer_qBFS · 2024-07-14

**Soundness:** 3
**Presentation:** 4
**Contribution:** 3
**Rating:** 5
**Confidence:** 4

**Summary:**

The paper tackles the problem of utilization of generative modelling to solve inverse problems. The main highlight is that the authors developed a technique that can solve inverse problems without the need for backward pass through the generative model. Hence enabling deriving the prior knowledge from even closed source models for solving inverse problems. Experiments are performed across multiple datasets to show the results. The results show that DEFT achieve a speed up and performance boost over some methods.

**Strengths:**

1. The paper introduces a new novel method for conditional generation thorough efficient fine-tuning of a small network using Doob's h-transform
2. The proposed method enables learning a solution for inverse problems without backpropagation through the diffusion network, hence enabling learning solution for inverse problems even from closed-source models since it doesn't require a backpropagation operation
3. By bypassing the backpropogation through the Diffusion U-Net, the method achieves a speed up over existing methods.
4. The paper is well written and extensive experiments are performed across multiple tasks to validate the effectiveness of the method.

**Weaknesses:**

1. Although the paper claims that existing baselines require backpropagation through U-Net for solving inverse problems (Ln 37-38). This is not always the case. I think some relevant baselines like

[1] Manifold Preserving Guided Diffusion
[2] FreeDoM: Training-Free Energy-Guided Conditional Diffusion Model

have missed the author's notice. These methods do not require any backpropagation through the UNet and hence are faster and perform better.

2. The baselines compared in the paper are very old and refer to works more than 1 years ago.
3. An analysis of the computational overhead caused due to the fine-tuning process is missing

**Questions:**

1. Could the authors give an analysis of the benefits of the method over [1,2] referred in weakness
2. I would also like to see the computational overhead in terms of memory and time involved in the fine-tuning process.

**Limitations:**

Yes, The limitations and potential negative impacts section looks reasonable to me.

---

> ### Author Rebuttal · Authors · 2024-08-06
>
> Thank you for pointing us to further baselines beyond the ones we provide that would provide an interesting axis of comparison to our method.
>
> ### Q1: Comparison against FreeDoM and MPGD
>
> **MPGD** [1]: MPGD proposes three variants: MPGD w/o projection, MPGD-AE and MPGD-Z. MPGD-Z requires a latent diffusion model and is not applicable to pixel-based diffusion. MPGD-AE requires to train an additional auto-encoder and thus, similar to DEFT, requires an additional dataset and training time. Unfortunately, the codebase for MPGD does not provide a pre-trained autoencoder, and we were unable to replicate the MPGD-AE variant to match the paper. We provide additional experiments of MPGD w/o projection on ImageNet for inpainting, super-resolution and HDR. In all these settings, DEFT is able to outperform MPGD w/o projection, see the following Table were we present the KID. DEFT also outperforms MPGD w/o projection on the other metrics, see the one page PDF.
>
> For the KID, we can see that DEFT outperforms MPGD w\o projection on both the linear (inpainting, super-resolution) as well as the non-linear task (hdr).
>
> | KID (↓)       	| Inpainting | Super-resolution | HDR   |
> |----------------|------------|------------------|-------|
> | MPGD w/o proj. | 3.02   	| 3.693        	| 3.571 |
> | DEFT       	| 0.29   	| 1.78         	| 0.10  |
>
> MPGD-AE defines an interesting conditional sampling approach. The use of an additional auto-encoder could also be used for DEFT to learn a conditional update in the latent space. This could possibly speed up the training. However, MPGD-AE requires training the auto-encoder, and thus an additional dataset is required, which is often much larger than our fine-tuning dataset, as an autoencoder such as VQVAE or VQGAN typically need a large amount of samples to train from scratch.
> Note, that the conditional update step of MPGD w/o projection mimics the initialization of DEFT (second term in Equation 13 of the manuscript), which can be derived by omitting the Jacobian of the unconditional diffusion model from the DPS [3] update.
>
> **FreeDoM** [2]: FreeDoM defines a distance-measuring function D. In the context of inverse problems and image reconstruction this is chosen as the negative log-likelihood (see also Section 3.3 and Appendix B in FreeDoM). In this case, the FreeDoM sampling scheme reduces to DPS [3], compare line 7 in Alg. 1 in FreeDoM against line 7 in Alg. 1 of DPS . Further, FreeDoM requires backpropagation through the trained score model, thus it would have the same computational cost as DPS. As we are already presenting comparisons against DPS as our reconstruction guidance baseline, we do not benchmark against FreeDoM.
>
>
>
>
> ### Q2: The baselines compared in the paper are very old and refer to works more than 1 years ago.
>  **A2:** We respectfully disagree on this point with the reviewer. While we compare against relevant, yet older, methods such as DPS (arxived Sep ‘22), we also compare against **RED-diff** (published at ICLR 2024 in May ‘24 with a corresponding complete arxiv on Sept. ‘23). Given that our submission to NeurIPS (May ‘24) falls within a year of this timeline, we believe RED-diff does not qualify as *very old*. At most, it is a year old, but in fact, the complete version (with the experiments we compare to) has only been available for about **nine months**.
>
> However, we thank the reviewer for pointing us to more recent methods that we could compare against, and we appreciate the references they have provided. We thank the reviewer for suggesting MPGD, (arxived Nov’23) which we now compare against in our rebuttal experiments (c.f. point above). We are also happy to run further experiments against other baselines that were released in more recent months if the reviewer has any suggestions. Based on suggestions by other reviewers, we have also added comparisons against further baselines such as Controlnet [5] and I2SB [4], providing a more thorough ablation (see general response).
> ### Q3: Computational overhead in terms of memory and time
> **A3**: In our image reconstruction experiments, we report the combined sampling and training time of DEFT in Table 1 and Table 2 of the submission. However, we only give the total time. However in the text for each experiment, we also have mentioned the split between finetuning and sampling time -
> > For DEFT, this computational time additionally includes the 3.9 hrs of training time of the h-transform additionally with the 1.2 hrs of evaluation).
>
> We will endeavor to make these numbers more clear in the text and the table.
> The size of the DEFT model is about 5-10% (depending on the task) the size of the pre-trained unconditional model. This means that during inference, the DEFT model incurs about an additional 5-10% memory overhead, whereas baselines such as DPS incur an additional memory overhead due to needing to backpropagate through the pretrained unconditional score model. In particular, the memory cost of DPS is O(sum_i^L  layer-width_i), where L is the number of layers on the unconditional model, as all activations have to be kept in memory for the backward pass. In contrast the memory cost of DEFT during inference is just O(max layer width) as only the forward pass is needed.
>
> [1] He et al., Manifold preserving guided diffusion. ICLR 2024
>
> [2] Yu et. al., Freedom: Training-free energy-guided conditional diffusion model. IEEE CVPR 2023
>
> [3] Chung et al., Diffusion Posterior Sampling for General Noisy Inverse Problems. ICLR 2023
>
> [4] Liu et al., I2SB: Image-to-Image Schrödinger Bridge, ICML 2023.
>
> [5] Zhang et al., Adding Conditional Control to Text-to-Image Diffusion Models. IEEE CVPR 2023

---

> > ### Author Response · Authors · 2024-08-12
> > **Further clarification on reported total cost of DEFT (fine-tuning + sampling)**
> >
> > To further aid the discussion and address the computational overhead comment raised, we would like to highlight some key points of our experiments.
> >
> > In Tables 1-3 in our paper, we report the total wall clock time for DEFT, which includes the sum of both training time (fine-tuning time) and inference time. Importantly, even when accounting for the fine-tuning time, DEFT still **take less time** than many inference-only or training-free methods such as DPS and REDDiff.
> >
> > Additionally, we believe that the memory complexity analysis presented in our rebuttal further clarifies our approaches efficiency. Please let us know if there are any other points we can clarify or if you have additional questions.

---

> > ### Comment · Reviewer_qBFS · 2024-08-13
> >
> > Dear author,
> >
> > I thank you for the detailed rebuttal. After going through the rebuttal, my concerns regarding the theoretical novelty is clear. have decided to improve my rating to BA. I believe the comparison with freedom and mid is not fair since these methods are training free and the authors were not give satisfactory explanation/ comparisons.

---

> > > ### Author Response · Authors · 2024-08-13
> > > **Title: Choice of comparison methods**
> > >
> > > Dear reviewer,
> > >
> > > Thank you for your feedback. We appreciate your decision to improve your rating and value your continued input and discussion.
> > >
> > > We understand your concerns regarding the comparisons with training-free methods such as MPGD and we will add a note to the manuscript making this explicitly clear so that no unfair conclusions can be drawn. It is not our intention to perform any unfair comparisons. We really want to make this clear and for it not to damage the perception of our work.
> > >
> > > We still think there is value in the comparison to training-free, and hope that you see it the same way since you suggested comparing to the training-free methods MPGD and FreeDoM in your first response. You can interpret DEFT as fine-tuning of a training-free method (see also the response to reviewer ESdx about the motivation of the network parameterization). Thus this comparison is almost like an ablation, i.e., how is method X improved when we fine-tune it this way.
> > >
> > > Moreover, it is important to note that many conditional sampling methods, while training-free, require setting hyperparameters, such as the time-dependent strength $\lambda(t)$ for DPS or RED-diff. As these methods can be quite sensitive to the choice of $\lambda(t)$, in practice a small dataset is often necessary to appropriately tune these hyperparameters.
> > >
> > > In addition, we see the ability of DEFT to leverage a small dataset as **a positive feature of our method**. In many practical applications such as the medical imaging or protein design setting described in our paper, there are small datasets specific to the task available. Leveraging these for optimal performance is a strength of the DEFT method.
> > >
> > > We would also like to add further clarification pertaining to the fair element of comparison. Many of the training free methods such as DPS require backpropagating through the score network over many iterations; from a computational budget perspective this is not so different to training (especially in the small-scale fine-tuning that DEFT requires, i.e. small fine-tuning network and small dataset). As you can see in our total wall clock time comparisons, such training free methodologies actually result in longer GPU hours overall than our fine-tuning approaches. Therefore,  we do believe this comparison to be very helpful as the validation compute time becomes **comparable / higher than our required finet-uning  time**.
> > >
> > > Finally, we would like to emphasize that we did add some trained comparisons (ControlNet and I2SB as proposed by the reviewers). To the best of our knowledge there are not many other applicable trained methods. We did try out all the methods suggested by reviewers and we are happy to commit to adding more comparisons in the final version of the manuscript if you think there are other suitable methods to compare against.

---

### Official Review · Reviewer_no8y · 2024-07-15

**Soundness:** 3
**Presentation:** 3
**Contribution:** 3
**Rating:** 7
**Confidence:** 3

**Summary:**

The paper proposes a new framework for fine-tuning unconditional diffusion models for conditional generation based on Doob’s $h$-transform. By utilizing a small set of observations and ground truth samples, the algorithm can learn the conditional $h$-transform that is used to

**Strengths:**

- The work is a novel approach to fine-tuning pre-trained unconditional diffusion models for conditional generation. The authors provide an extensive formulation of sampling from the posterior given a diffusion prior using Doob's $h$-transform. The proposed method has the potential to unify existing diffusion posterior sampling methods under a common framework.
- The work can be extended to non-linear inverse problems, which is a significant limitation of many existing diffusion posterior sampling methodologies.
- By learning the transform, the proposed algorithm significantly speeds up the conditional inference in comparison to existing methods. Many of the previous approaches also required backpropagating through the denoising network during inference, making their usage impractical in many applications.

**Weaknesses:**

- The method requires a non-negligible dataset of observations and samples to train the $h$-transform on. Although there is a significant speed advantage during inference, access to this dataset is not guaranteed for every task and there could be issues with generalization. I.e. for training a generic non-linear deblurring operator, the network that parametrizes the $h$-transform has to be trained on a diverse enough set of observations and images. This is an important limitation that some other methods (such as [9]) do not suffer from. Even in the stochastic optimal control case, which is training-free, using VarGrad or Trajectory Balance requires significant computational resources that can exceed the requirements of previous posterior inference approaches.

**Questions:**

- How do the $h$-transform networks perform on out-of-distribution samples? Given that they still contain a sizeable number of parameters how important is the number of samples used to train them in the final image quality?

**Limitations:**

The limitations have been addressed.

---

> ### Author Rebuttal · Authors · 2024-08-06
>
> We thank the reviewer for the valuable comments. We ran additional experiments, the complete results are available in Table 1 in the 1-page PDF.
>
> ### Q1: How does DEFT perform on OOD samples?
> **A1**: We evaluated DEFT, trained on ImageNet (Section 4.1), on a subset of 200 images of the ImageNet-O [1] dataset for both inpainting and HDR. The ImageNet-O dataset contains unique images that do not belong to any of the classes present in the original ImageNet dataset and is considered OOD. The results for inpainting are similar to the the evaluations of the main paper, see also Table 1 and Figure 3 in the rebuttal pdf. For a nonlinear inverse problem such as HDR, we still significantly outperform our best baseline RED-Diff, showing that our method is robust on images outside the training and finetuning distribution, and it has the potential to learn an agnostic solution to an inverse problem.
> ### Q2: Dependence on number of training samples and image quality?
> **A2**: As DEFT requires a dataset for fine-tuning, we ablate the number of training samples that can still result in good performance. We trained DEFT on a subset of 10, 100 and 200 ImageNet images for Inpainting. We see improvements of all metrics, when training on a larger dataset. For the KID, we can outperform RED-diff (KID: 0.86) even when trained on only 200 images.
> See also Figure 2 in the rebuttal pdf for an example reconstruction. Here, we can see that with increasing number of training samples, the resulting image looks more realistic. However, even with 10 images, we perform quite competitively, showcasing that our method is very sample-efficient when it comes to learning a conditional transform.
>
> See the following table for the results on inpainting on ImageNet. Here, we see that even with 10 samples we can outperform DPS w.r.t. to the KID and with 200 samples we outperform RED-diff.
>
> | DEFT, train on | 10 | 100 | 200 | 1000 (original) | | RED-diff | DPS |
> |----------------|----|-----|-----|-----------------|-|--------|-----|
> | KID        	| 1.85   | 0.978	| 0.401	| 0.29   | | 0.86 |15.2   |
>
>
> Further, we view the requirement of a fine-tuning dataset as a **feature**, not only as a limitation. While inference-time, training free methods (e.g. FreeDoM, DPS or RED-diff) have a ceiling on their performance when applied to new tasks or datasets, DEFT leverages fine-tuning to achieve high efficiency and performance even with a small dataset. We see that even with only 10 images, we can achieve a PSNR of 20.87 compared to 21.27 for DEFT. This efficiency is mostly due to the initialisation and network parameterization in Eq (13), where DEFT is initialized to mimic a cheap guidance term, similar to what is proposed in DPS or MPGD.
>
> ### Q3: Computational expense of VarGrad/Trajectory Balance for the online fine-tuning loss (Section 3.2)
>
> **A3**: We acknowledge that the online fine-tuning in its current form comes with a high computational cost. We included the online objective first and foremost to highlight the interesting connection of stochastic optimal control and conditional sampling.
> After the Neurips submission deadline Venkatraman et al. [2] published a fine-tuning approach for text-to-image diffusion models based on trajectory balance with a similar architecture (cf. eq. (11) in [2] with eq. (13) in our work). They were able to scale trajectory balance to higher dimensional problems by using off-policy training, i.e., re-using previous samples, and stochastic subsampling, i.e., calculating the gradient only for a randomly sampled subset of the trajectory. Similar tricks can be used for our online fine-tuning objective, to reduce the computational cost and scale it to high-dimensional settings. We believe that our proposed online objective can lead to interesting future work to scale up, similar to [2].
>
> [1] Hendrycks et al., Natural Adversarial Examples, IEEE CVPR 2021.
>
> [2] Venkatraman et al., Amortizing intractable inference in diffusion models for vision, language, and control, arXiv preprint (2024)

---

> > ### Comment · Reviewer_no8y · 2024-08-08
> > **Final Score**
> >
> > Thank you for the detailed responses.
> >
> > Considering the additional clarifications and results, I believe that this work could have significant impact and will be raising my score to reflect it.

---

### Author Rebuttal · Authors · 2024-08-06

We thank the reviewers for their valuable and thorough feedback, we will revise the paper accordingly. Below, we will address the main points and describe improvements we made to our submission.
#### Strengths and contributions
We appreciate the recognition of our work's **novelty** (reviewers `no8y, qBFS, ESdx`) in fine-tuning pre-trained unconditional diffusion models for conditional generation using Doob’s ℎ-transform. The **solid theoretical foundation** (reviewers`no8y, qBFS, ESdx, ZJuc`), **comprehensive experiments** (reviewers `qBFS, ESdx, ZJuc`), and **speed improvements during inference** (reviewers `no8y, qBFS`) were also noted by the reviewers.

#### Addressing weaknesses and reviewer questions
Please find detailed point-to-point responses to each reviewer in the corresponding thread. Here, we summarize the main points from these discussions:


1. **DEFT requires a fine-tuning dataset**: We acknowledge that DEFT requires a fine-tuning dataset for training the h-transform. In applications where only a handful of fine-tuning examples are available this may limit performance. Yet, our additional experiments show that even in low data settings (n=100,200) DEFT is able to produce similar results to DPS or RED-diff. For many other applications for which fine-tuning datasets are available (conditional protein design, medical imaging) the fine-tuning phase of DEFT allows the model to **benefit from this data** (as compared to inference-time-only, training-free strategies such as DPS) leading to improved performance, as demonstrated in the experiments in our paper, with comparable or faster total evaluation time on the eval dataset even when taking the fine-tuning time into account (as the faster DEFT inference makes up for the computational overhead of fine-tuning).
In the experiments on conditional protein design, we saw that reconstruction guidance methods such as DPS often fail on giving convincing results, whereas already a small fine-tuning dataset can improve the performance for DEFT.


2. **DEFT fine-tuning has a computational overhead**: Reviewers rightly point out that DEFT requires an initial training phase. This is common across many fine-tuning methods. However, the overall computational cost, combining both training and inference phases, may vary depending on the specific training budget and application.
We demonstrate that DEFT provides a fast and efficient sampling during inference time. This speed-up during inference helps to offset the initial training time. To illustrate this, our experiments include the total sampling time, which encompasses both training and inference phases (see Table 1 and Table 2 in the manuscript). These results show that DEFT is comparable to or faster than other baseline methods, such as DPS and RED-Diff, when the goal is to sample 1000 images. In the revised version, we include a detailed breakdown of the computation time, not just the total time in the tables.

3. **Additional experiments for rebuttal**:
According to the comments of reviewers, we performed a variety of experiments on the ImageNet dataset. We evaluated:
- *MPGD [2] on inpainting, hdr and super-res* as a new training-free baselines, which does not require to backpropagate through the diffusion model. Our results show that DEFT is able to outperfrom MPGD.
- *ControlNet [4] on inpainting, hdr and super-res* as a comparison to a different fine-tuning method. Our results show that DEFT is able to outperform ControlNet on these image reconstruction tasks.
- *I2SB [3] on inpainting and super-res* as an example of a fully-trained conditional diffusion model. Here, our results show that I2SB outperform DEFT on some tasks. However, I2SB was trained on the full ImageNet training set with a big compute budget, whereas DEFT was trained on a subset of 1000 images.
- *DEFT on inpainting, trained on a subset of 10, 100 and 200 images* to study the effect of the size of the training set. Here, we see that a larger training set, results in better image quality. However, already with 100 images is DEFT comparable to training-free methods as RED-diff or DPS.
- *DEFT on inpainting, hdr for out-of-distribution data (ImageNet-O [1])* to study the generalisability. On ImageNet-O the quality metrics deteriorate. However, we are still competitive to RED-diff.

**A table with all new results and some examples can be found in the 1-page PDF**, notice we strongly outperform the other requested inference-time baselines and ControlNet whilst having comparable to slightly worse performance compared to fully conditional training methods like I2SB, with significantly smaller models, training time and datasets.


[1] Hendrycks et al., Natural Adversarial Examples, IEEE CVPR 2021.

[2] He et al., Manifold preserving guided diffusion. ICLR 2024

[3] Liu et al., I2SB: Image-to-Image Schrödinger Bridge, ICML 2023.

[4] Zhang et al., Adding Conditional Control to Text-to-Image Diffusion Models, IEEE CVPR 2023

---

> ### Author Response · Authors · 2024-08-14
> **Final Rebuttal Response**
>
> Thank you to all of the reviewers for engaging with our rebuttal and the discussion period. We'd like to highlight a few of the comments that came up during the discussion period:
>
>
> 1. “I believe that this work could have significant impact.” - Reviewer no8y
> 2. “the strengthened experimental results demonstrate significant value in this work.” - Reviewer ESdx
> 3. “comprehensive clarifications and additional comparisons. I will maintain my score.” - Reviewer ZJuc
> 4. “my concerns regarding the theoretical novelty is clear. have decided to improve my rating” - Reviewer qBFS
>
> We are glad that we were able to run additional experiments and provide further clarifications to the reviewers to address their initial questions. Furthermore, we thank the reviewers for a productive rebuttal period and their continued engagement with our work.
>
> We believe the main remaining points raised by the reviewers are as follows, and we summarize our clarifications to them below -
>
> > “I believe the comparison with freedom and mid is not fair since these methods are training free and the authors were not give satisfactory explanation/ comparisons.”
>
> We have now provided several comparisons to methods that are not training free (such as I2SB and ControlNet), furthermore we have demonstrated how the compute budget (total GPU time) on many of these training free methods exceeds the highly efficient training done by DEFT, thus from a compute perspective there is a value  in these comparisons. Finally comparisons to methods such as MPGD serve as an ablation to DEFT as this showcases how extra fine-tuning such as the one we do improves on MPGD.
>
> We will be super clear in the final version of the manuscript when mentioning which methods are trained/fine-tuned and which methods are not so that no unfair conclusions can be drawn.
>
> > “I find some aspects of the DEFT network parameterization, including the rationale behind it and the inductive bias, to be unclear. “
>
> We provided a detailed explanation in response to reviewer ESdx carefully deriving (from the h-transform) the cheap “warm start” inductive bias in the DEFT architecture (i.e. $\mathrm{NN1}(t) \nabla_{\hat{x}_0} \ln p(y|\hat{x}_0)$). We then discuss how this term empirically leads to a
>
> * Lower loss / Better initialisation
> * Faster and better training  (Converges in less epochs)
> * Quickly (in early epochs) tunes $\mathrm{NN1}(t)$ to a reasonable/ expected guidance scale (low near the noise, then smoothly increases as we reach the target/data)
>
> Intuitively we also discuss how this architecture brings a trade off between compute (cheap approximate MPGD styled warm start) and accuracy - learns a final network $\mathrm{NN2}(x,y,t)$ which approximates the h-transform without strong assumptions, yet leveraging the cheap warm start. Finally we provided the reviewer with an ablations table from the manuscript empirically demonstrating the success of this architecture.
>
>
> We hope these additional points address the remaining issues the reviewers might have had with our submission, and we thank the reviewers and the AC for an excellent NeurIPS reviewing experience!

---

### Decision · Program_Chairs · 2024-09-25

**Decision:**

Accept (poster)

**Comment:**

This paper proposes a novel approach based upon the Doob's h-transform for turning an unconditional generative diffusion process into a conditional one.

The reviewers all appreciate the paper's originality and empirical work with some still seeing the paper as borderline. The authors have gone out of their way to answer the reviewers also providing new benchmarking. Taking together it lends to convince that the paper deserves to be presented at Neurips and that given the popularity of the topic, the paper can have a substantial impact.

In this AC's opinion the developed methodology can be presented more concisely. So the authors are encouraged to work on this for the final version.